



# Single particle measurements of bouncing particles and in-situ collection efficiency from an airborne aerosol mass spectrometer (AMS) with light scattering detection

Jin Liao[1,2,a], Charles A. Brock[1], Daniel M. Murphy[1], Donna T. Sueper[3], André Welti[1,2,b], Ann M. Middlebrook[1]

[1]NOAA Earth System Research Laboratory (ESRL), Chemical Sciences Division, Boulder, CO 80305, USA
[2]Cooperative Institute for Research in Environmental Sciences, University of Colorado at Boulder, Boulder, CO 80309, USA
[3]Aerodyne Research Inc., Billerica, MA 01821, USA
[a]Now at: Universities Space Research Association, Columbia, MD 21046, USA and NASA Goddard Space Flight Center, Atmospheric Chemistry and Dynamic Laboratory, Greenbelt, MD 20771, USA
[b]Now at: Leibniz Institute for Tropospheric Research, Department of Physics, Leipzig, 04318, Germany

*Correspondence to*: Ann M. Middlebrook. (ann.m.middlebrook@noaa.gov)

**Abstract.** A light scattering module was coupled to an airborne, compact time-of-flight aerosol mass spectrometer (LS-ToF-AMS) to investigate collection efficiency (CE) while obtaining non-refractory aerosol chemical composition measurements during the Southeast Nexus (SENEX) campaign. In this instrument, particles typically larger than ~ 250 nm in vacuum aerodynamic diameter scatter light from an internal laser beam and trigger saving individual particle mass spectra. Over 33,000 particles are characterized as either prompt (27%), delayed (15%), or null (58%), according to the appearance time and intensity of their mass spectral signals. The individual particle mass from the spectra is proportional to the mass derived from the vacuum aerodynamic diameter determined by the light scattering signals ($d_{va-LS}$) rather than the traditional particle time-of-flight (PToF) size ($d_{va}$). The delayed particles capture about 80% of the total chemical mass compared to prompt ones. Both field and laboratory data indicate that the relative intensities of various ions in the prompt spectra show more fragmentation compared to the delayed spectra. The particles with a delayed mass spectral signal likely bounced on the vaporizer and vaporized later on a lower temperature surface within the confines of the ionization source. Because delayed particles are detected at a later time by the mass spectrometer than expected, they can affect the interpretation of PToF mass distributions especially at the larger sizes. CE, measured by the average number or mass fractions of particles optically detected that have measureable mass spectra, varied significantly (0.2-0.9) in different air masses. Relatively higher null fractions and corresponding lower CE for this study may have been related to the lower sensitivity of the AMS during SENEX. The measured CE generally agreed with the CE parameterization based on ambient chemical composition, including for acidic particles that had a higher CE as expected from previous studies.





# 1 Introduction

Aerosol size, chemical composition and mass loading are important to estimate the impact of aerosols on directly scattering sunlight or being cloud condensation nuclei to indirectly affect radiation balance and climate (e.g. Ramanathan et al., 2001). The spatial and temporal distribution of ambient aerosols is highly inhomogeneous, owing to different sources, meteorological
conditions, atmospheric processes, and their relatively short atmospheric lifetime compared to greenhouse gases. The Aerodyne aerosol mass spectrometer (AMS) is a fast time response instrument capable of quantifying size resolved non-refractory aerosol chemical composition (e.g. Jayne et al., 2000; Jimenez et al., 2003; Drewnick et al., 2005; Canagaratna et al., 2007) and has been widely used to measure the real time aerosol ensemble organic, sulfate, nitrate, ammonium, and chloride (non-sea salt) mass loadings globally (e.g. Zhang et al., 2007a; Jimenez et al., 2009). Evaluation of aerosol mass accuracy
measured by the AMS is important to estimate the impact of aerosols on climate, biogeochemical health, and aerosol formation processes such as aerosol hygroscopicity (Levin et al., 2014; Brock et al., 2016) and aerosol acidity (Hennigan et al., 2015; Zhang et al., 2007b).

The basic principle of the AMS method is to focus ambient aerosols with an aerodynamic lens onto a hot vaporizer and analyze
the evolved gases with an electron-impact ionization mass spectrometer. Not all particles introduced to the inlet are vaporized and ionized. A varying collection efficiency (CE) of particles by the AMS potentially introduces large uncertainty in AMS measurements. CE is the ratio of the mass (or number) of particles detected by the AMS to that of particles introduced into the inlet (Matthew et al., 2008). It ranges from 0.3 to one in ambient measurements (Middlebrook et al., 2012) and therefore may induce an uncertainty as large as a factor of 3 in the aerosol mass measured by AMS. CE less than 100% in AMS measurements
was previously demonstrated by comparing aerosol mass loadings measured by the AMS with that measured by other instruments such as the Particle-Into-Liquid Sampler combined with an Ion Chromatography analyzer (PILS-IC) or an optical particle counter (ultra-high sensitivity aerosol spectrometer or UHSAS) (e.g. Takegawa et al., 2005; Middlebrook et al., 2012). Beam width probe experiments (Huffman et al., 2005) found that CE less than 100% is not due to particle beam broadening but is likely due to particles bouncing on the vaporizer. Laboratory and field studies showed that CE values depend on aerosol
chemical composition and relative humidity (Matthew et al., 2008; Middlebrook et al., 2012). Based on this, a parameterization for the composition-dependent CE was developed (Middlebrook et al., 2012) and is now being applied to ambient AMS measurements. In situ CE measurements and evaluation of the CE parameterization are therefore important to reduce the uncertainty in the AMS measurements.

A light scattering (LS) module has been developed to integrate into AMS instruments (LS-AMS) to detect single particles before they impact on the vaporizer (Cross et al., 2007). This provides an opportunity to directly investigate the in situ CE of the AMS by comparing the number or mass of particles optically and chemically detected to the total number or mass of particles optically sensed. Using the LS-AMS instrument, the CE for ambient particles at three ground sites near Mexico City



(Cross et al., 2009), Bakersfield, CA (Liu et al., 2013), and downtown Toronto (Lee et al., 2015) was about 0.49, 0.52 and 0.37, respectively. The CE for the Mexico City ground site varied only about ±10% over the full sampling period (Cross et al., 2009), which may be due to relatively constant ambient aerosol chemical composition there. Airborne studies of air masses with widely different chemical composition provide an opportunity for investigating the capability of LS-AMS in capturing

CE variations.

Beside the ability to determine in situ CE, the LS-AMS has also been used to derive particle density by comparing the optical size with the vacuum aerodynamic size (Cross et al., 2007), distinguish single particle chemical composition types (Cross et al., 2007; Liu et al., 2013; Freutel et al., 2013), particle internal and external mixing properties (Robinson et al., 2013), and

validate the interpretation of AMS factors from a positive matrix factorization using cluster analysis of the LS module data (Lee et al., 2015). As this work aims to use LS-AMS to investigate AMS measurement uncertainties, analysis regarding the above perspectives is not included.

This study provides the first airborne single particle measurements from an LS-ToF-AMS instrument. These measurements

were performed onboard the NOAA WP-3D aircraft sampling various air masses over the continental United States during the Southeast Nexus of Air Quality and Climate (SENEX) campaign in May and June 2013 (Warneke et al., 2016). Many airborne studies have reported ambient single particle properties (e.g. chemical composition types and internal and external mixing states) (Murphy et al., 2006; Murphy et al., 2003; Cahill et al., 2012; Pratt and Prather, 2010). This study focuses on using single particle data to investigate airborne AMS measurement uncertainties. The CE was measured by LS-ToF-AMS during

this field study and compared to the CE parameterization based on the aerosol chemical composition and relative humidity (Middlebrook et al., 2012). The single particle data are also used to examine particle bouncing on vaporizer and the impact of "delayed" particles on the chemical ion signals and the traditional AMS size resolved mass distribution.

## 2 Experimental

For the SENEX field project, a compact time-of-flight aerosol mass spectrometer (AMS, Aerodyne Inc., Billerica,

Massachusetts) was integrated aboard the NOAA WP-3D aircraft with a pressure-controlled inlet (Bahreini et al., 2008) and an LS module. With the LS module, the AMS is capable of not only measuring the ensemble chemical composition and traditional size resolved mass distribution but also detecting the chemical composition and size of single particles with light scattering intensities above threshold. Detailed descriptions of the AMS instrument are provided in previous publications (Jayne et al., 2000; Jimenez et al., 2003; Drewnick et al., 2005; Bahreini et al., 2009) and only the differences are pointed out

here. Compared to most AMS instruments, our instrument has a longer chamber, with a distance from the chopper wheel to the laser beam of 26.5 cm and from the chopper wheel to the particle vaporizer of 39.5 cm, the locations of which are shown in Figure S1. The added light scattering module is very similar to that described in Cross et al. (2007) and Cross et al. (2009),



with the single particle data acquisition triggered by the light scattering signals as described by Liu et al. (2013). The major differences in the configuration were a smaller and more rigid optical table mounted directly onto the AMS chamber with a cover and locking screws to mount ellipsoidal and external mirrors and laser to ensure the stability of the laser and optical mirrors during airborne measurements. In order to fit the AMS with the LS module into the aircraft, we built a shorter extension
for the laser beam dump, which was redesigned as small, baffled, multi-angled chamber containing several knife edges, all painted matte black.

The data acquisition software version used during SENEX was 4.0.30, which included hourly measurement of both the single ion area and the detector baseline along with the mass scale calibration. The threshold setting for saving MS signals with the
AP240 data acquisition card (Acqiris, Geneva, Switzerland) was two bits above baseline for all the data. In addition to being checked hourly during each flight, the integrated detector signal for single ion pulses (single ion area in units of bits-ns/ion) was measured during preflight and postflight. Raw mass spectral signals are digitally recorded by the data acquisition card in units of bits-ns/extraction and in post-processing are converted into ions/s with the ToF pulser period/extraction, the number of extractions (adjacent ToF pulses) that were combined, and the single ion area calibration.

Although the mass spectrometer was tuned prior to the field project, the microchannel plate (MCP) detector set was nearing the end of its life and was replaced in the middle of the project on June 17. As a precaution, the new MCP set was initially operated with a reduced gain voltage that was increased several times during the rest of the project. Because of this, the detector sensitivity changed with the highest sensitivity (where the single ion area $\geq$ 13 bit-ns/ion) for flights starting from June 26 to
July 8. However, this sensitivity is still relatively low compared to our other field projects; for example, the single ion area for a recent ground-based study (Öztürk et al., 2013) was typically higher than 20 bit-ns/ion. Furthermore, the single ion peak shape is not ideal for our c-ToF due to ringing on the higher ion time-of-flight side. Consequently, low detector sensitivity has a non-linear effect on low ion signals, which was previously reported by Hings et al. (2007).

Two operating modes of this AMS instrument are the same as instruments without an LS module. The mass spectrum (MS) mode is used to measure ensemble submicron aerosol mass concentrations and particle time-of-flight (PToF) mode is used to measure size dependent submicron mass concentrations. The added light-scattering single particle (LSSP) mode with the LS-AMS is used to measure single particle size and mass. During each flight, the AMS was run alternatively among the MS mode, PToF mode, and LSSP mode with one cycle every 5 minutes. During the first 270 seconds of the cycle, the instrument switched
between MS (background for 2 seconds with the particle beam blocked and sampling for 4 seconds) and PToF (3.5 seconds with a chopped particle beam) modes, saving ensemble data roughly every 10 seconds. Then the instrument was run with the LSSP mode for last 30 seconds. The chopper wheel has a 2% duty-cycle slit and rotated with a frequency of ~110 Hz during both LSSP and PToF modes. The mass spectrometer was pulsed at a frequency of 62.5 kHz and mass spectra were added together for two ToF pulser periods prior to saving, resulting in 32 μs between mass spectra for each saved LSSP or PToF





chopper cycle. The PToF data were combined for all chopper cycles during the saving period resulting in size resolved mass distributions over the full submicron size range whereas LSSP data were saved for each chopper cycle when LS-triggered particle scattered light signal above threshold. Table 1 summarizes the relevant LSSP parameters and definitions used here and in other recent studies.

In LSSP mode, the particle beam is chopped and particles traverse an unfocussed beam of a continuous, solid-state, 405-nm-wavelength, 50 mW laser (CrystaLaser model DL405-050-0 with a CL-2005 power supply, Reno, NV) placed perpendicular to the particle beam prior to particles entering the vaporization/ionization section of the AMS vacuum chamber. The scattered light from the particles is focused with an ellipsoidal mirror onto a photomultiplier tube (PMT). As first employed by Liu et

al. (2013), the scattered light intensity is monitored and when above a specified threshold it triggers saving all of the chemical ion signals (mass spectra) obtained during the course of the current, ~9.1-ms chopper cycle along with the corresponding scattered light signal from the PMT. For each triggered event, the mass spectra from that chopper cycle are analyzed during post-processing. The spectrum with the maximum MS signal is located and then integrated within the adjacent ±5 mass spectra of that maximum to generate the total chemical ion signal for the triggering particle.

The cases when more than one particle passes through the chopper slit per chopper cycle with scattered light signals above the threshold are called coincident particles. Flight average coincident particle fractions ranged from less than 1% to about 3% in this study. Unfortunately, the analysis software currently does not correctly account for spectra in coincident cases because only one spectrum of the maximum MS signal is generated per chopper cycle. This spectrum can be from either of the particles

since the timing of the scattered light signals does not necessarily correspond to the magnitude of chemical ion signals. Hence, the coincident particle data are excluded in the following sections.

There are three potential ways to measure single particle size in LSSP mode: 1) velocities using particle arrival time from maximum mass spectral signal and distance from chopper slit to vaporizer (a.k.a. the traditional AMS vacuum aerodynamic

diameter or $d_{va}$), 2) velocities using particle arrival time from the maximum scattered light signal and the distance from the chopper slit to the laser beam (vacuum aerodynamic diameter $d_{va\text{-}LS}$), and 3) scattered light intensities for individual particles. The third measurement of particle size, the optical diameter or $d_o$, can be obtained from a calibration of scattered-light intensity and compared to $d_{va}$ (Cross et al., 2007). As we show in Section 3.1, $d_o$ is not the optimal measurement for size in this system. Also, the two measurements of vacuum aerodynamic diameter ($d_{va}$ and $d_{va\text{-}LS}$) provide additional information on how particles

are detected by the AMS.

Vacuum aerodynamic diameter ($d_{va}$) is traditionally defined for the AMS by Eq. 44 from DeCarlo et al. (2004):

$$d_{va} = d_m \times \frac{\rho_{eff}}{\rho_0} \tag{1}$$



where $d_m$ is the electrical mobility diameter, $\rho_{eff}$ is the effective particle density (in g cm$^{-3}$), and $\rho_0$ is the standard density (= 1 g cm$^{-3}$). Polystyrene latex spheres of known sizes are typically used to calibrate $d_{va}$ in the AMS as a function of particle velocity using the PToF time ($t_{ms}$) and the distance between the chopper wheel and the vaporizer ($L_{vp}$). In this manner, mass distributions as a function of $d_{va}$ or $d_m$ can be determined from AMS instruments in PToF mode without the LS module (Jayne et al., 2000).

5 Note that the measured PToF time ($t_{ms}$) includes an additional brief amount of time needed for particles to vaporize, the neutral molecules to move from the vaporizer to the electron beam, ions to move from the ion source to the orthogonal extraction region, and different signal paths in the data acquisition system. For prompt particles and not including the uncertainty in the time for particles to pass through the chopper, the slowest of these processes and largest uncertainty in PToF sizing is due to vaporization (D. Day, personal communication, November 16, 2015).

The second method of determining size using an LS-AMS is the vacuum aerodynamic diameter from the timing of the scattered light signal ($d_{va-LS}$). This size can be directly calibrated with known particles in a similar manner to $d_{va}$, where the calculated particle velocity uses the time of the maximum scattered light signal ($t_{LS}$) and the distance between the chopper wheel and the laser beam ($L_{LS}$). In this case, $t_{LS}$ does not include additional time for detecting mass spectral signals, so the calibration coefficients will vary slightly from the traditional $d_{va}$ calibration. Alternatively, $d_{va-LS}$ can be determined using the same calibration values as $d_{va}$ after accounting for the additional time. Assuming that the velocity of a particle is constant in the vacuum chamber, the estimated arrival time at the vaporizer, $t_{est}$, is calculated from the time the particle passes through the laser beam as:

$$t_{est} = t_{LS} \times \frac{L_{vp}}{L_{LS}} \qquad (2)$$

20 where $t_{LS}$ and $L_{LS}$ are defined above and $L_{vp}$ is the distance between the chopper wheel and the vaporizer. For prompt particles (e.g., ammonium nitrate), a histogram of the time differences between the maximum mass spectrum signal time ($t_{ms}$) and the estimated arrival time ($t_{est}$) has a Gaussian distribution (Figure 1). The mean of this distribution is the offset time ($t_{offset}$), and this value for SENEX was 0.35 ms (see Table 1). Twice the width of the Gaussian distribution for prompt particles is approximately the time available for particles to pass through the chopper slit; here it is ~ 0.12 ms. Particles with $t_{ms} > (t_{est} +$ 25 $t_{offset}) + 3\times$the Gaussian width are defined as "delayed" (see Table 1 and Sections 3.1.1 and 3.1.2 below) and are represented by particles on the right hand side of the cyan line at 0.53 ms in Figure 1. $d_{va-LS}$ is then obtained by the $d_{va}$ laboratory calibration values and the derived particle velocity accounting for the offset time as:

$$\text{velocity} = \frac{L_{LS}}{t_{LS} + t_{offset} \times \frac{L_{LS}}{L_{vp}}} \qquad (3)$$

It is worth noting that the laser not only counts and allows saving scattered light and chemical signals of particles above a 30 light-scattering threshold set in the data acquisition software in LSSP mode, but also counts the number of all sampled particle above this threshold in MS (both sampling and background) and PToF modes. The ratio of these LS counts per second in LSSP mode to that in the adjacent MS mode can be used to calculate the light scattering duty cycle due to dead time while saving individual LSSP events. The LSSP light scattering duty cycle was number-concentration dependent with an average value of



35% compared to MS mode. Therefore, each LSSP mode measured single particle mass or number was normalized by the average light scattering duty cycle factor from the preceding and following MS cycles to account for the dead saving time. These internal LS counts when sampling in MS mode are also compared below to number concentrations from independent particle number distribution measurements.

The ionization efficiency of the instrument was calibrated with pure, dry ammonium nitrate particles several times before, during, and after the field project. The Igor ToF AMS calibration analysis software version 3.1.5 was used with the PToF-calibrated size ($d_{va}$) to calculate the nitrate ionization efficiency (IE). When all of the calibration data were combined, the ionization efficiency was linearly proportional over a wide range of detector sensitivities to the airbeam (AB) signal at *m/z* 28

with a slight offset: IE = $1.29 \times 10^{-7}$ + $1.24 \times 10^{-12} \times$AB. For a typical AB value of $4.5 \times 10^5$ Hz, the IE for nitrate was about $7 \times 10^{-7}$ ions/g.The unit mass resolution MS and PToF data were analyzed using the Igor ToF AMS analysis toolkit (a.k.a. Squirrel) version 1.52L. Five coefficients from the standard AMS fragmentation table described by Allan et al. (2004) were adjusted for each flight for the measured fragmentation pattern from water and measured contributions of various species to the filtered air signals (see Table S1 for values). The current relative ionization efficiency (RIE) default values for sulfate,

organic and non-refractory chloride of 1.2, 1.4 and 1.3 were used except that the values of nitrate and ammonium were changed to 1.05 and 3.9. The LS data were processed using the Igor LS analysis toolkit (a.k.a. Sparrow) version 1.04F, with a modification to account for the longer particle flight chamber.

The LS-ToF-AMS was onboard on NOAA WP-3 aircraft and flew over the continental US to sample a variety of air masses

during the Southeast Nexus (SENEX) campaign from June to July 2013, as part of a large collaboration study Southeast Atmospheric Study (SAS). Detailed information of the field campaign is provided in Warneke et al. (2016) and http://www.esrl.noaa.gov/csd/projects/senex/. Over the course of the project, there were 17 research flights. Besides the AMS measurements presented here, dry particle number distributions from ~ 0.07 to 1.0 μm were measured by an ultra-high sensitivity aerosol size spectrometer (UHSAS) (Brock et al., 2016) onboard the same aircraft and are used to derive the particle

mass distributions for comparison in this study. Since the AMS sensitivity was poor at the beginning of the field project and UHSAS data were not available for the last flight, the data reported here are from 7 flights from June 26 until July 8 (Flights 10-16).



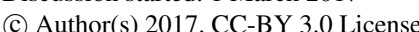


## 3 Results and discussion

### 3.1 Particle bounce at the vaporizer

#### 3.1.1 LSSP mode particle size measurements and indication of particle bouncing

Particle light scattering signal can be used as an indicator of particle size. In LSSP mode, light scattering signals above a certain

threshold in the data acquisition software were saved and also triggered saving of their mass spectra. This threshold was set low to save particles with light scattering signals near the detection limit and therefore include some saved data that were triggered by noise (<10%). The correlation between single particle maximum light scattering signals and the derived particle diameter $d_{va\text{-}LS}$ from velocity (see method section 2) for all LS-triggered events (red, above the optical detection limit, and black, below the optical detection limit) for the flight on July 6, 2013 is plotted in Figure 2a. The optical detection limit for

SENEX is defined as the maximum light scattering intensity > 0.04 and a signal-to-noise ratio (S/N) ≥ 3. The smallest particle with light scattering signal above this limit was ~170 nm in $d_{va\text{-}LS}$, which is close to what was reported by Liu et al. (2013) of 180 nm for slightly different criteria (signal-to-noise S/N ≥ 5). Using signal-to-noise ratio (S/N) ≥ 5 only as criteria for our dataset results in less triggers below this threshold. $d_{va\text{-}LS}$ has a positive correlation with light scattering signal intensity as expected. However, because the laser beam is designed to be broad to capture the particles, the light scattering intensities

varied over a significant range for the same size particles. Therefore the light scattering intensities were only used as a diagnostic and not used to derive particle size in this study.

To calculate $d_{va}$ from individual particles, they must have viable chemical ion signals (mass spectra). The chemical ion signal detection limit varied from flight to flight depending on detector sensitivity, and for the flight data shown in Figure 2b it was

600 bit-ns/particle with a single ion area = 16.9 bit-ns/ion or about 36 ions in the individual particle mass spectra. A significant fraction of single particles with detectible light scattering signals have either no chemical signal or a chemical signal arriving later than expected. The arrival time of single particles as measured by the maximum mass spectral signals is plotted against the time of maximum light scattering signals for the flight on July 6, 2013 in Figure 2b. The corresponding $d_{va}$ and $d_{va\text{-}LS}$ values are plotted on the right and top axis. The solid line, defined as $t_{ms} = t_{est} + t_{offset}$, is the expected mass spectrum arrival time. The

slope of the solid line is determined by the ratio of the distances in the AMS between the chopper wheel, laser beam, and vaporizer and the intercept is the same offset time as indicated in Figure 1. The dashed line, defined as ($t_{ms} = t_{est} + t_{offset}+3\times$Gaussian width), is used to distinguish the particles with mass spectral signal arriving significantly later than expected. This delayed maximum mass spectra signal arrival time varied over a wide range, from 0.02 –3.1 ms with an average of 1.0 ms. The time delays are also apparent in the Figure 1 histograms of ammonium sulfate, mixed composition, and SENEX data.


Our data are consistent with the hypothesis that delayed particles represent particles that bounce off the vaporizer and subsequently vaporizing off of another surface in the source region (Cross et al., 2009). The delays are too long to be explained solely by the time it takes particles to vaporize, for example $(NH_4)_2SO_4$ in <50 µs (Drewnick et al., 2015). Laboratory data




show that ammonium nitrate, with no measurable bounce, has no delayed particles either (Figure 1). The ion source is less than two centimeters in size, so particles that are delayed 500 to 1000 µs before hitting another surface would have velocities of a few to 20 m s$^{-1}$. Given the range of particle velocities measured in the PToF region prior to particle impaction on the vaporizer, this represents a loss of over 90% of the initial kinetic energy upon a bounce or multiple bounces within the vaporizer. Particles without detectible chemical signals could be those that bounced far away from the ionization source cage region and could not be vaporized and ionized efficiently.

The ion source is designed so that many of the molecules leaving the vaporizer pass through the electron beam. Some bouncing particles will pass through the electron beam and such particles would acquire significant charge (Ziemann et al., 1995). At velocities below very roughly 10 m s$^{-1}$, the charged particle trajectories can be considerably modified by the electric fields inside the source region, possibly affecting what happens to particles that are not detected promptly by the mass spectrometer. For example, if a 150 nm $d_{va}$ particle with a velocity of 5 m s$^{-1}$ after bouncing goes through the electron beam, it will probably be pulled through the ion extraction hole. Particles first entering the ionization source are traveling too fast to be deflected from their initial paths if they are charged in the electron beam. Detailed modeling of this effect is beyond the scope of this paper.

### 3.1.2 Fractions of prompt, delayed, and null particles

According to the chemical ion signal intensities and the relationship between expected and real chemical ion signals arrival time, single particles detected in LSSP mode are classified as prompt, delayed, and null. The criteria for each of these classifications and comparison among different studies are presented in Table 1. An optically detected particle that has chemical ion signals below the threshold is classified as null. Optically and chemically detectible single particles are classified as "prompt" or "delayed" according to their arrival time of maximum mass spectra signal below or above the dashed line in Figure 2b, which are also defined by the time differences in the Figure 1 histograms. Figure 3 shows the particle types (prompt, delayed, and null) as a function of $d_{va-LS}$ for the research flights from June 26 to July 8. On average the prompt, delayed and null fractions are 27%, 15%, and 58%, respectively, in this study. The respective fractions are 23%, 26%, and 51% in Cross et al. (2009) and 46%, 4%, and 48% in Liu et al. (2013). Due to the improvement of LS module and its data analysis software, the definition to separate prompt and delayed particles in this study is the same as the more recent study by Lee et al. (2015) but different from that in Cross et al. (2009) and Liu et al. (2013) (see table 1 for comparison). The definition in Cross et al. (2009) would likely result in more delayed particles compared to this study and a consistently higher delayed particle fraction was reported in their study. The definition in Liu et al. (2013) is difficult to directly compare to this study without information about their offset time. The prompt + delayed fractions are about 50% for both Cross et al. (2009) and Liu et al. (2013). In this study, the combined fractions are slightly lower which may be due to a lower sensitivity leading to higher null rates in addition to potentially different chemical detection thresholds.





Prompt and delayed fractions decline at small ($d_{\text{va-LS}} < 350$ nm) or large ($d_{\text{va-LS}} > 550$ nm) size particles. The single particle mass is near the chemical signal detection limit in this study due to low sensitivity, so the smallest particle size ($d_{\text{va-LS}} = 250$ nm) shown here is larger than $d_{\text{va}} = 200$ nm observed by Cross et al. (2009) and particles with $d_{\text{va-LS}} < 250$ nm in this study are mostly null particles (not shown). This is further supported by lower prompt + delayed fractions at the smallest particle size,

depicted in Figure 3, compared to that observed by Cross et al. (2009) and the prompt + delayed fractions at the smallest particle size were even lower for the flights (not shown here) with a lower MCP gain. The LSSP measurements reported for the Bakersfield study had a larger fraction of null particles at the smallest sizes (Liu et al., 2013) and it is unclear if that may also be related to sensitivity. The reduced prompt + delayed fractions at the largest sizes are similar to what was observed by Cross et al. (2009) and Liu et al. (2013) and are likely due to the larger particles having more kinetic momentum when arriving

at the vaporizer or containing more refractory material (e.g. dust) as suggested by Cross et al. (2009). The maximum fraction of delayed particles appeared at a larger size (525 nm) than that of prompt particles (375 nm), which may be a result of more bouncing at larger size.

### 3.1.3 Chemical signal differences between prompt and delayed particles

Because this study had a significant fraction of delayed particles (about a third of the particles with chemical signals), potential

differences in their chemical composition were explored. The average mass spectra of prompt and delayed particles for all SENEX flights analyzed are plotted in Figure 4a. Delayed particles have relatively higher organic signals at $m/z$ 43, 45, and > 60, sulfate signals at $m/z$ 98 and 81, and nitrate signals at $m/z$ 46. Prompt particles have relatively higher organic signals at $m/z$ 44, sulfate signals at $m/z$ 48 and 64, and nitrate signals at $m/z$ 30. The chemical ion signals of prompt and delayed particles are different on average, which may indicate the mechanism for producing these delayed particles. Differences in the spectra of

prompt and delayed particles have not been previously reported, and it is possible that chemical differences may have caused these particles to have different properties.

To better interpret the difference in spectra of prompt versus delayed particles from SENEX, LSSP data with varying detector sensitivities were collected and analyzed for dry, poly-dispersed, laboratory particles composed of internally-mixed organic

dicarboxylic and carbonyl acids, ammonium organic acid salts, ammonium sulfate, and ammonium nitrate. A total of 1058 light-scattering events were recorded and analyzed using the same criteria as the SENEX particles except for a lower limit on the number of ions detected in the mass spectra. Of these, about 12% were below the noise level for actual particle light-scattering events. Of the particles above the light-scattering noise, 47% were prompt, 13% were delayed, and 37% were null.

The laboratory mass spectra for the mixed composition particles with comparable detector sensitivity to SENEX are shown in Figure 4b and had similar patterns to the SENEX data, with more prominent peaks at $m/z$ 44, 48, and 64 in the prompt particle spectra and more prominent peaks at $m/z$ 43, 45, 46, 81, 98 and organic peaks with $m/z \geq 60$ in the delayed particle spectra. For these mixed composition particles, the $m/z$ 30 peak was slightly more prominent in the prompt particles and $m/z$ 46 was





more prominent in the delayed ones. At higher detector sensitivities (not shown), the main sulfate peaks at $m/z$ 48, 64, 80, 81, and 98 and the main nitrate peaks at $m/z$ 30 and 46 appeared more prominent in the delayed spectra. For the sulfate-containing calibration particles shown in Figure 1, the sulfate pattern of high $m/z$ 81, and 98 in the delayed particles and high $m/z$ 48 and 64 in the prompt particles was consistently observed (not shown). The peaks associated with ammonium (and water) did not

appear to show any systematic differences between prompt and delayed particles. For pure ammonium nitrate calibration particles, none of the LSSP data were classified as delayed (see Figure 1). The laboratory data confirm that nominally identical particles can produce differences in the prompt versus delayed spectra. Thus, a difference in aerosol chemical composition is unlikely to be the only reason for the chemical ion signal difference in the SENEX data.

The consistent differences in the mass spectra between prompt and delayed particles from both ambient and nominally-identical chemical composition laboratory particles can be due to different processes for prompt and delayed particles. Prompt particles probably vaporize at higher temperature surfaces resulting in more thermal decomposition. The gas molecules vaporized from prompt particles have a higher temperature and internal energy, and thus may fragment more in the electron beam. Moreover, collisions of vaporized gas with the vaporizer during prompt events can also contribute to additional thermal

decomposition. The location where the delayed particles impact and vaporize is likely further away from the vaporizer center which could result in fewer wall collisions of the vaporized species and consequently less potential for additional thermal decomposition. The data showed higher detected signals at $m/z > 60$ for larger molecular weight organic compounds in delayed particles compared to prompt particles, probably resulting from less decomposition and fragmentation of the species from delayed particles. The signal at $m/z$ 44 is more prominent in prompt particles compared to delayed particles. This is consistent

with more thermal decomposition and fragmentation to $CO_2^+$ with prompt particles. Hence, the emerging picture is that the observed signal difference in the prompt and delayed particles for SENEX was likely due to the delayed particles vaporizing from a surface at slightly lower temperature than the vaporizer or due to fewer collisions between vaporized gas and hot surfaces.

For the mixed particles generated specifically to compare here with the SENEX data, the mass spectral peak widths were analyzed to check for slower vaporization (Drewnick et al., 2015). A proxy for peak width in the mass spectra (peak area divided by height) was not statistically significant. Because the spectra were saved every 32 μs and vaporization event lengths in the AMS are on the order of 30-60 μs and constant for pure ammonium sulfate particles at temperatures between 400 and 800 °C, the data collected in this brief lab study could not be used to validate the possibility of differing vaporization

temperatures.

### 3.1.4 Aerosol mass difference between prompt and delayed particles

For a given size determined by $d_{va\text{-}LS}$, the ion signals from prompt particles were slightly larger than those from delayed particles. Single particle mass obtained from mass spectrometer chemical ion signals was compared to that derived from





aerosol size $d_{\text{va-LS}}$ of prompt (red) and delayed (blue) particles of the flight on July 6, 2013 in Figure 5. The effective particle density is estimated to be 1.55 g cm$^{-3}$ according to the average ammonium sulfate to organic mass ratio of 0.96, ammonium sulfate density of 1.77 g cm$^{-3}$ and typical organic density of 1.4 g cm$^{-3}$. Single particle masses derived from the mass spectrometer chemical ion signals were well correlated with that derived from measured $d_{\text{va-LS}}$ for prompt particles with a correlation coefficient of 0.87 and a slope of 1.04 with an intercept defined to be 0 for this flight. This good correlation between the mass of the particle from the mass spectrum and the size of the particle measured by the timing of the scattered light signal indicates that LSSP mode is reasonably quantitative on a single particle basis. The slope depends on the accuracy of the effective particle density, $d_{\text{va-LS}}$ measurements, and relative ionization efficiency (RIE). For the same size $d_{\text{va-LS}}$, the mean ratio of individual particle mass from chemical ion signals from the delayed particles (blue points) was 0.78 of the mass from prompt particles (red points). This ratio does not depend on the accuracy of the effective particle density, $d_{\text{va-LS}}$, or RIE and was similar in all flights sampling different air masses during this field campaign. While the timing and spectra suggest that these delayed particles vaporized outside of the vaporizer at a lower temperature, most of the mass was detected.

In contrast to the SENEX study, the individual particle mass signal for delayed particles during the Mexico City study was less than half of that for the prompt particles (Cross et al, 2009). Although the two studies used different definitions for prompt and delayed particles, changing this definition does not alter the measured average chemical ion signals. Also, both studies used the particle size from light scattering information to calculate the volume. Here the particle size was the vacuum aerodynamic diameter from the maximum time of the scattered light signal ($d_{\text{va-LS}}$) whereas for the Cross et al. (2009) work the volume was determined from $d_{\text{va-LS}}$ and the optical diameter ($d_{\text{o}}$) from the intensity of the scattered light signal. However, these slightly different particle size-based volume calculations between this study and Cross et al. (2009) will not affect the relative signal intensity between prompt and delayed particles. The vaporizers in these two campaigns are designed to be identical. The reasons for much lower delayed particle mass compared to prompt ones remain unclear, but differences in the ambient aerosols measured may contribute to the different delayed particle mass. The Mexico City study was conducted on the ground near the metropolitan area where ~10% of PM2.5 mass was black carbon (Retama et al., 2015). As the null fraction in Cross et al. (2009) was not higher than in this study, the much lower mass from delayed particles chemical signals observed by Cross et al. (2009) could be due to particles containing more refractory material during the Mexico City study than during SENEX.

## 3.2 Observations using the AMS LSSP mode

### 3.2.1 Comparing mass fractions between MS and LSSP modes

One method of evaluating data from the LSSP mode is to compare the average mass fractions of the main species from the LSSP spectra to that measured in the ensemble MS mode. Mass fractions of non-refractory aerosol organic, sulfate, ammonium, nitrate and chloride measured by MS mode versus LSSP mode for all the SENEX flights analyzed (June 26 to July 8) are



shown in Figure 6. Data in the MS mode adjacent to the LSSP mode were interpolated and compared to the LSSP mode data. Considering that the standard deviation of sulfate mass fraction of standard $(NH_4)_2SO_4$ aerosols measured by LSSP mode is about 10% and the MS and LSSP mode were not sampling at the same time during the aircraft measurements, the chemical mass fractions of the various species are well-correlated between the two modes. When averaged, the 30-second single particle

data sampled in lower troposphere over the continental US could be representative of the ensemble chemical composition mass fractions. The reasons why the ammonium and nitrate mass fractions were slightly higher in the single particle data are not clear. Since nitrate mass fractions were low for the ensemble data, the uncertainties are relatively larger. Potential explanations for the higher ammonium in the LSSP data may be related to the difference in detected particle size range between LSSP and MS mode or the low sensitivity which could artificially increase the signals for ammonium (Hings et al., 2007) and may be

especially important for the single particle data. Overall, the AMS single particle mass fractions are generally comparable to the ensemble measurements, which was also reported previously for Mexico City (Cross et al., 2009).

### 3.2.2 Size-resolved mass distribution comparison

There are three slightly independent ways to generate mass distributions from an AMS instrument with a light-scattering module: (1) traditional PToF mode distributions, with non-refractory ensemble composition as a function of $d_{va}$, (2) particle

counts from the LSSP mode laser as a function of $d_{va-LS}$, converted into a mass distribution by assuming an effective particle density of 1.55 g cm$^{-3}$, and (3) LSSP mode mass from the single particle chemical ion signals as a function of $d_{va-LS}$. For this work, the PToF distributions were normalized to the mass loadings from MS mode that were derived from the complete fragmentation patterns in the mass spectra (Allan et al., 2004) and the composition-derived collection efficiency (Middlebrook et al., 2012). The count-based and mass-based LSSP mode mass distributions were scaled to the laser counts from the adjacent

MS mode, which did not record $d_{va-LS}$ and mass spectrum information, to account for the significant time spent in saving the single particle information in LSSP mode. This potentially counted particles that were above the light-scattering threshold set in the data acquisition software yet below the optical detection limit set for analyzing the LSSP data. These particles accounted for 4% of the total LS-triggered events on average and the mass percentage from these particles would be much smaller considering that they are the small particles below optical detection limit. Details on the LSSP mass distribution calculations

are in the SI. Figure 7a shows these three mass distributions from ambient air mass below 3000 m in altitude during the flight on July 6, 2013. A calculated mass distribution from the UHSAS instrument (solid black curve) is also depicted in Figure 7a, where the UHSAS number distribution is multiplied by the AMS lens transmission efficiency (Liu et al., 2007) and converted to mass as a function of $d_{va}$ by assuming an effective particle density of 1.55 g cm$^{-3}$.

The four curves in Figure 7a demonstrate various properties of the LS-ToF-AMS system. The LSSP mode count-based mass distribution (red curve) compared to that from the UHSAS instrument (black curve) indicates that the AMS laser system here accounted for the mass from most aerosol particles with $d_{va} > 440$ nm and essentially none of the mass from particles smaller than $d_{va} < 280$ nm. Although particles as small as $d_{va} \sim 170$ nm could trigger the LSSP data acquisition (Figure 2), a very small



number of these particles were detected and they did not contribute significantly to the LSSP-mode mass distributions until they were larger than $d_{va}$ ~280 nm. The laser and optics used in the LS-AMS are clearly not optimized to detect small ambient particles. Because the two (black and red) distributions are nearly identical for $d_{va} > 600$ nm, the standard AMS lens transmission function (Liu et al., 2007) appears to be valid for the upper size range. The particle mass from chemical ion

signals (grey) is lower than the laser counted particle mass (red) because not all of the particles optically detected produced detectible chemical signals. The ratio of these two is a mass-based measure of the collection efficiency and is discussed further in Section 3.3.2. The uncertainty by including the particles with light scattering signals below the detection limit is eliminated when calculating this mass ratio. The AMS PToF mass distribution (dashed curve) has a factor of four more mass than all of the other distributions from particles at the large size end (> 700 nm) (discussed further below).

**3.3 General AMS operation and data analysis suggestions from AMS LS module results**

**3.3.1 Measurements of the traditional AMS vacuum aerodynamic diameter $d_{va}$**

Delayed particles appear at larger sizes of $d_{va}$ in PToF mode, creating a bias towards the larger size end of the mass distribution. The mass from the traditional PToF distribution (Figure 7a, dashed curve) is higher than all the other curves for particles larger than 700 nm even considering the overall uncertainty. This is also demonstrated by plotting the mass distributions using only

the LSSP data from particles with detectable chemical ion signals as a function of the two different particle sizes, $d_{va-LS}$ (Figure 7b) and $d_{va}$ (Figure 7c). Compared to the distributions plotted as a function of $d_{va-LS}$, mass distributions as a function of $d_{va}$ have less mass for the intermediate sizes ($d_{va}$ ~ 350 to 500 nm) and more mass at the larger sizes ($d_{va} > 700$ nm). The broadening of the traditional AMS mass distribution to larger sizes due to delayed particles was also observed during the Mexico City and Bakersfield studies (Cross et al., 2009; Liu et al., 2013). Thus, $d_{va-LS}$ instead of $d_{va}$ is a more reliable parameter to represent

particle size in the AMS. On average, the relative composition as a function of size for the different distributions does not appear substantially different. The bias toward increased mass at the larger sizes from delayed particles needs to be considered when interpreting standard AMS PToF mass distributions data.

**3.3.2 Accuracy of the AMS parameterized collection efficiency**

The light scattering module can be used to measure in situ AMS collection efficiency (CE), defined here as the ratio of number

(or mass) of particles with both detectable chemical (prompt and delayed) and optical signals to the total number (or mass) of all particles with detectable optical signals. The number-based CE from the LSSP data does not include counts from saved data where the light scattering signal was close to the noise level and is defined here as the (prompt + delayed) particle counts divided by (prompt + delayed + null) counts. The mass-based CE is defined in Section 3.2.2. Both number- and mass-based LSSP mode CE values are calculated for the 30-second intervals average of LSSP data every five minutes. As mentioned

before, because significant time was needed to save single particle optical and chemical information, LSSP mode did not record all particles sampled. There is an assumption for the measured CE that the undetected mass is the same as the detected



mass, which is likely true due to random detection. It is also assumed that particles detected optically are representative of all particles sampled by the AMS and have the same chemical composition as the particles that are too small to be detected by LSSP mode. In air masses where newly formed and growing (Aitken mode) particles are present, this assumption is not necessarily valid.

The CE measurements based on LSSP particle number or mass varied from about 0.2 – 0.9 for this study. The flight that had the largest range of CE from about 0.4 to 0.9 based on the measurements was on July 6, 2013 and is shown in Figure 8. Besides the CE measurements, the CE parameterization based on ensemble measured chemical composition from MS mode (Middlebrook et al., 2012), which is commonly used in AMS data analysis software, is also plotted in Figure 8. In contrast to

CE measurements from LSSP mode data, the CE parameterization covers the entire particle size range detected in PToF mode (Figure 7a). The three values for CE shown in Figure 8 were correlated and generally agreed considering experimental uncertainties for this flight. The statistical variation of measured CE values ($\sigma_{CE}$) shown in Figure 8a is estimated to be less than ±0.08 for the 5 min average data based on variations of a binomial distribution as:

$$\sigma_{CE} = \frac{\sqrt{np(1-p)}}{n} \tag{4}$$

where n is number of particles with optical signals above detection limit, and p is the probability of optically detected particles that can be chemically detected, varying from 0.3 to 0.9. The statistical variation is shown as error bars in measured number fractions based-CE. The large CE changes were not due to statistical variation. CE values above 0.5 were primarily due to the presence of acidic-sulfate particles during this flight and were determined by both composition-dependent CE parameterization from the ensemble MS mode data and in situ LSSP mode measurements. The default CE parameterization value of 0.5 may

be too high for some cases. The CE values for other flights (in supplementary information Figure S2) also indicated similar over estimation of CE default value of 0.5 in some cases. However, this may be an artifact of operating the LS-AMS during SENEX with less than optimal sensitivity. In general, the good correlation indicates that aerosol chemical composition and relative humidity dependent CE parameterization (Middlebrook et al., 2012) can accurately capture the general variability of CE at least for the cases when significant variation in CE is due to change of aerosol acidity.

While the aircraft data reported here show a wide range of CE due to air mass variations, such variability in the LSSP mode CE has not been reported previously. The Bakersfield study described a discrepancy between the average number- and mass-based CEs, where the number based value was ~0.5 and the mass-based value from ensemble measurements was 0.8 (Liu et al., 2013). The authors proposed that a mismatch of vaporization and data acquisition time scales reduced the detected chemical

ion signals from single particles compared to the ensemble measurements; yet this discrepancy was not resolved. The in situ CE from LSSP mode measurements were also determined and compared with AMS ensemble and independent measurements for the Mexico City study (Cross et al., 2009). The number- and mass-based CE was on average ~0.5 for the 75-h sampling period of LSSP data and showed some size dependence with the smallest particles having a high CE (low null fraction and



higher prompt fraction) than the larger particles. Variations in ensemble CE were not reported but could have been possible due to the diurnal variability in the ambient measurements, where in the morning there were small particles composed of predominantly hydrocarbon-like organic aerosol (HOA) which appeared to have a higher CE. Hence, LSSP data could also show that mixing state plays a role in the measured CE.

**4 Conclusions**

This paper reports the first airborne single particle measurements from a light scattering module coupled to an AMS instrument and uses these measurements to investigate the AMS in situ CE values, particle bouncing physics, and impacts and uncertainties in the AMS size resolved mass distribution. Results from these unique airborne single particle measurements showed that the LS module has the capability to measure in situ AMS collection efficiency despite many assumptions and

uncertainties in this method. The comparison of measured and calculated CE demonstrated that the aerosol chemical composition and humidity dependent CE parameterization that is commonly used to reduce AMS data is reasonable for the cases when aerosol acidity plays a role in ambient CE variations. Single particle data derived average aerosol chemical composition fractions were generally well correlated with the ensemble data. This shows that the airborne AMS with an LS module is a potentially useful tool for measuring the ambient aerosol chemical composition. However, the effectiveness of the

LSSP mode for ambient measurements is limited to particles that are large enough to scatter light and generate sufficient chemical ion signals to be detected.

The different chemical ion signals in prompt and delayed particles indicated that delayed particles are likely those that bounce off the vaporizer surface and vaporize after impacting another surface at a lower temperature. The field measurements and

laboratory data both demonstrate that the mass spectra from delayed particles have less fragmentation and slightly lower ion signals than spectra from prompt particles. The individual particle mass appears to be consistent with most of the material vaporizing for both prompt and delayed ambient particles from SENEX. Delayed particles appear as larger particles in the traditional AMS PToF mass distribution. Caution should therefore be used when interpreting AMS PToF mode particle size data.

**Acknowledgements**

We thank Joel Kimmel, Tim Onasch, Steven Brown and Annie Davis for their help to tune AMS instrument, troubleshoot light scattering module, align the laser, and plot figure 1S, respectively.



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



**Table 1: Summary of AMS light scattering results.**

| | This work | Cross et al. (2009) | Liu et al. (2013) | Lee et al. (2015) |
|---|---|---|---|---|
| Sampling location | Over the SE U.S. | Mexico City | Bakersfield, CA | Toronto |
| Mass Spectrometer | C-ToF | C-ToF | HR-ToF | HR-ToF (SP laser off) |
| Chopper-to-Vaporizer Length (cm) | 39.5 | 29.0 | NR | 39.5 |
| LS Particle events analyzed | 33,861* | 12,853 | 271,641 | 84,218 |
| $d_{va}$ for 50% LS detection efficiency (nm) | 360 | 370 | 430 | 340 |
| Min. # Ions for MS Detection | 38** | NR | 6 | 6 |
| Definition of "delayed" | $t > (t_{est}+t_{offset})$ +3×Gauss. width | $t > (t_{est}+t_{offset})$ | $t > 1.20×$ $(t_{est}+t_{offset})$ | $t > (t_{est}+t_{offset})$ +3×Gauss. width |
| Offset time (ms) | 0.35 | 0.2 | NR | ~ 0.42 |
| Prompt Fraction | 27% | 23% | 46% | 33.6% |
| Delayed Fraction | 15% | 26% | 6% | 0.4% |
| Null Fraction | 58% | 51% | 48% | 63.2% |

NR = Not Reported

*Total particle number analyzed of research flights June 26-July 8.

**Average minimum ions for MS detection of research flights June 26-July 8.

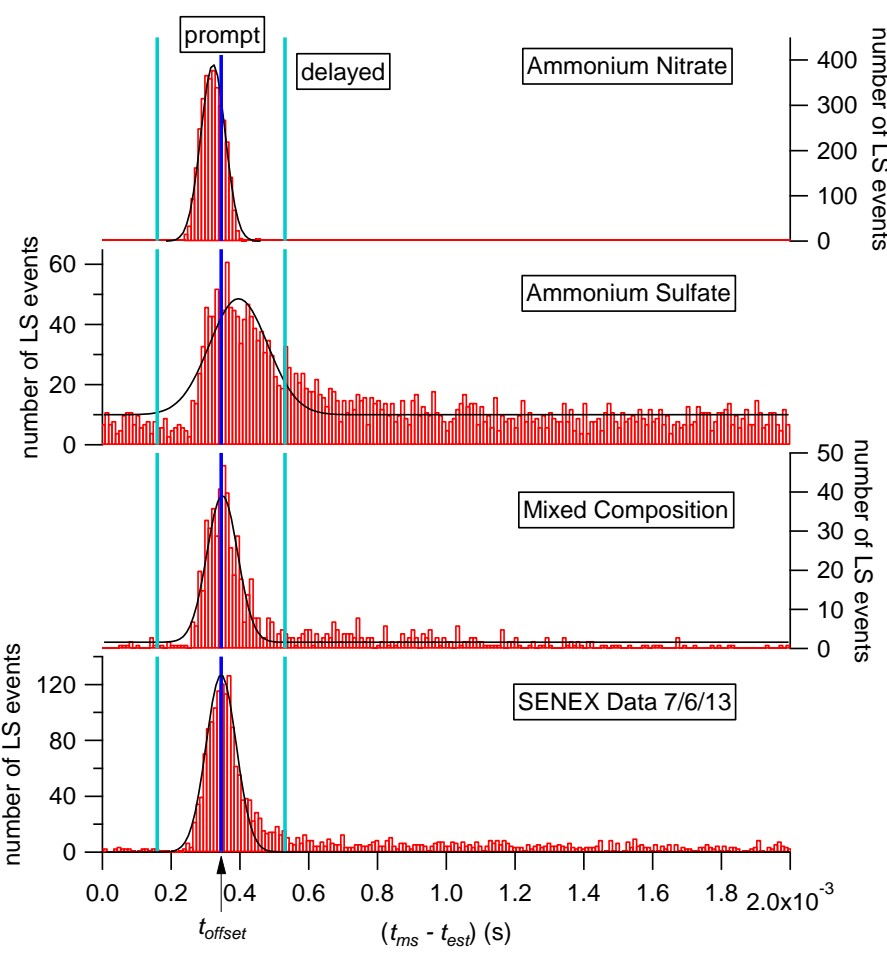

**Figure 1. Histograms (red) of the maximum mass spectra signal time ($t_{ms}$) minus the estimated particle arrival time ($t_{est}$) at the vaporizer for particles composed of pure ammonium nitrate, pure ammonium sulfate, mixed composition, and the flight data on 07/06/13. Gaussian fits of the histogram data are in black. The offset time ($t_{offset}$) from the fit for the SENEX data (the dark blue vertical line) is similar to the offset times for the mixed composition laboratory particles. The distances between dark blue line and cyan lines are three times the width of the SENEX Gaussian fit. Prompt particles are defined as those with times between the cyan lines and delayed particles are detected at later times on the right hand side of the distribution.**



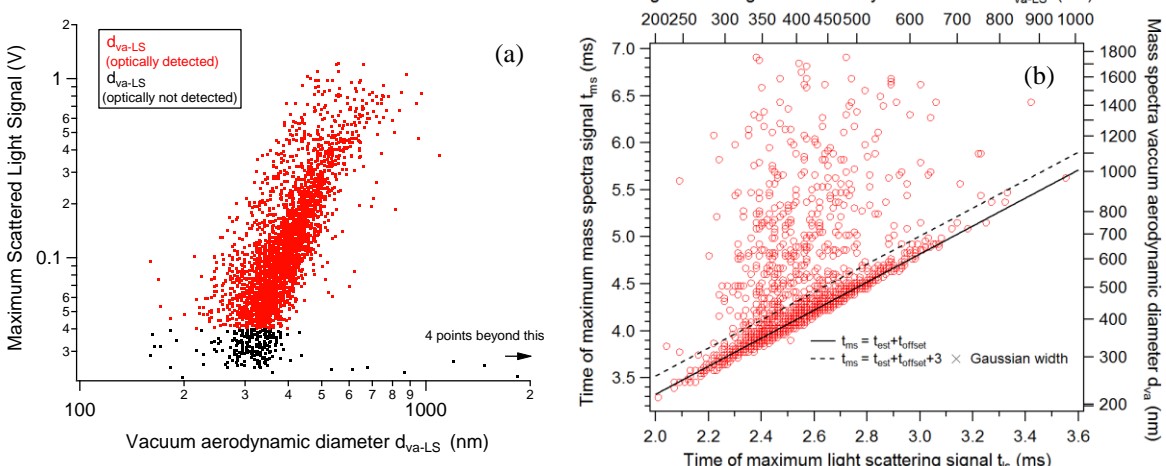

**Figure 2. (a)** The maximum scattered light signal from all LS-triggered events for the flight on July 6, 2013, versus vacuum aerodynamic diameter from the timing of the maximum scattered light signal, $d_{va\text{-}LS}$. The limit of optical detection was defined for this paper as having a maximum scattered light signal above 0.04 (red points). Particles with signals below this are defined as "optically not detected" (black points). **(b)** Maximum mass spectra appearance time ($t_{ms}$) versus maximum light scattering signal appearance time ($t_{ls}$) and their corresponding vacuum aerodynamic diameter $d_{va}$ and $d_{va\text{-}LS}$ of all chemically detectable particles (prompt and delayed) from the flight on July 6, 2013. In the legend, $t_{est}$ is from Equation 2 and $t_{offset}$ is the intercept of the solid line.

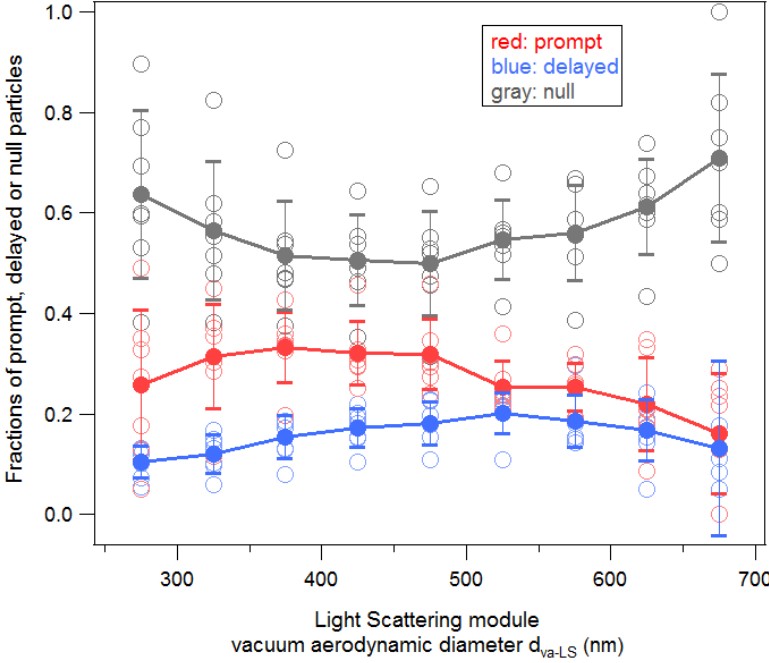

**Figure 3.** The number fractions of prompt (red), delayed (blue) and null (grey) particles as a function of particle vacuum aerodynamic diameter from the light scattering module, $d_{va\text{-}LS}$, for flights from June 26 to July 8 during SENEX. Circles represent the average data of each flight. The solid dots and lines represent the average data of these flights. Error bars represent one standard deviation from the average. The flight on July 6 had the lowest null fractions while the flight on July 3 had the highest null fractions.





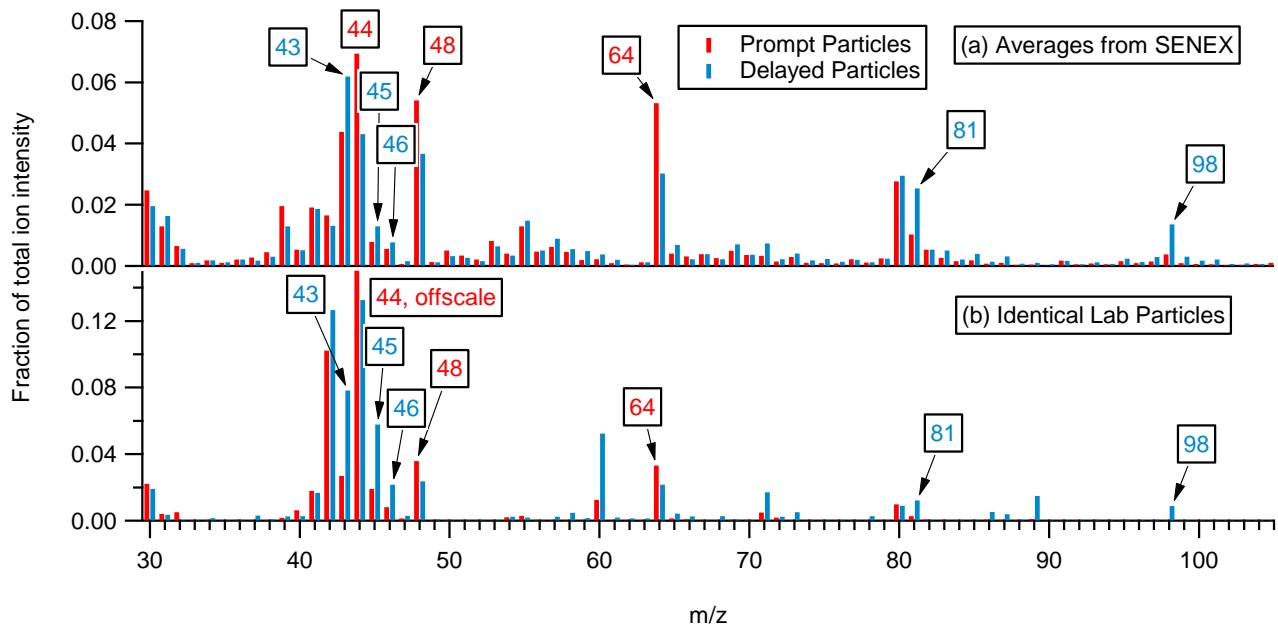

**Figure 4. (a) Average mass spectrum (in total ion intensity fraction) of prompt (red) and delayed (blue) particles for the SENEX flights analyzed. The average prompt and delayed spectrum are shifted -0.2 and +0.2 units, respectively, in m/z for display clarity.**
5 **(b) Average mass spectrum (in total ion intensity fraction) of prompt (red) and delayed (blue) laboratory particles composed of internally-mixed organic dicarboxylic and carbonyl acids, ammonium organic acid salts, ammonium sulfate, and ammonium nitrate.**





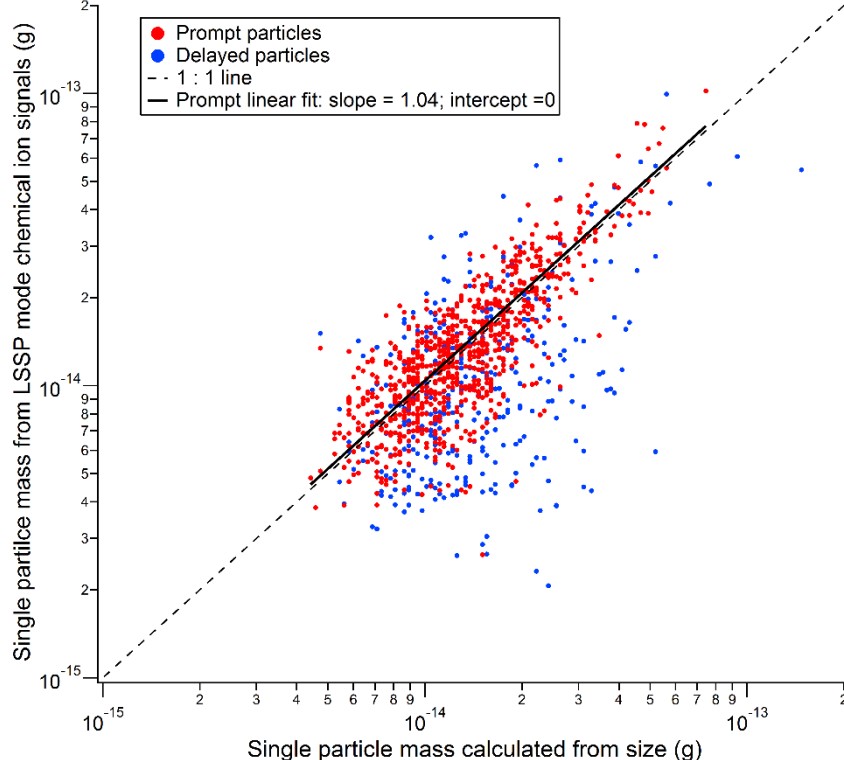

**Figure 5.** Scatter plot of the single particle mass from the LSSP mode chemical ion signals versus that derived from particle size $d_{va\text{-}LS}$ by assuming a density of 1.55 g cm⁻³ for prompt (red) and delayed (blue) particles during the research flight on July 6, 2013. For prompt particles, the slope of the linear correlation is 1.04 with an intercept set to be 0.




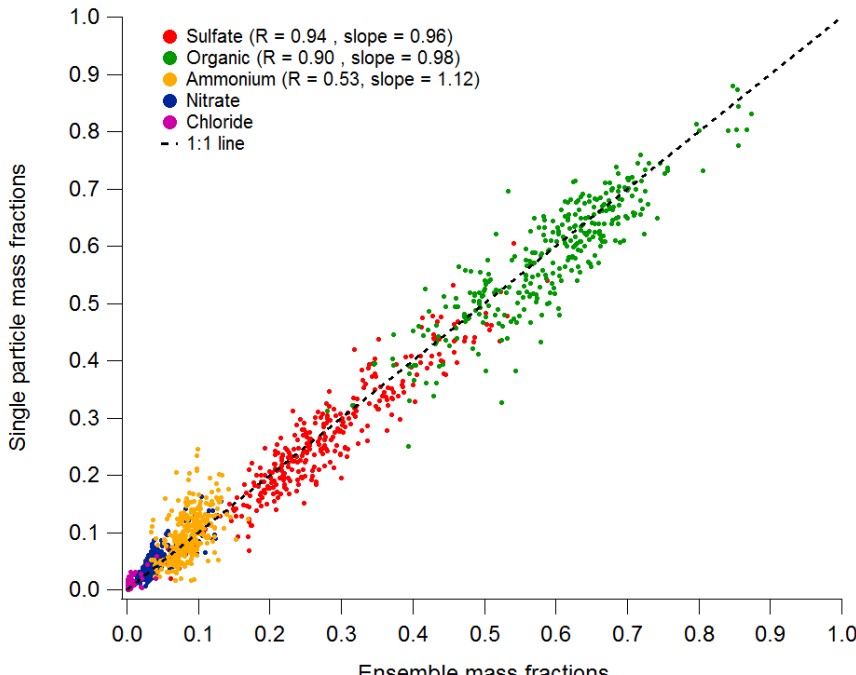

**Figure 6. Organic (green), sulfate (red), ammonium (yellow), nitrate (blue), and chloride (purple) mass fractions from single particle measurements versus ensemble measurements from SENEX.**





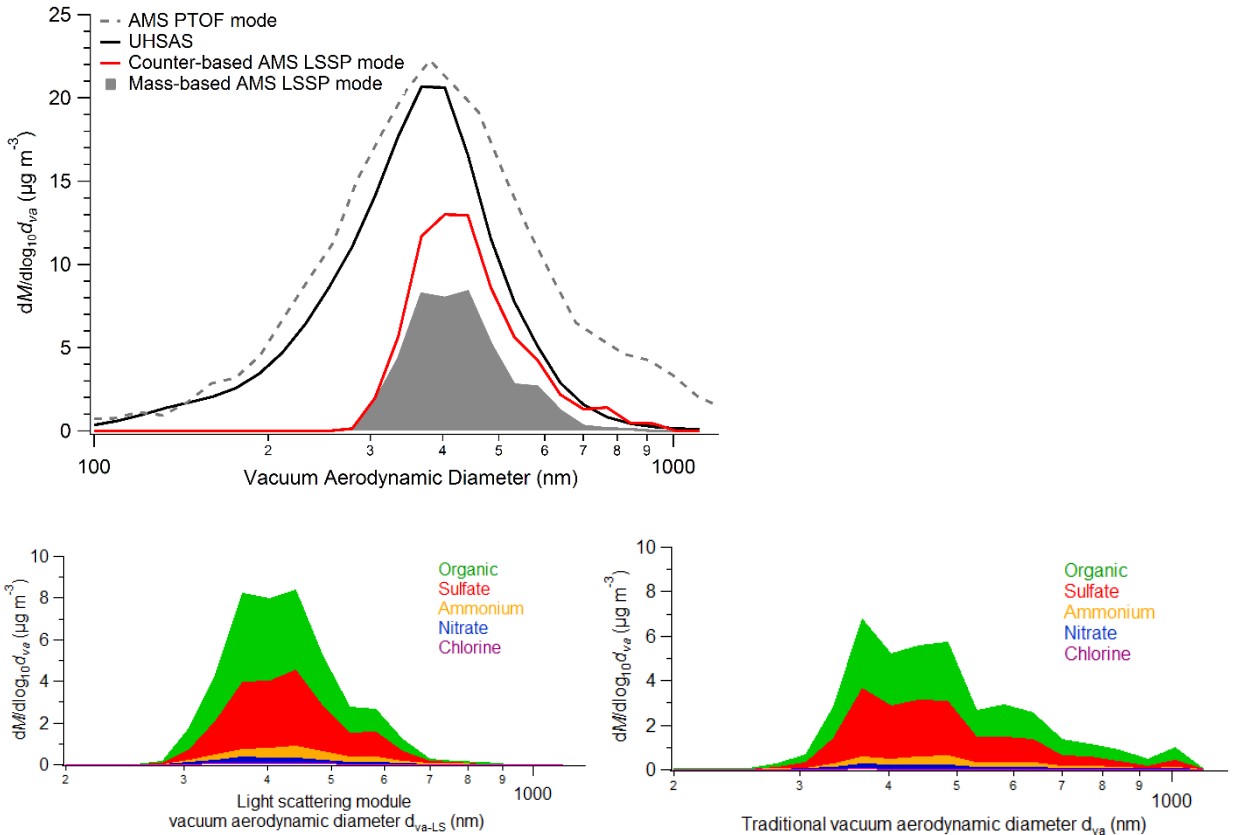

**Figure 7.** Size dependent mass distributions as a function of vacuum aerodynamic diameter from low altitudes during the flight on July 6, 2013. (a) The mass is determined from the traditional AMS PToF mode vs. $d_{va}$ (dashed), from the UHSAS instrument vs. $d_{va}$ (black) after correcting the particle number distribution for the AMS lens transmission and using an estimated density of 1.55 g cm$^{-3}$, from the AMS LSSP internal particle counts vs. $d_{va\text{-}LS}$ (red) also using an estimated density of 1.55 g cm$^{-3}$, and from the AMS LSSP mass spectra vs. $d_{va\text{-}LS}$ (grey area). Parts (b) and (c) highlight the differences between plotting the mass distributions for each species measured using AMS LSSP mass spectra as a function of either (b) the light scattering module derived vacuum aerodynamic diameter ($d_{va\text{-}LS}$) or (c) the traditional vacuum aerodynamic diameter ($d_{va}$).



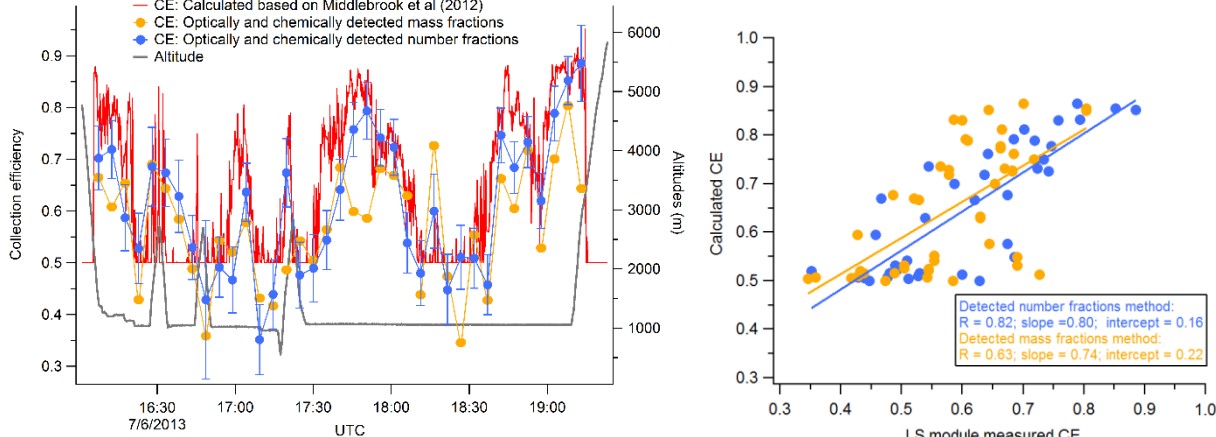

**Figure 8. (a) Time series plot of AMS collection efficiency calculated based on aerosol chemical composition and relative humidity method described in Middlebrook et al. (2012) (red) and measured by AMS LSSP mode based on number (blue) or mass (yellow) ratio of optically and chemically detected particles to total optically detected particles of July 6, 2013 flight. (b) Scatter plot of the parameterized, composition-dependent CE vs. the LS module measured CE based on optically and chemically detected particle number fractions (blue, correlation coefficient R = 0.82) and mass fractions (yellow, correlation coefficient R = 0.64) for the July 6, 2013 flight.**