# Peer review of "Single particle measurements of bouncing particles and in-situ collection efficiency from an airborne aerosol mass spectrometer (AMS) with light scattering detection"

_Atmospheric Measurement Techniques, 2016_

## Referee Comment (RC1) · Anonymous Referee #2 · 9 Mar 2017

In their manuscript "Single particle measurements of bouncing particles and in-situ collection efficiency from an airborne aerosol mass spectrometer (AMS) with light scattering detection", Liao and coworkers use results from airborne measurements with an AMS with implemented LS detection to investigate effects associated with vaporization of individual particles. From these measurements the authors draw conclusions about particle bounce and vaporization in the AMS and how this affects collection efficiency correction factors or measurements of particle size distributions. With this approach and focus the manuscript is well suited for Atmospheric Measurement Techniques and I recommend publication of the manuscript AMT after the following issues were resolved.

Generally, the manuscript is well written. However, at places the authors should select wording that is scientifically more correct (see examples below). Furthermore, some of the conclusions regarding the processes within the ionizer chamber after bouncing of particles off the vaporizer are rather speculative and should be supported by additional evidence or rephrased. Most of the comments below, however, are more associated with clarity or correctness of presentation than with the content of the work.

Detailed comments:

P1L15-16: Not only particles larger than ∼250 nm scatter light, but also smaller ones. Rephrase.

P1L24: Delayed particles are detected later than "expected". I suggest rewording to something like "delayed particles are detected later than appropriate for their size" or ". . . later than expected from their measured velocity".

P1L26-27: "higher null fractions and corresponding lower CE for this study may have been related to the lower sensitivity of the AMS during SENEX". While the lower sensitivity of the AMS probably causes some of the single particles to produce insufficient signal for single particle identification, all the aerosol signal is probably measured in MS mode where mass spectra are averaged over large time intervals. While the missing of single particle counts using the LSSP mode results in smaller CE factors, the real CE factor should probably higher, thus this should result in a CE factor biased low. Is this correct? How does the "lower sensitivity" affect LSSP and MS mode measurements? How does this potentially affect m/z signals with low relative intensity?

P2L5: aerosol inhomogeneity is blamed on "short atmospheric lifetimes, compared to greenhouse gases". While it is correct that the lifetime of aerosol particles is short compared to that of greenhouse gases, this is irrelevant here. In this context it is more important that aerosol lifetimes are short compared to timescales of mixing processes.

P2L17: Here and in several other places in the manuscript CE is described as the

fraction of mass or number of particles detected by AMS. Traditionally CE is defined as the fraction of mass of particles detected by the AMS. As shown in Figure 3, the bouncing fraction of particles is a function of particle size. Therefore using only the fraction of number of particles that are detected will likely introduce a bias.

P2L24: "bouncing on the vaporizer" should read "bouncing off the vaporizer".

P3L16-18: The sentence starting with "Many airborne ..." contains information that is not relevant here and should be removed.

P3L23: Section 2 ("Experimental") already includes a large number of results. Either the section should be renamed or the results moved into the results section.

P4L11 (and many other places): the unit "bits-ns" should read "bits*ns" since it is the product of the hight (bits) and the width (ns) of the peaks in the ToF mass spectra.

P4L25: Between "as" and "instruments" a word is missing, either "of" or "for".

P4L28: The LS-AMS is not capable of measuring "single particle mass" but either "single particle composition" or "non-refractory single particle mass".

P5L2-3: What does "LS-triggered particle scattered light signal" mean? Rephrase.

P5L12: "the corresponding scattered light signal" should probably be "the corresponding light scattering signal".

P5L13-14: Is there any background subtraction for the mass spectra applied for single particle information?

P5L16-17: "The cases ... are called coincident particles." This does not make sense. Rephrase.

P5L27: Replace "the third measurement of ..." with "the third type of measurement of ..."

P6L5: "... an additional brief amount of time ...". Can you provide typical numbers for

this?

P6L7: What do you mean with "different signal paths"?

P6L8: "the slowest of these processes and largest uncertainty ...". Can you provide an estimate on how long this is and how large uncertainty is?

P6L11-30: In these paragraphs there are several statements that should be phrased more correctly, e.g. "The second method ... is the vacuum aerodynamic diameter ...", "... calibrated with known particles ...", "... where the calculated particle velocity uses ...", or "... that the laser not only counts ... but also counts ..." (the laser does not count, it can be used to count).

P6L22: How can the maximum of the Gaussian distribution be before Toffset, which is defined as the mean of the distribution?

P6L26-28: Why is dva-LS calculated from the laboratory calibration of dva and not calibrated individually?

P7L1: What do you mean with "compared to MS mode"? Reword.

P7L11: The IE for nitrate is claimed to be 7E-7 ions/g. This unit is wrong – or the IE of this instrument is orders of magnitude lower compared to other AMSs.

P8L4 (and many other locations in the text): A clear differentiation between "signal" and "signal intensity" (i.e. the magnitude of a signal) should be made throughout the text. Similarly, wording like "maximum light scattering intensity" (P8L10) should be replaced by something like "maximum light scattering signal intensity" since not the intensity of the scattered light but of the signal from the measurement of the scattered light is meant.

P8L12: "criteria" should be "criterium"

P8L14-15: The light scattering intensities do not vary over a significant range because the laser beam is broad, but because the beam intensity has a shape (likely Gaussian

shape) across its cross section.

P8L24: "axis" should be "axes". What is a "mass spectrum arrival time"?

P8L31-P9L6: The "delayed particles" were explained by particles that bounce off the vaporizer and vaporize at other, colder surfaces. To explain the time delays observed in the measurements the particles have to travel with very low velocities after bouncing off the vaporizer. This explanation is very speculative. Do the authors have any evidence for this? I could imagine if the particles bounce of the vaporizer in such a way that they have to loose most of their energy, they must be strongly deformed during the bounce process which I could imagine results in good contact with the vaporizer. Therefore I would expect that they stick to the vaporizer like ammonium nitrate and other liquid particles do. What do the authors suggest as the process that causes a loss of the vast majority of kinetic energy of the particles and at the same time avoid sticking of the particles to the vaporizer?

P9L8-15: This paragraph is very speculative. Do the authors have any evidence for this behavior of the particles?

P10L1-2: "The single particle mass is near the chemical detection limit . . .". For which size of particles is the mass near the LOD?

P10L1-12: According to this discussion appearance of a particle as a "null" particle is not a proof or hint of particle bounce but for a large fraction of the particles (most of the mass measured with an AMS is typically in the 100-400 nm range) could as well simply be due to low particle signal. This should be discussed when discussing bounce.

P10L26: Add "for" between "as" and "the SENEX".

P10L30-P11L8: Without more information on how detector sensitivity affects data acquisition and without any interpretation or discussion of the differences in the mass spectra observed at different detector sensitivities, this paragraph does not provide very helpful information.

P11L15: "The location where the delayed particles impact and vaporize is likely further away from the vaporizer center . . .". Do the authors mean that delayed particles impact near the edge of the vaporizer? In this case a shift towards more delayed particles should be observed when scanning the particle beam across the vaporizer. Furthermore, not every location further away of the vaporizer is colder than the vaporizer. The hottest location in the ion source is the filament. Be more clear here.

P11L10-23: Here the different chemical signal (e.g. less fragmentation) of the delayed particles is explained by vaporization of these particles off cooler surfaces. Where do the authors assume that these surfaces are? In the AMS ionizer the hottest point is the filament, not the vaporizer. Therefore some of the other surfaces might be even hotter than the vaporizer. If particles vaporize from other colder surfaces one would expect longer vaporization times for the individual particles. The authors report that this is not observed and explain this by the large range of vaporization temperatures where the vaporization time is not extended (400-800 °C). Which vaporization temperature would be expected to obtain the observed changes in the fragmentation pattern? Is this consistent with this range of vaporization temperatures for quick vaporization?

P11L25: "mass spectral peak widths" sounds like the width of the peaks in the mass spectra while the duration of particle vaporization is meant. Be clearer.

P12L8-12: The AMS ionization efficiency strongly depends on the exact location of the vaporizer within the ion source. If the vaporizer (i.e. the location from where the evolved vapor originates) is moved, IE drops strongly. Therefore I would expect strongly reduced ion signals (i.e. measured particle mass) from particles vaporizing at a completely different location within the ion source. This would not be in agreement with the observations made in this work and the suggested processes occurring in the ion source. Furthermore, in section 3.1.4 I miss a discussion on the reasons for the missing mass in the delayed particles. Is there a different composition for the delayed and prompt particles? In addition: how certain is it that the null particles are also of the same composition as the measured ones? Is the loss in particle mass different for

different instruments or what influence does the large difference in the criterion for MS detection (see Table 1) have?

P12L14-27: With about 10% black carbon content in the fine aerosol observed in Mexico City it is hard to believe that the much lower measured delayed particle mass (less than half of the prompt ones) observed in this study is caused by larger black carbon content during that measurement. What was the BC content during this study?

P12L28: The title of section 3.2 is not very well chosen. Most of the findings presented here are not observations using the AMS LSSP mode but are still characterization of the LSSP and PTOF measurements.

P13L16: Since here general ways to generate mass distributions from AMS instruments with LS module are described it is not appropriate to use an effective particle density of 1.55 g/cm3. This is valid for this data set only.

P13L25: Define "UHSAS".

P14L8: The ratio between the AMS PTOF and the other distributions for large particle sizes is not constantly four as claimed here, but varies strongly.

P14L12: The delayed particles do not only create a bias towards the larger size end of the mass distribution, but as shown in Figure 1 can also leak into the next chopper cycle where they contribute to the lower size end of the distribution and increase the measured background, i.e. lower the effectively measured distribution.

P14L24ff: In section 3.3.2 I miss a discussion on the potential influence of the mixture state of the particles and size-dependent composition on determined CE factors. While the traditional CE factor is defined on a mass basis, using the LS a number-based CE factor (which is partially transformed into a mass-based CE) is used. This relies on the assumption that all particles (the bounced and the detected ones) have the same composition – even for particles smaller than the lower particle size cut-off of the LSP. Externally mixed aerosol should result in different fractions of bouncing particles (e.g.

pure ammonium nitrate and other liquid particles would not bounce while the majority of pure ammonium sulfate particles would bounce). This should at least be discussed.

P15L10: In this context not the PTOF but the MS mode is relevant.

P15L6-24: In Figure 8 the mass and number-based LSSP CE factors sometimes differ by much more than their given uncertainty. What could be the reason for this? Is the assumed uncertainty too small?

P16L7: The authors claim that "particle bouncing physics" was investigated. What kind of particle bouncing physics was measured in this work?

P16L14-16: Please provide clear information on the limitations of the method.

Figure 1/2/3/5/7/8: Present the units as recommended by IUPAP etc.: e.g. "Maximum Scattered Light Signal (V)" should be "Maximum Scattered Light Signal / V".

Figure 1: For the ammonium sulfate particles almost no time dependence of the frequency of delayed particles occurs. How can this be explained? Is this potentially an artefact? What is the "mixed composition"?

Figure 2/4: The labels "a)" and "b)" are very hard to see.

Figure 2, caption: make clear that not the scattered light signal but the intensity of the scattered light signal is shown. It should read "above 0.04 V (red points)", i.e. add "V".

Figure 8 (and Figure S2): Since the altitude at which the measurements were taken is never mentioned or discussed in the manuscript it could be removed from the graph to make it a bit less crowded.

Supplement P1L19: I suggest using the term "dead time" instead of "extra time".

P1L27: "multiply by chopper" should read "multiplied by the chopper".

P2, Table S1: Some of the coefficients change by almost a factor of 2. How large is the uncertainty of determination of these coefficients. Is it expected that they change over

the course of a flight? Would not an average number be the better choice?

---

## Referee Comment (RC2) · Anonymous Referee #1 · 12 Mar 2017

This work represents an important and unique data set that provides a deeper insight into AMS collection efficiency. The authors use airborne measurements made with a light scattering time-of-flight aerosol mass spectrometer (LS-AMS) to draw conclusions about particle bounce and how differences in vaporization processes affect collection efficiency and AMS particle size distributions. This analysis is very valuable to the community of AMS users who frequently must deal with issues related to collection efficiency. Overall the manuscript is of good quality, but the following comments should be addressed before publication.

General Comments:

(1) If the lowered sensitivity of the CToF-AMS during Senex was significant enough to impact the CE (i.e., a substantial fraction of low signals were missed) how do the results of this work, in terms of low CE (e.g., < 0.5), relate to AMS measurements in general? Can we expected that CE is often below 0.5 in other measurements or is this largely an artefact of the low instrument sensitivity?

(2) The filament is hotter than the vaporizer, and is very close to the ion chamber. Do we expect that some areas of the ion chamber are actually hotter than the vaporizer? If so, how does this impact the authors hypothesis that differences in prompt and delayed spectra arise from particles vaporizing at lower temperature or fewer collisions of gas-phase molecules with hot surfaces.

(3) Wording in this manuscript is generally confusing. Problems with the structure of this paper and difficulties with grammar tend to obscures meaning in the manuscript. The Results and Discussion section in particular would benefit from the use of focused topic sentences to guide the reader through the sections and paragraphs. At present it is very difficult for the reader to discern where the discussion is going. More descriptive subsection titles might help in this effort.

(4) The notation used to indicate the different $d_{va}$ types is somewhat confusing. Could $d_{va-MS}$ and $d_{va-LS}$ be used instead?

(5) Various acronyms are used to describe the instrument throughout the paper (LS, LSSP, LS-AMS, LSSP-AMS etc.). The authors would gain some clarity by choosing only one acronym, or by explicitly stating the reasons for using different ones.

Specific Comments:

(1) Abstract:

P1 L18-20: "The individual particle mass from the spectra is proportional to the mass derived from the vacuum aerodynamic diameter determined by the light scattering signals ($d_{va-LS}$ ) rather than the traditional particle time-of-flight (PToF) size ($d_{va}$ )." Do the authors mean that the particle mass from single particle spectra is proportional to the LS $d_{va}$, but not to the pToF $d_{va}$? It is not clear.

P1 L20: "The delayed particles capture about 80% of the total chemical mass compared to prompt ones." Do the authors mean that delayed particles capture 80% of the mass while prompt ones capture 20P1 L24-25: Change to "..especially at larger sizes."

P1 L27-28: Can this be clarified? The measured and calculated CE seemed to agree well when CE > 0.5, but when the calculated (assumed) CE was 0.5 the measured CE was often lower.

(2) Introduction:

P2 L9: The AMS has also been used to quantify sea-salt.

P2 L9-10: I doubt that mass accuracy (usually referring to the accuracy of a particular mass-to-charge ratio) is what is meant here. Please re-phrase for clarity.

P2 L15: " Not all particles introduced to the inlet are vaporized and ionized." Since the particles are not ionized, rather the resulting gas-phase molecules are ionized, I suggest this is re-phrased.

P2 L16: Do the authors wish to suggest here that the CE is the largest source of uncertainty in AMS measurements? This should be referenced and discussed with

more depth.

P2 L18: It could be a bit misleading to simply state the range in CE as 0.3-1. A discussion of the actual distribution of values presented in Middlebrook 2012 would be more informative. How frequently did Middlebrook 2012 see CE <0.5? > 0.5? A more accurate discussion of the magnitude of likely uncertainty is needed here.

P2 L25-28: The paragraph would benefit from an explicit discussion of why an in situ CE is better than comparing to other instruments like the PILS-IC and UHSAS.

P2 L31-32: The description of the LS-derived CE is made needlessly confusing here. Please re-phrase.

P3 L1: Are these the only studies that have applied the LS module to measure CE? "...was about..." Are these the mean values reported by these authors? If so, please state this.

P3 L3: delete "there"

P3 L4-5: change to "...the capability of the LS-AMS to capture CE variations."

P3 L14: Be consistent on the use of LS-AMS and LS-ToF-AMS. LS-ToF-AMS has not been introduced up to this point.

P3 L19: CE, not "The CE..."

P3 L20: change to "particles bouncing on the vaporizer"

P3 L21-22: "delayed" particles have not been introduced yet. It would be more clear to say something like "inefficiently collected particles" or "particles with less then unity collection efficiency." Also, it is not clear what "traditional" refers to.

P3 L22: (and other instances) Is "size resolved mass distribution" the best way to describe pToF data? It appears redundant.

(3) Methods:

P4 L24: Can the authors describe the implications of this non-linearity in sensitivity to small signals?

P5 L14: Where there frequently more than once mass spectrum per particle? Could the authors include representative or averaged particle event profiles in terms of summed signal over time?

P6 L9: How is this known?

P7 L6-17: Were these frag table corrections applied to LSSP mass spectra? Or, were these applied to MS mode only, in order to calculate CE using the CDCE algorithm?

(4) Results and Discussion:

P8-9/S3.1.1: The point of this section seems to be that delayed particles are likely the result of bouncing in the vaporizer. This is important for the reader to understand and it would really help the reader if the authors stated the major conclusion of this section at some point, either at the beginning or the end of the section. At present it is buried in this middle.

P8 L13: "...as expected.." It would help the reader to state why this is expected.

P8 L17: perhaps state which measure of $d_{va}$ is referred to here

P8 L18-20 (Figure 2): Plots of summed MS signal (minus air) versus $d_{va}$ would fit well in figure 2 and with the discussion of detection limit for a single particle, and don't seem to be in the paper at all. This plot should be coloured in a similar manner for Figure2a, showing the particle events above and below the MS detection limit

P8 L20: "about 36 ions" seems vague, is there a measured uncertainty in the single ion signal that can be converted into a range of ions per spectra here?

P8 L32: change to "subsequently vaporize off..."

P8 L33: should this read (NH4)2SO4 vaporizes in less than 50us?

P9 L1: rephrase to: "...,with no measurable bounce, does not produce delayed particle events"

P9 L1: Except for the reference to the figure, it is not clear here that the authors refer to their own laboratory data. This is a grammar issue.

P9 L 1-5: What are the velocities of particles over the size range we expect? Much larger than this, but not obvious to the reader. What fraction of particles would have this velocity?

P9 L17-18: This is a confusing sentence. Consider re-phrasing or breaking into two sentences. Reference to Figure2b would be useful here, too.

P9 L20-22: This sentence seems to repeat the information given in L17-18. Also, it is not clear what is meant by "...which are also defined by the time differences in the Figure1 histograms." What is being defined here? As far as I understand, Figure2B is used to identify prompt and delayed particles.

P9 L25: What exactly are these improvements and how do these definitions differ? This needs some explanation, rather than just referring to the table. It would help the reader to state exactly why the definition of Cross 2009 would result in more delayed particles.

P9 L30: It would be useful the remind the reader here that the "prompt + delayed fraction" represents the fraction of particles that are detected by the mass spectrometer. This quantity could be affected by (1) particle bounce or (2) below detection limit number of ions. (1) and (2) are both impacted by particle composition and phase, as well as instrument sensitivity, so it is unclear to me why we should expect very similar fractions between studies. The various factors affecting prompt+delayed fractions need to be explained more clearly than is currently done in P8 L31-32 and P10 L1-13.

P10 L9: kinetic momentum or kinetic energy?

P10 18-19: What might this mechanism be? It would be more helpful to the reader if the authors summarized their hypothesis here before embarking on a long discussion.

P10 L21: Different physical properties that resulted in different bouncing characteristics?

P10 L23-25: This sentence is very difficult to follow.

P10 L23-28: Do the individual particle event profiles (i.e., ions in an individual mass spectrum versus time) collected from this laboratory data (and/or field data?) show longer evaporation times for delayed particles? Could the authors collect laboratory

data at a higher frequency to allow for this analysis?

P11 L 7-8: Is this because delayed particles are evaporating at a lower temperature? This section could be re-organized to make the conclusion more clear

P11 L11: "processes for" is really unclear. Do the authors mean "processes during detection" or "processes at the vaporizer"?

P11 L11-12: Since the filament is hotter than the vaporizer, what does this tell us about the vaporization of delayed particles? do delayed particles generally show lower total ion signals?

P11 L15: Canagaratna et al., JPCA, 2015 (DOI: 10.1021/jp510711u) could provide some supporting discussion here about vaporization temperature and fragmentation. At present this discussion seems to lack references.

P12 L1-5: Is a simplified UMR fragmentation table being applied to LSSP data to derive total sulphate and organic masses?

P13/S 3.2.1: Despite being observed previously, that the correlation in Figure 6 shows the LSSP data to be representative of the bulk relative composition is notable since all sizes are not detected with the same efficiency in LSSP mode.

P13 L13: What is meant by "slightly independent" is ambiguous

P13 L26: define UHSAS here

P13 L27: Did the authors measure transmission in their lens? Lens transmission as illustrated in Liu et al., may not be representative (or expected) for most AMS lenses. See this User's Meeting presentation: http://cires1.colorado.edu/jimenez-group/UsrMtgs/UsersMtg16/CampuzanoJostLensAndePToF-AMSUsersMeeting2015.pdf

P14 L6: Please be specific about what "these two" are

P14 L7: "Uncertainty by including..." should be "Uncertainty introduced by including..."?

P14 L9: please refer to a specific section

P14 L28: It might be helpful to briefly define mass-based CE here with a brief reference to the size distributions

P15 L3: Was this the case during SENEX?

P15 S3.3.2: What causes CE <0.5? Other factors beyond RH and acidity? Did chemical composition or other parameters vary more significantly with CE on the July 6, 2013 flight relative to other flights?

(5) Conclusions:

P16 L7: change to: ..."and uses these measurements to investigate the AMS CE in situ."

P16 L11: What does "reduce" mean in the context?

P 16 L23-24: Under what conditions? Specifically low CE?

(6) Figures:

Figure 2: This figure would benefit from the addition of a total MS signal (or total non-air ions) versus $d_{va}$, coloured to indicate which particle events are below the detection limit. Prompt and delayed particles should also be indicated in this plot. Part (b): Please colour data in this figure to indicate which particle events are classified as prompt and delayed. I assume that null particles are not included in this figure, this could be explicitly stated.

Figure 7: This figure is missing labels (i.e., a, b, and c). That the difference between the $d_{va-LS}$ and $d_{va}$ size distributions arises because delayed particles are excluded from the $d_{va-LS}$ distribution and not because $d_{va-LS}$ somehow accounts for the delay time should be make clear here and in the text

(7) Supplement:

Table S1: Make it clear that these corrections are (presumably) applied to single particle data? What frag table/assumptions were applied to single particle data? How can you have a negative $CO_2$ correction?

Figure S2: There are times of significant discrepancy between the calculated CE and measured LS CE. Is there any more information that can explain this? What do the particle size distributions look like? How acidic are these particles? Is their mixing state changing substantially? Could the authors attempt to understand these differences?

---

## Author Comment (AC1) · 12 Aug 2017

This work represents an important and unique data set that provides a deeper insight into AMS collection efficiency. The authors use airborne measurements made with a light scattering time-of-flight aerosol mass spectrometer (LS-AMS) to draw conclusions about particle bounce and how differences in vaporization processes affect collection efficiency and AMS particle size distributions. This analysis is very valuable to the community of AMS users who frequently must deal with issues related to collection efficiency. Overall the manuscript is of good quality, but the following comments should be addressed before publication.

We thank the reviewer for their comments.

General Comments:

(1) If the lowered sensitivity of the CToF-AMS during Senex was significant enough to impact the CE (i.e., a substantial fraction of low signals were missed) how do the results of this work, in terms of low CE (e.g., < 0.5), relate to AMS measurements in general? Can we expected that CE is often below 0.5 in other measurements or is this largely an artefact of the low instrument sensitivity?

To be clear, the CE measurements reported here were obtained using the light scattering module, which examines signals from individual particles. When the mass spectrometer has a low sensitivity, it is more difficult to detect mass spectral signals from individual particles. This effect is apparent in Figure 3, where the null fraction is higher for particles less than 350 nm compared to 350-500 nm. When the smallest particles were removed from the CE measurements here, the average CE increased only slightly (by < 5%) because relatively few particles were detected by the light scattering system at small sizes. Hence, the low sensitivity of the mass spectrometer during SENEX does not have a huge impact on the measured CE and we revised the manuscript accordingly. For most of the SENEX dataset, measured CEs that were less than 0.5 were within the experimental uncertainties, with the exception of the flight with biomass burning influence on July 3. We added a new paragraph to discuss those specific results. We also added more text to the discussion and conclusions about the effect of low sensitivity on these results.

(2) The filament is hotter than the vaporizer, and is very close to the ion chamber. Do we expect that some areas of the ion chamber are actually hotter than the vaporizer? If so, how does this impact the authors hypothesis that differences in prompt and delayed spectra arise from particles vaporizing at lower temperature or fewer collisions of gas-phase molecules with hot surfaces.

This question was also raised by the second referee.

There are many surfaces on the interior of the vaporization/ionization source region for bounced particles to strike, including a baffle to reduce stray light from the hot filament from reaching the LS detector. This baffle has a small hole to transmit particles into the vaporization/ionization source region and roughly forms the side opposite of the vaporizer. The

microporous vaporizer itself could provide the first surface for particles to strike after an initial bounce. The ionization chamber has ceramic washers and is thermally grounded compared to the baffle, filament, and vaporizer. Heat is conducted radiantly between the various hot surfaces and the ionization chamber, and we do not expect much heat transfer in the vacuum chamber. Since we do not have thermocouples on the ionization chamber or the baffle, there are no direct measurements of their temperatures.

The differences in fragmentation patterns for prompt versus delayed particles may indicate differences in the vaporization temperatures or the number of wall collisions. Other studies have shown that larger m/z ion fragments (less fragmentation) on average are produced when the vaporizer is at lower temperatures (Drewnick et al., 2015;Canagaratna et al., 2015), and smaller m/z ion fragments (more fragmentation) on average appear when the particles are confined in the vaporizer such that there are more particle (or gas) collisions with the same temperature surface prior to ionization (Hu et al., 2016). Therefore, it is plausible that the delayed particles vaporized at lower temperatures than the prompt particles or that the vapors from the delayed particles experienced few collisions. We cannot preclude either possibility.

We note, however, an inconsistency in this explanation with the vaporization time scales which were not significantly longer (such as doubled) for the delayed particles. Our field observations for prompt and delayed particles (differences in fragmentation patterns without differences in vaporization time scales) were also reported for nominally-identical secondary organic aerosols formed in the laboratory (Robinson et al., 2017). Other laboratory studies of varying the vaporizer temperature indicate that slower vaporization occurs with lower vaporizer temperatures and the vaporization timescales on average increase with coincident reduction in fragmentation at a function of decreased vaporizer temperature (Drewnick et al., 2015). Therefore, the vaporization/ionization process for delayed particles is probably not identical to simply changing the vaporizer temperature. We added more discussion of this in the manuscript.

(3) Wording in this manuscript is generally confusing. Problems with the structure of this paper and difficulties with grammar tend to obscures meaning in the manuscript. The Results and Discussion section in particular would benefit from the use of focused topic sentences to guide the reader through the sections and paragraphs. At present it is very difficult for the reader to discern where the discussion is going. More descriptive subsection titles might help in this effort.

We made many changes to the manuscript to reduce confusion, revising text, moving parts to different paragraphs/sections, and adding more focus topic sentences and descriptive subsection titles.

(4) The notation used to indicate the different dva types is somewhat confusing. Could $d_{va-MS}$ and $d_{va-LS}$ be used instead?

We originally used the notation of $d_{va}$ in the submitted manuscript to designate the traditional AMS measurement of particle size that it is measured with the mass spectrometer along with the standard calculation of vacuum aerodynamic diameter from the effective density and particle size. To make the manuscript notation more clear, $d_{va-MS}$ and $d_{va-LS}$ instead of $d_{va}$ and $d_{va-LS}$ are used now in the various places where the method of deriving vacuum aerodynamic diameter was related to either the MS or LS timing calibration. We kept the notation of $d_{va}$ wherever the general term of vacuum aerodynamic diameter was appropriate.

(5) Various acronyms are used to describe the instrument throughout the paper (LS, LSSP, LS-AMS, LSSP-AMS etc.). The authors would gain some clarity by choosing only one acronym, or by explicitly stating the reasons for using different ones.

We searched through the "LS" acronyms and kept those which are needed to designate either the light scattering module (LS, used when discussing the module), the AMS instrument with the light scattering module (LS-AMS, used when discussing the instrument), and the light scattering single particle (LSSP) mode of the LS-AMS for acquiring data (used when discussing the data acquisition mode or the data acquired from this mode). We removed all instances of LS-ToF-AMS, which designates a time-of-flight mass spectrometer on the LS-AMS but was not necessary for this paper.

The acronyms are defined in the manuscript as follows:

"A light scattering (LS) module has been developed to integrate into AMS instruments (LS-AMS) to detect single particles before they impact on the vaporizer (Cross et al., 2007)."

"Two operating modes of this AMS instrument are the same as for instruments without an LS module. The mass spectrum (MS) mode is used to measure ensemble submicron aerosol mass concentrations, and particle time-of-flight (PToF) mode is used to measure size dependent submicron mass concentrations. The added light-scattering single particle (LSSP) mode with the LS-AMS is used to measure single particle size and non-refractory mass."

Specific Comments:

(1) Abstract:

P1 L18-20: "The individual particle mass from the spectra is proportional to the mass derived from the vacuum aerodynamic diameter determined by the light scattering signals (dva–LS) rather than the traditional particle time-of-flight (PToF) size (dva)." Do the authors mean that the particle mass from single particle spectra is proportional to the LS dva, but not to the pToF dva? It is not clear.

Yes. Changed "The individual particle mass from the spectra is proportional to the mass derived from the vacuum aerodynamic diameter determined by the light scattering signals (dva–LS) rather than the traditional particle time-of-flight (PToF) size (dva)."

To "The particle mass from single particle spectra is proportional to that derived from the light scattering diameter ($d_{va\text{-}LS}$) but not to that from the particle time-of-flight (PToF) diameter ($d_{va\text{-}MS}$) from the time of the maximum mass spectral signal."

P1 L20: "The delayed particles capture about 80% of the total chemical mass compared to prompt ones." Do the authors mean that delayed particles capture 80% of the mass while prompt ones capture 20

No. Changed "The delayed particles capture about 80% of the total chemical mass compared to prompt ones."

To "The total mass spectral signal from delayed particles was about 80% of that from prompt ones for the same $d_{va\text{-}LS}$."

P1 L24-25: Change to "..especially at larger sizes."

Changed.

P1 L27-28: Can this be clarified? The measured and calculated CE seemed to agree well when CE>0.5, but when the calculated (assumed) CE was 0.5 the measured CE was often lower.

Yes. Changed "The measured CE generally agreed with the CE parameterization based on ambient chemical composition, including for acidic particles that had a higher CE as expected from previous studies."

To "The measured CE agreed well with a previous parameterization when CE > 0.5 for acidic particles, but was sometimes lower than the minimum parameterized CE of 0.5."

The preceding sentence about reduced sensitivity was deleted because we found it did not have a large impact on the measured CE for this study.

(2) Introduction:

P2 L9: The AMS has also been used to quantify sea-salt.

While sea salt has been measured with the AMS at a slightly higher vaporizer temperature in some specific studies (e.g., (Ovadnevaite et al., 2012)) it has not been widely used by the community to quantify sea-salt. We think our statement "has been widely used to measure the real time aerosol ensemble organic, sulfate, nitrate, ammonium, and chloride (non-sea salt) mass loadings globally" is reasonable.

P2 L9-10: I doubt that mass accuracy (usually referring to the accuracy of a particular mass-to-charge ratio) is what is meant here. Please re-phrase for clarity.

Changed "Evaluation of aerosol mass accuracy measured by the AMS is important to estimate the impact of aerosols on climate, biogeochemical health, and aerosol formation processes such as aerosol hygroscopicity (Levin et al., 2014;Brock et al., 2016) and aerosol acidity (Hennigan et al., 2015;Zhang et al., 2007)(b)."

To "Evaluation of aerosol mass loading accuracy measured by the AMS is important to estimate the impact of aerosols on climate, biogeochemical health, and aerosol formation processes such as aerosol hygroscopicity (Levin et al., 2014; Brock et al., 2016) and aerosol acidity (Hennigan et al., 2015; Zhang et al., 2007b)."

P2 L15: " Not all particles introduced to the inlet are vaporized and ionized." Since the particles are not ionized, rather the resulting gas-phase molecules are ionized, I suggest this is re-phrased.

Changed "Not all particles introduced to the inlet are vaporized and ionized."

To "Not all particles introduced to the inlet are thermally desorbed and the resulting gas-phase molecules ionized."

P2 L16: Do the authors wish to suggest here that the CE is the largest source of uncertainty in AMS measurements? This should be referenced and discussed with more depth.

In this paragraph, we are focusing on CE and the dominant uncertainty in CE from bouncing particles related to accurate measurements from the AMS. An evaluation of the overall AMS uncertainties is beyond the scope of this paragraph. We added some sentences to clarify our discussion here.

"The net overall transmission and detection efficiency is called collection efficiency (CE) and expressed as the product of aerodynamic lens transmission efficiency for spherical particles, the loss of transmission due to particle beam broadening, and the efficiency of detecting a particle

that vaporizes on impaction (Huffman et al., 2005). The particle transmission losses in the AMS lens can be determined by the lens particle transmission curve (e.g., (Bahreini et al., 2008)). Beam width probe experiments (Huffman et al., 2005) found that CE less than 100% is not due to particle beam broadening but is likely due to particles bouncing off the vaporizer. Therefore, with well-characterized lens transmission efficiency, the dominant uncertainty in CE is likely to be the particles bouncing on the vaporizer."

P2 L18: It could be a bit misleading to simply state the range in CE as 0.3-1. A discussion of the actual distribution of values presented in Middlebrook 2012 would be more informative. How frequently did Middlebrook 2012 see CE<0.5?>0.5? A more accurate discussion of the magnitude of likely uncertainty is needed here.

Changed "It ranges from 0.3 to one in ambient measurements (Middlebrook et al., 2012) and therefore may induce an uncertainty as large as a factor of 3 in the aerosol mass measured by AMS."

To "It ranged from 0.3-1 with the highest frequency around 0.45 in our ambient measurements (Middlebrook et al., 2012). This indicates that the varying CE in some cases may induce an uncertainty as large as a factor of 3 in the aerosol mass measured by AMS."

P2 L25-28: The paragraph would benefit from an explicit discussion of why an in situ CE is better than comparing to other instruments like the PILS-IC and UHSAS.

Added "CE obtained by comparison of AMS measurements with other instruments like the PiLS-IC and UHSAS may be affected by the measurement uncertainties of other instruments and lens transmission efficiency. In situ measurements of CE can directly quantify the detection efficiency of particles that impact the vaporizer."

P2 L31-32: The description of the LS-derived CE is made needlessly confusing here. Please rephrase.

Changed "This provides an opportunity to directly investigate the in situ CE of the AMS by comparing the number or mass of particles optically and chemically detected to the total number or mass of particles optically sensed."

To "The LS-AMS determines the in situ CE by comparing the number (or mass) of particles detected by the mass spectrometer signals to the total number (or mass) of particles detected by the scattered light signals."

P3 L1: Are these the only studies that have applied the LS module to measure CE? "... was about ..." Are these the mean values reported by these authors? If so, please state this.

There is one additional study reporting an average CE from the LS module in eastern Canada (Slowik et al., 2010), but the details of the LSSP results were not included in the paper.

Changed "Using the LS-AMS instrument, the CE for ambient particles at three ground sites near Mexico City (Cross et al., 2009), Bakersfield, CA (Liu et al., 2013), and downtown Toronto (Lee et al., 2015) was about 0.49, 0.52 and 0.37, respectively."

To "Using the LS-AMS instrument, the mean values of CE for ambient particles at three ground sites near Mexico City (Cross et al., 2009), Bakersfield, CA (Liu et al., 2013), downtown Toronto (Lee et al., 2015), and in the eastern Canadian forest (Slowik et al., 2010) were reported to be 0.49, 0.52, 0.37, and 0.6, respectively."

P3 L3: delete "there"

Deleted

P3 L4-5: change to "...the capability of the LS-AMS to capture CE variations."

Changed "… the capability of LS-AMS in capturing CE variations."

To "… the capability of LS-AMS to measure CE variations."

P3 L14: Be consistent on the use of LS-AMS and LS-ToF-AMS. LS-ToF-AMS has not been introduced up to this point.

Changed all instances of "LS-ToF-AMS" to "LS-AMS"

P3 L19: CE, not "The CE … "

Changed

P3 L20: change to "particles bouncing on the vaporizer"

Changed "particle bouncing on vaporizer" to "particles that bounce on the vaporizer"

P3 L21-22: "delayed" particles have not been introduced yet. It would be more clear to say something like "inefficiently collected particles" or "particles with less then unity collection efficiency." Also, it is not clear what "traditional" refers to.

Changed "delayed" to "inefficiently collected"

Changed "traditional AMS size resolved mass distribution" to "AMS particle time-of-flight data"

P3 L22: (and other instances) Is "size resolved mass distribution" the best way to describe pToF data? It appears redundant.

Changed "size resolved mass distribution" to "particle time-of-flight data" or "speciated mass distribution" where appropriate.

(3) Methods:

P4 L24: Can the authors describe the implications of this non-linearity in sensitivity to small signals?

Added "This can lead to ions with low signal to be biased low and ions with high signal to be biased high. The overall AMS response is generally not affected by decreased sensitivity since the calibration ionization efficiency is linearly proportional to the signal from air over a wide range of sensitivities (see the next to the last paragraph in this section for details)."

P5 L14: Where there frequently more than once mass spectrum per particle? Could the authors include representative or averaged particle event profiles in terms of summed signal over time?

Yes and yes. The total mass spectral signal often had a sharp maximum spectrum followed by lower total signals in next few spectra. An example of the raw LSSP data for a prompt particle from SENEX is now shown in Figure S2.

P6 L9: How is this known?

This sentence and the one preceding it were moved from the experimental section to the discussion section and revised. The new paragraph topic is the offset time that was observed in comparing the observed time for maximum mass spectral signal to the time estimated from the maximum light scattering signal.

The new relevant sentence is properly cited as "For prompt particles, not including the uncertainty in the time for particles to pass through the chopper, the largest uncertainty in PToF sizing from $t_{ms}$ is believed to be due to vaporization (Huffman et al., 2005;Day et al., 2011)."

P7 L6-17: Were these frag table corrections applied to LSSP mass spectra? Or, were these applied to MS mode only, in order to calculate CE using the CDCE algorithm?

No and yes. The fragmentation table corrections were used for the MS data and not the LSSP data. Since the LSSP data are analyzed for the ±5 spectra around the maximum particle signal, the calculations of speciated mass from each particle do not include the air fragmentation pattern. We removed Table SI with the fragmentation table corrections and revised the text to make this point clear.

(4) Results and Discussion:

P8-9/S3.1.1: The point of this section seems to be that delayed particles are likely the result of bouncing in the vaporizer. This is important for the reader to understand and it would really help the reader if the authors stated the major conclusion of this section at some point, either at the beginning or the end of the section. At present it is buried in this middle.

We revised the first part of the results section to explain more clearly how the LSSP data were categorized, so now the major conclusion is at the beginning of the last paragraph in Section 3.1.2 Intensities of the LSSP signals, particle $d_o$ sizing, and categories based on intensity.

P8 L13: " ... as expected.." It would help the reader to state why this is expected.

Changed "… as expected."

To "… as scattered light signals are approximately proportional to cube of the particle physical diameter."

P8 L17: perhaps state which measure of dva is referred to here

Changed "dva" to "$d_{va\text{-}MS}$"

P8 L18-20 (Figure 2): Plots of summed MS signal (minus air) versus dva would fit well in figure 2 and with the discussion of detection limit for a single particle, and don't seem to be in the paper at all. This plot should be coloured in a similar manner for Figure2a, showing the particle events above and below the MS detection limit

We added the suggested plot as a new Fig. 2a, showing the total MS signal versus the maximum scattered light signal, and included the two thresholds for detection (LS and MS). This appears to clearly demonstrate how the LSSP data were categorized from the two thresholds. The points are colored as shown in Fig. 3, with black points below the optical detection limit as in the old Fig. 2a. We also added more discussion in the text about the two detection limits. Note that a form of the suggested figure is already presented in Fig. 5: single particle mass from the spectra versus single particle mass from the size (proportional to the cube of $d_{va\text{-}LS}$). Points where the LSSP data were below the MS detection limit were not included since this figure was about the signals from prompt and delayed particles.

P8 L20: "about 36 ions" seems vague, is there a measured uncertainty in the single ion signal that can be converted into a range of ions per spectra here?

We added text just after this statement: "The average LSSP MS detection limit for all flights analyzed is 38 ions (see Table 1). For dry particles composed of pure ammonium nitrate, this MS

detection limit corresponds to an individual particle with a minimum size $d_p$ of about 200 nm or $d_{va-MS}$ of about 275 nm."

P8 L32: change to "subsequently vaporize off … "

Changed "… subsequently vaporizing off of another surface …"

To "…subsequently vaporize on another surface …"

P8 L33: should this read (NH4)2SO4 vaporizes in less than 50us?

Changed to "… ammonium sulfate vaporizes in less than 60 µs …"

P9 L1: rephrase to: "… , with no measurable bounce, does not produce delayed particle events"

Changed

P9 L1: Except for the reference to the figure, it is not clear here that the authors refer to their own laboratory data. This is a grammar issue.

Changed "Laboratory data" to "Moreover, our laboratory data"

P9 L 1-5: What are the velocities of particles over the size range we expect? Much larger than this, but not obvious to the reader. What fraction of particles would have this velocity?

Changed "Given the range of particle velocities measured in the PToF region prior to particle impaction on the vaporizer …"

To "Given the range of particle velocities measured in the PToF region of 70-150 m s$^{-1}$ for particles between 1000 and 100 nm, respectively, prior to particle impaction on the vaporizer …"

P9 L17-18: This is a confusing sentence. Consider re-phrasing or breaking into two sentences. Reference to Figure2b would be useful here, too.

The first five sentences of this section were revised and Figures 1 and 2 were referenced for clarity of the definitions for null, prompt, and delayed particles.

"For the particles that are detected by light scattering, the overall fraction of particles that are detected by the mass spectrometer are affected by particle bounce, refractory composition, and/or ion detection limit. The particles that are detected optically but not detected chemically with the mass spectrometer for any of these reasons are defined as "null" particles (see Figure 2a). Prompt and delayed particles are defined above in Section 3.1.1 by the timing of their mass spectral signals (see Figures 1 and 2c)."

P9 L20-22: This sentence seems to repeat the information given in L17-18. Also, it is not clear what is meant by " … which are also defined by the time differences in the Figure1 histograms." What is being defined here? As far as I understand, Figure2B is used to identify prompt and delayed particles.

We revised L17-18. The histograms in Figure 1 are now more clearly explained in the new Section 3.1.1 (on the timing of the LSSP signals) to define the timing boundaries for "prompt" and "delayed" particles, which are included in Figure 2c (old Figure 2b).

P9 L25: What exactly are these improvements and how do these definitions differ? This needs some explanation, rather than just referring to the table. It would help the reader to state exactly why the definition of Cross 2009 would result in more delayed particles.

Changed "Due to the improvement of LS module and its data analysis software, the definition to separate prompt and delayed particles in this study is the same as the more recent study by Lee et al. (2015) but different from that in Cross et al. (2009) and Liu et al. (2013) (see table 1 for comparison). The definition in Cross et al. (2009) would likely result in more delayed particles compared to this study and a consistently higher delayed particle fraction was reported in their study."

To "Delayed particles were defined by Cross et al. (2009) as those with $t_{ms} > (t_{est}+200$ μs), whereas here they are defined as those with $t_{ms} > (t_{est}+t_{offset}+3×$Gaussian width $≈ t_{est}+530$ μs) (see Table 1). Consequently, a higher delayed particle fraction was reported in Cross et al.'s (2009) study."

P9 L30: It would be useful the remind the reader here that the "prompt + delayed fraction" represents the fraction of particles that are detected by the mass spectrometer. This quantity could be affected by (1) particle bounce or (2) below detection limit number of ions. (1) and (2) are both impacted by particle composition and phase, as well as instrument sensitivity, so it is unclear to me why we should expect very similar fractions between studies. The various factors affecting prompt+delayed fractions need to be explained more clearly than is currently done in P8 L31-32 and P10 L1-13.

The definitions of "prompt" and "delayed" particles were added to text in Section 3.1.1. We also added the following text to the beginning of Section 3.1.3 Fractions of prompt, delayed, and null particles.

"Our previous studies suggest that particle phase is an important factor determining whether or not particles bounce. Larger particles may also have a tendency to bounce more than smaller particles. Some particles contain refractory components (e.g., soot, sea salt, or dust) that do not vaporize at typical AMS vaporization temperatures. If particles are predominantly composed of these species, they will not produce viable mass spectra. The ion detection limit is relevant for particles that are too small to generate sufficient ions from an individual particle. Furthermore, this detection limit varies with the sensitivity of the specific mass spectrometer and likely between different mass spectrometers. Note that the optical detection limit can vary between instruments too.

As discussed later in Sections 3.2.1 and 3.3.2, the in situ CE from the LS-AMS is defined as the fraction of particles that are detected by the mass spectrometer (all particles detected by light scattering minus the particles without viable mass spectra) divided by the total number of particles detected by light scattering. This definition of CE is equivalent to the fraction of prompt plus delayed particles. The mass-based in situ CE detected by light scattering is defined as the mass of particles from their MS signals divided by the mass of particles from their volume determined by their light scattering size, $d_{va-LS}$, and an estimated density. Thus, both the fractions of these particles and CE from LSSP data are affected by phase, size, composition, and mass spectrometer sensitivity."

P10 L9: kinetic momentum or kinetic energy?

Changed "kinetic momentum" to "kinetic energy"

P10 18-19: What might this mechanism be? It would be more helpful to the reader if the authors summarized their hypothesis here before embarking on a long discussion.

Changed "The chemical ion signals of prompt and delayed particles are different on average, which may indicate the mechanism for producing these delayed particles. Differences in the

spectra of prompt and delayed particles have not been previously reported, and it is possible that chemical differences may have caused these particles to have different properties."

To "The chemical ion signals of prompt and delayed particles are different on average, which may indicate the mechanism for producing these delayed particles. Two possible explanations are that the delayed particles vaporized at different conditions because they did not vaporize upon initial impaction or that chemical and/or physical differences may have caused these particles to have different bouncing characteristics, resulting in different spectra."

P10 L21: Different physical properties that resulted in different bouncing characteristics?

See above response.

P10 L23-25: This sentence is very difficult to follow.

Changed "To better interpret the difference in spectra of prompt versus delayed particles from SENEX, LSSP data with varying detector sensitivities were collected and analyzed for dry, poly-dispersed, laboratory particles composed of internally-mixed organic dicarboxylic and carbonyl acids, ammonium organic acid salts, ammonium sulfate, and ammonium nitrate."

To "To better interpret the difference in spectra of prompt versus delayed particles from SENEX, we conducted limited laboratory experiments using nominally identical particles that produced both prompt and delayed spectra. For these experiments, dry, poly-dispersed particles were generated from a simple aqueous mixture of organic dicarboxylic and carbonyl acids, ammonium organic acid salts, ammonium sulfate, and ammonium nitrate."

P10 L23-28: Do the individual particle event profiles (i.e., ions in an individual mass spectrum versus time) collected from this laboratory data (and/or field data?) show longer evaporation times for delayed particles? Could the authors collect laboratory data at a higher frequency to allow for this analysis?

We examined the peak widths and did not see any difference between the prompt and delayed particle spectra. This is discussed in the second to last paragraph of this section on the laboratory particles. Spectra could be collected on faster time scales, but the signal-to-noise at each mass-to-charge ratio for individual particles will be lower and higher frequency data may not yield good results unless large particles are used.

P11 L 7-8: Is this because delayed particles are evaporating at a lower temperature? This section could be re-organized to make the conclusion more clear

We revised this entire section based on incorporating the results from recent papers on other laboratory particles with an LS-AMS (Robinson et al., 2017) and on fragmentation patterns due to varying temperatures or using the new capture vaporizer (Canagaratna et al., 2015; Hu et al., 2016). Essentially we are unable to conclude exactly why the spectra are different and added several paragraphs on this topic.

"Differences in the spectra of prompt and delayed particles were also recently reported for ammonium sulfate and laboratory-generated secondary organic aerosols (SOA), both with a substantial fraction of delayed particles (Robinson et al., 2017). Indeed, the pattern of sulfate ions between prompt and delayed particles in their study is similar to what is shown in Figure 4 for the SENEX and mixed composition particles and what we observed with our ammonium sulfate particles (not shown), with the peaks at $m/z$ 81 and 98 more prominent in the delayed particles and the peaks at $m/z$ 48 and 64 more prominent in the prompt particles. Hence, it appears that the delayed particles have less fragmentation of sulfate and nitrate ions than the

prompt ones. Thus, differences between the prompt and delayed spectra in the SENEX data set could be solely due to identical particles vaporizing under distinct conditions. All of the differences in the mass spectra are consistent with more thermal decomposition and fragmentation with prompt particles.

The consistent differences in the mass spectra between prompt and delayed particles from both ambient and nominally-identical chemical composition laboratory particles suggests that there are different processes at the vaporizer for prompt and delayed particles. The two likely explanations are that the evolved gas from the delayed particles experienced fewer wall collisions before ionization and/or that the delayed particles vaporized from surfaces at different temperatures. The bottom of the conical vaporizer, where prompt particles should vaporize, is a location where the evolved gas molecules likely experience some wall collisions and could decompose before ionization. Gas molecules from particles vaporizing from other surfaces, such as the top of the vaporizer or the ionization chamber, would experience fewer wall collisions where they could decompose. For both ammonium nitrate and ammonium sulfate particles, there is far more fragmentation when the vapors are contained in a capture vaporizer prior to ionization compared to the conical vaporizer at the same temperature (Hu et al., 2016).

Vaporizer temperatures also affect fragmentation. Lower vaporizer temperatures increase the ions at $m/z$ 46 relative to $m/z$ 30 for ammonium nitrate and at $m/z$ 80 and 81 relative to the other ions for ammonium sulfate (Hu et al., 2016). Spectra from most but not all organic compounds have more fragmentation when vaporized at 600 C than at 200 C (Canagaratna et al., 2015). One piece of evidence about vaporization temperature is the width of the vaporization event, which is wider at lower temperatures and is fairly constant above species-dependent temperatures (Drewnick et al., 2015). For the mixed particles that were generated specifically to compare here with the SENEX data, the peaks in the mass spectra as a function of vaporization time were analyzed to check for slower vaporization. A proxy for peak width in the mass spectra (peak area divided by height) was not statistically significant between the prompt and delayed particles, indicating that the vaporization times were roughly comparable. Since a doubling of the peak widths was not observed for the delayed particles here, it is unlikely that they vaporized at temperatures lower than 300 °C (Drewnick et al., 2015). Because the spectra were saved every 32 µs and vaporization event lengths in the AMS are on the order of 30-60 µs and constant for pure ammonium sulfate particles at temperatures between 400 and 800 °C, the data collected in this brief lab study could not be used to validate the possibility of a less drastic change in vaporization temperature. Equivalent vaporization timescales between prompt and delayed particles along with the same issue of insufficient precision were also reported for the laboratory study of SOA particles (Robinson et al., 2017). The increased fragmentation with a hotter vaporizer could be due to changes in the vaporization process itself or to evolved gas molecules hitting a hotter surface, or both.

While the timing indicates that the delayed particles may have lost a large amount of kinetic energy or bounced multiple times before vaporization, it is unclear where the delayed particles are vaporizing in the vaporization/ionization source region. The microporous vaporizer itself could provide the first surface for particles to strike after an initial bounce. Another possible surface is the baffle which reduces stray light from the hot filament from reaching the LS detector. This baffle has a small hole to transmit particles into the vaporization/ionization source region and roughly forms the side opposite of the vaporizer. Bounced particles could strike the room-temperature baffle if they exit the vaporizer on a trajectory nearly opposite of

the initial particle beam. The interior of the ionization chamber is another possible surface. It has ceramic washers and its mounting points are thermally grounded relative to the baffle, filament, and vaporizer. Heat is conducted radiantly between the various hot surfaces (filament and vaporizer) and the ionization chamber, and by conduction through the thin metal of the ionization chamber. The temperature may be different on the sides near and away from the filament. Since we do not have thermocouples on the ionization chamber or the baffle, there are no direct measurements of their temperatures. This makes it difficult to evaluate the relative importance of vaporization temperature compared to fewer wall collisions of the evolved gases from the delayed particles."

P11 L11: "processes for" is really unclear. Do the authors mean "processes during detection" or "processes at the vaporizer"?

Changed "processes for" to "processes at the vaporizer for"

P11 L11-12: Since the filament is hotter than the vaporizer, what does this tell us about the vaporization of delayed particles?

This is related to the general comment above about vaporization temperature. Since we did not measure the temperatures inside the vaporization/ionization source region and differences between the prompt/delayed particle spectra are inconclusive with regard to vaporization temperatures and/or wall collisions, we removed statements indicating that the delayed particles vaporized at lower (or higher) temperatures than the prompt particles.

do delayed particles generally show lower total ion signals?

For the SENEX data, the ion signals from the delayed particles comprised 80% of the expected mass, whereas the ion signals from the prompt particles accounted for essentially all of the expected mass. This was mentioned in the abstract, discussed later in section 3.1.5 (old section 3.1.4), and shown in Figure 5. We did a more thorough analysis of the average number of ions per unit LS-volume as a function of delay time and did not see a significant decreasing trend like the alpha-pinene SOA that was observed by Robinson et al. 2017. We added a new figure with this analysis (new Figure 6) and added some discussion to the manuscript in the new section 3.1.5.

"The delayed particle mass could be lower on average than the prompt particle mass because the evolved gas from the delayed particles was produced in a region where the vapors are not as efficiently ionized by the electron beam. This explanation was recently proposed based on a laboratory study of monodisperse, alpha-pinene secondary organic aerosols (SOA) that had a low number-based CE (0.30), a significant fraction of delayed particles (53% of all LS particles with MS signals), and a clear trend of more ions per particle for prompt particles than for delayed particles (Robinson et al., 2017). Since the SENEX data are from a range of particle sizes, we normalized the total LSSP ion signals to the cube of their $d_{va\text{-}LS}$ and plotted the averages as a function of delay time up to 3.5 ms. Figure 6 shows the results from two flights on July 6 and July 3 along with results from ammonium sulfate particles and confirms that delayed particles have a slightly lower total ion signal per unit volume than prompt particles. When considering only delayed particles, all three cases show a negative slope for the total ion signal per unit volume as a function of delay time, but individually the slopes are not significant or marginally statistically significant at the 2 sigma level. The largest change appears to be an average of ~50% lower signal for particles with the longest delay times for the flight on July 3. Thus, the efficiency of producing ions from the delayed particles is not always consistently lower than from prompt particles and more experiments are needed to investigate this possible explanation for a

reduced mass. It is not clear from the field data why there was a minimal reduction in ion signal; inefficient ionization of the vapors that evolved from delayed particles cannot be precluded."

P11 L15: Canagaratna et al., JPCA, 2015 (DOI: 10.1021/jp510711u) could provide some supporting discussion here about vaporization temperature and fragmentation. At present this discussion seems to lack references.

We thank the reviewer for pointing out this reference and included it in our discussion here. We also noticed more fragmentation in spectra obtained with the capture vaporizer (Hu et al., 2016) and added that citation too.

P12 L1-5: Is a simplified UMR fragmentation table being applied to LSSP data to derive total sulphate and organic masses?

Yes and no. The LSSP data are analyzed using the standard UMR fragmentation table to derive the mass of each species. However, only the mass spectral signals from the particles are processed, so the contribution for air is removed from the fragmentation table. We added this text to the experimental section.

"The default list of $m/z$ values in Sparrow was used to generate the mass spectra from individual particles. LSSP mass spectra are then processed using the default fragmentation table matrix and the above relative ionization efficiencies, except the contribution from air is removed from the matrix since only the mass spectral signals from the particle are selected."

P13/S 3.2.1: Despite being observed previously, that the correlation in Figure 6 shows the LSSP data to be representative of the bulk relative composition is notable since all sizes are not detected with the same efficiency in LSSP mode.

We thank the reviewer for noting this. We added a new figure, so now this is Figure 7, and added to the end of that section, "The good correlation in Figure 7 indicates that the LSSP data were on average representative of the bulk relative composition in spite of the sampling and processing biases of the single particle technique."

P13 L13: What is meant by "slightly independent" is ambiguous

Deleted "slightly"

P13 L26: define UHSAS here

Defined

P13 L27: Did the authors measure transmission in their lens? Lens transmission as illustrated in Liu et al., may not be representative (or expected) for most AMS lenses. See this User's Meeting presentation: http://cires1.colorado.edu/jimenez-group/UsrMtgs/UsersMtg16/CampuzanoJostLensAndePToF-AMSUsersMeeting2015.pdf

During tests of our pressure controlled inlet that we use on the aircraft, we measured our lens transmission and it was similar to the Liu et al. calculations (Bahreini et al., 2008). We added our reference in this location and added this sentence to the experimental section:

"For comparisons between the AMS and UHSAS instruments, we account for the AMS lens transmission efficiency (Liu et al., 2007), which was measured for our lens to be similar to the predicted transmission (Bahreini et al., 2008)."

P14 L6: Please be specific about what "these two" are

Changed "these two" to "the gray shaded area to the area under the red curve in Figure 8a"

P14 L7: "Uncertainty by including ... " should be "Uncertainty introduced by Including ... "?

Changed

P14 L9: please refer to a specific section

Added "in Section 3.3.1"

P14 L28: It might be helpful to briefly define mass-based CE here with a brief reference to the size distributions

Added " as the ratio of the particle mass from the chemical ion signals (e.g., mass distribution shown as the gray area of Figure 8a) to the particle mass from the laser counts (e.g., area under the red curve in Figure 8a)."

P15 L3: Was this the case during SENEX?

For narrow power plant plumes, yes.

We realized that this section could be split into one about the CE comparisons and one about additional considerations on the LSSP-based CEs. We renamed Section 3.3.2 to "Comparing the LSSP collection efficiency to the parameterized collection efficiency." We then moved the text starting with "There is an assumption …" to the new section (3.3.3) on "Additional considerations of LSSP-based collection efficiencies" and added this informational along with other statements on the application of CE (in general) and the LSSP-based CEs.

The second sentence in that section is: "There is an assumption in applying either a calculated or an in situ CE to the measured mass loadings from MS mode that the undetected mass is the same as the detected mass, which is likely true due to random sampling of air masses with mostly secondary aerosol particles."

We also added these two sentences to the end of the following paragraph: "In power plant plumes that were sampled during SENEX, we sometimes observed a small mode of mostly acidic sulfate particles with mixed-composition accumulation-mode particles. Because these power plant plumes were transected quickly by the aircraft, they were not sampled consistently with LSSP mode, and so the effect of a varying composition with size could not be evaluated here."

P15 S3.3.2: What causes CE < 0.5? Other factors beyond RH and acidity? Did chemical composition or other parameters vary more significantly with CE on the July 6, 2013 flight relative to other flights?

The first two questions seem to be contradictory, since RH and acidity tend to increase CE above 0.5. We think the reviewer may have meant CE > 0.5 in the first question.

For the example shown in Figure 9, the variability in CE above 0.5 is likely due to variations in acidity. The relative humidity for that flight was slightly higher than for the other flights. We rearranged Section 3.3.2, creating a new paragraph:

"The large range of CE values for the flight on July 6 were clearly not due to statistical variation. For this flight, the ratio of MS-mode ammonium to predicted ammonium from full neutralization of sulfate plus nitrate varied more than for all of the other flights, indicating that the aerosol on this flight was at times significantly more acidic on average than for other flights. The relative humidity for this particular flight was also a bit higher on average than the other flights. Thus, the acidity and relative humidity likely had an influence on the CE for this flight more than on the other flights."

On the topic of CE < 0.5, we noted that the null fractions were highest for the flight on July 3 in Figure 3, which corresponded to a lower CE. One possible explanation is that this flight had a higher contribution of biomass burning particles and the aerosol may have contained black carbon (not detected by the AMS) or was less spherical and perhaps not focused as well into the vaporizer. CE of 0.37 was observed for urban particles containing soot (Lee et al., 2015). We made a new paragraph describing this in more detail.

"The default CE parameterization value of 0.5 may be too high during some parts of this flight and large parts of other flights (Figure S3). This was also observed with other field data using a mass-based comparison to evaluated CE (Middlebrook et al., 2012) and may indicate that the default CE of 0.5 is slightly high in the parameterization. One flight in particular (July 3) had the highest null fractions (Figure 3) and corresponding lower CEs for most of the flight compared to other flights. This flight had an overall higher mass loading of refractory black carbon (rBC) from biomass burning in the sampled air (0.36 $\mu$g sm$^{-3}$, whereas the average for all of the flights analyzed here was 0.14 $\mu$g sm$^{-3}$). If an individual particle is mostly rBC, there may not be enough ion signal from the non-refractory components to detect it with the mass spectrometer. In Toronto, a higher null fraction was measured when urban, rBC-containing particles were evaporated with a vaporizer instead of an infrared laser (Lee et al., 2015). It is also possible that the collection efficiency is lower for the organic fraction of biomass burning aerosols. The effect of biomass burning particles on the measured CE with the LS-AMS needs further investigation.."

(5) Conclusions:

P16 L7: change to: ... "and uses these measurements to investigate the AMS CE in situ."

Changed "… uses these measurements to investigate the AMS in situ CE values, particle bouncing physics, and impacts and uncertainties in the AMS size resolved mass distribution."

To "… uses these measurements to investigate the collection efficiency (CE) obtained in situ for ambient particles."

P16 L11: What does "reduce" mean in the context?

Changed "reduce" to "process"

P 16 L23-24: Under what conditions? Specifically low CE?

Added to the last sentence "… when delayed particles occur."

(6) Figures:

Figure 2: This figure would benefit from the addition of a total MS signal (or total non-air ions) versus dva, coloured to indicate which particle events are below the detection limit. Prompt and delayed particles should also be indicated in this plot. Part (b): Please colour data in this figure to indicate which particle events are classified as prompt and delayed. I assume that null particles are not included in this figure, this could be explicitly stated.

We added a form of the first suggested plot as a new Figure 2a, showing the total MS signal versus the maximum scattered light signal, and included the two thresholds for detection (LS and MS). The points are colored as shown in Figure 3, with black points below the optical detection limit as in the previous Figure 2a. We also revised the previous part b as suggested and moved it to part c.

Figure 7: This figure is missing labels (i.e., a, b, and c). That the difference between the dva−LS and dva size distributions arises because delayed particles are excluded from the dva−LS

distribution and not because dva−LS somehow accounts for the delay time should be make clear here and in the text

We added a new figure, so this is now Figure 8, and added labels. All particles that scatter light (including the delayed ones) are included in the $d_{va\text{-}LS}$ distribution. The difference between the two distributions arises because the size from light scattering ($d_{va\text{-}LS}$) is more representative of the particle sizes, whereas $d_{va\text{-}MS}$ is influenced by particles that are delayed. We added text in the caption and manuscript to make this clear.

(7) Supplement:

Table S1: Make it clear that these corrections are (presumably) applied to single particle data? What frag table/assumptions were applied to single particle data? How can you have a negative CO2 correction?

None of these fragmentation table corrections were applied to the LSSP data, which used the default fragmentation table with nothing ("nans") in the table for air. The negative CO2 correction is from a negative ratio of 44/28 in the first part of the flight on 6/29/13 and does not affect the overall processing. This rarely occurs with our aircraft filter measurements. Since the manuscript is focusing on LSSP data, we removed Table S1 and revised the manuscript to make the application of the fragmentation table clear for this work.

Figure S2: There are times of significant discrepancy between the calculated CE and measured LS CE. Is there any more information that can explain this? What do the particle size distributions look like? How acidic are these particles? Is their mixing state changing substantially? Could the authors attempt to understand these differences?

We thank the reviewer for this suggestion since we then re-examined Figure S2 (now Figure S3). We added the following text to the paper:

"There are many factors influencing the point-by-point CE comparison shown in Figure 9b and all three types of CE determinations have limitations. The CE parameterization has about 20% uncertainty (based on Middlebrook et al., 2012) and it could contribute to the noise in the red traces of Figures 9a and S3. In addition, there are statistical variations on the LSSP-based CEs as described above. Also we did not parse the data sets for low statistics from either low LS particle counts or low total bulk mass, which is approximately only an issue for the high-altitude data points. LSSP and MS data were not obtained at the same time and the air masses sampled could be changing rapidly since the aircraft is moving about 100 m s$^{-1}$, such that each CE data point from the parameterization is about 1 km apart, each LSSP CE data point represents a 3 km average, and each LSSP CE data point is 30 km apart. Two minute averages of the CEs from the parameterization were used to generate the comparison plot in Figure 9b. For this flight on July 6, the (observed or calculated) range in CE is much larger on average than for the other flights, it varied on larger temporal (spatial) scales, and the CE variability was outside of the combined error bars."

After considering error bars and the sampling differences, however, it appears that flight averages for both count- and mass-based CE on July 3 were significantly lower than the CDCE values. We believe this is due to a potentially lower CE from biomass burning particles. We revised the manuscript to make a new paragraph and added the following text:

"The default CE parameterization value of 0.5 may be too high during some parts of this flight and large parts of other flights (Figure S3). This was also observed with other field data using a mass-based comparison to evaluated CE (Middlebrook et al., 2012) and may indicate that the

default CE of 0.5 is slightly high in the parameterization. One flight in particular (July 3) had the highest null fractions (Figure 3) and corresponding lower CEs for most of the flight compared to other flights. This flight had an overall higher mass loading of refractory black carbon (rBC) from biomass burning in the sampled air (0.36 µg sm$^{-3}$, whereas the average for all of the flights analyzed here was 0.14 µg sm$^{-3}$). If an individual particle is mostly rBC, there may not be enough ion signal from the non-refractory components to detect it with the mass spectrometer. In Toronto, a higher null fraction was measured when urban, rBC-containing particles were evaporated with a vaporizer instead of an infrared laser (Lee et al., 2015). It is also possible that the collection efficiency is lower for the organic fraction of biomass burning aerosols. The effect of biomass burning particles on the measured CE with the LS-AMS needs further investigation..”

In the first paragraph of the new section (3.3.3), we also changed “There is an assumption for the measured CE that the undetected mass is the same as the detected mass, which is likely true due to random detection.”

To “There is an assumption in applying either a calculated or an in situ CE to the measured mass loadings from MS mode that the chemical composition of the undetected mass is the same as the detected mass, which is likely true during sampling of air masses with mostly mixed secondary aerosol particles.”
In their manuscript "Single particle measurements of bouncing particles and in-situ collection efficiency from an airborne aerosol mass spectrometer (AMS) with light scattering detection", Liao and coworkers use results from airborne measurements with an AMS with implemented LS detection to investigate effects associated with vaporization of individual particles. From these measurements the authors draw conclusions about particle bounce and vaporization in the AMS and how this affects collection efficiency correction factors or measurements of particle size distributions. With this approach and focus the manuscript is well suited for Atmospheric Measurement Techniques and I recommend publication of the manuscript AMT after the following issues were resolved.

Generally, the manuscript is well written. However, at places the authors should select wording that is scientifically more correct (see examples below). Furthermore, some of the conclusions regarding the processes within the ionizer chamber after bouncing of particles off the vaporizer are rather speculative and should be supported by additional evidence or rephrased. Most of the comments below, however, are more associated with clarity or correctness of presentation than with the content of the work. Detailed comments:

We thank the reviewer for carefully examining our manuscript and their positive comments about publication.

P1L15-16: Not only particles larger than ~ 250 nm scatter light, but also smaller ones. Rephrase.

We agree with the reviewer that particles smaller than 250 nm scatter light in our LS-AMS instrument. However, we think that stating a smaller detection limit in the abstract would be misleading, since very few of these smaller particles triggered useful light scattering signals (see old Figure 2a or new Figure 2b) or mass spectra (see right hand axis in old Figure 2b or new Figure 2c). We revised a couple of statements in the results section to clarify our LS detection limits and the statement in the abstract.

In the results section, we changed from "Although particles as small as $d_{va}$ ~170 nm could trigger the LSSP data acquisition (Figure 2), a very small number of these particles were detected and they did not contribute significantly to the LSSP-mode mass distributions until they were larger than $d_{va}$ ~280 nm."

To "Although a few triggers for LSSP data acquisition were recorded for $d_{va-LS}$ as small as ~ 170 nm (Figure 2), a very small number of these particles were detected and there generally were no corresponding MS signals until the particles were larger than $d_{va-LS}$ ~280 nm."

Since we identified $d_{va-LS}$ ~280 nm for meaningful LSSP data in the results, we revised the statement in the abstract from "In this instrument, particles typically larger than ~ 250 nm in vacuum aerodynamic diameter scatter light from an internal laser beam and trigger saving individual particle mass spectra."

To "In this instrument, particles scatter light from an internal laser beam and trigger saving individual particle mass spectra. Nearly all of the single particle data with mass spectra that were triggered by scattered light signals were from particles larger than ~ 280 nm in vacuum aerodynamic diameter."

P1L24: Delayed particles are detected later than "expected". I suggest rewording to something like "delayed particles are detected later than appropriate for their size" or " ... later than expected from their measured velocity".

Changed "… detected at a later time by the mass spectrometer than expected, …"

To "…detected by the mass spectrometer later than expected from their $d_{va\text{-}LS}$ size, …"

P1L26-27: "higher null fractions and corresponding lower CE for this study may have been related to the lower sensitivity of the AMS during SENEX". While the lower sensitivity of the AMS probably causes some of the single particles to produce insufficient signal for single particle identification, all the aerosol signal is probably measured in MS mode where mass spectra are averaged over large time intervals. While the missing of single particle counts using the LSSP mode results in smaller CE factors, the real CE factor should probably higher, thus this should result in a CE factor biased low. Is this correct? How does the "lower sensitivity" affect LSSP and MS mode measurements? How does this potentially affect m/z signals with low relative intensity?

Yes, the reviewer is correct that the CE from LSSP can be biased low when the single particle sensitivity is low. We excluded LSSP data from the first part of the SENEX study because the single particle sensitivity was too low. For these flights, the null fractions of LSSP data were significantly higher and the *m/z* signals with low relative sensitivity in MS mode were not detected as efficiently. This effect for small particles is apparent in Figure 3, where the null fraction is higher for particles less than 350 nm compared to 350-500 nm. When the smallest particles were removed from the CE measurements here, the average CE increased only slightly (by < 5%) because relatively few particles were detected by the light scattering system at small sizes. Hence, the low sensitivity of the mass spectrometer during SENEX does not have a huge impact on the measured CE and we revised the manuscript accordingly.

We mention these topics in separate places in the manuscript and added a bit more text in the revised manuscript for clarification.

Since the CEs from particles near the MS detection limit did not significantly affect the average CEs reported here, we removed the original statement in the abstract, "Relatively higher null fractions and corresponding lower CE for this study may have been related to the lower sensitivity of the AMS during SENEX."

In Section 2 Experimental: "… low detector sensitivity has a non-linear effect on low ion signals, which was previously reported by (Hings et al., 2007). This can lead to ions with low signal to be biased low and ions with high signal to be biased high. The overall AMS response is generally not affected by decreased sensitivity since the calibration ionization efficiency is linearly proportional to the signal from air over a wide range of sensitivities (see the next to the last paragraph in this section for details)."

In Section 3.1.3 (old Section 3.1.2) Fractions of prompt, delayed, and null particles: "For the particles that are detected by light scattering, the overall fraction of particles that are detected by the mass spectrometer are affected by particle bounce, refractory composition, and/or ion detection limit. … The ion detection limit is relevant for particles that are too small to generate

sufficient ions from an individual particle. … Thus, both the fractions of these particles and CE from LSSP data are affected by phase, size, composition, and mass spectrometer sensitivity."

At the end of Section 3.1.3 (old Section 3.1.2), we added "It is important to note that the single particle mass spectral detection limit affects whether or not individual particles are detected in LSSP mode. However, this is not necessarily a factor for the bulk collection efficiency during normal AMS operation because particles are aggregated in both MS and PToF modes. In other words, while single particles may not be detected individually by the mass spectrometer at the smallest sizes, the mass of small particles can be detected when their signals are added together. This is discussed later with the comparisons of mass distributions in Section 3.2.2."

P2L5: aerosol inhomogeneity is blamed on "short atmospheric lifetimes, compared to greenhouse gases". While it is correct that the lifetime of aerosol particles is short compared to that of greenhouse gases, this is irrelevant here. In this context it is more important that aerosol lifetimes are short compared to timescales of mixing processes.

Changed "The spatial and temporal distribution of ambient aerosols is highly inhomogeneous, owing to different sources, meteorological conditions, atmospheric processes, and their relatively short atmospheric lifetime compared to greenhouse gases."

To "The spatial and temporal distribution of ambient aerosols is highly inhomogeneous, owing to different sources, meteorological conditions, atmospheric processes, and their relatively short atmospheric lifetimes."

P2L17: Here and in several other places in the manuscript CE is described as the fraction of mass or number of particles detected by AMS. Traditionally CE is defined as the fraction of mass of particles detected by the AMS. As shown in Figure 3, the bouncing fraction of particles is a function of particle size. Therefore using only the fraction of number of particles that are detected will likely introduce a bias.

We agree that CE is traditionally defined as mass-based and the count-based CE may be different, as briefly described by Huffman et al. (2005). We added a new section (3.3.3), gathering bits of this topic from the manuscript together and discussing its relevance to the LSSP-based CEs and the SENEX results.

"3.3.3 Additional considerations for the LSSP-based collection efficiencies

CE is traditionally defined based on mass comparisons. There is an assumption in applying either a calculated or an in situ CE to the measured mass loadings from MS mode that the chemical composition of the undetected mass is the same as the detected mass, which is likely true during sampling of air masses with mostly mixed secondary aerosol particles. In general for SENEX, the number- and mass-based CE from the LSSP data shown in Figure 9 and Figure S3 are comparable within experimental uncertainties for the particles sampled. In Figure 8a, the integrated mass from the PToF mass distribution using the average CE from the parameterization of 0.6 for the flight on July 6 (dashed curve) is also within the combined experimental uncertainties of the integrated mass from the UHSAS mass distribution (solid black curve). The flight-averaged number-based CE was 0.58. Hence, the SENEX field data did not show any large discrepancies between the number- and mass-based CEs when averaged for the entire flight.

It is also assumed that particles detected optically are representative of all particles sampled by the AMS and have the same chemical composition as the particles that are too small to be detected by LSSP mode. In air masses where newly formed and growing (Aitken mode) particles

are present, this assumption is not necessarily valid. The number- and mass-based CEs may be different if there are significant differences in the number-based CE as a function of size, as briefly described by Huffman et al. (2005). In power plant plumes that were sampled by the PToF mode during SENEX, we sometimes observed smaller acidic sulfate particles with larger mixed-composition particles. Because these power plant plumes were transected quickly by the aircraft, the small particles were not sampled consistently with LSSP mode, and so the effect of a varying composition with size could not be evaluated here."

P2L24: "bouncing on the vaporizer" should read "bouncing off the vaporizer".

This sentence was moved up in the paragraph and changed from "on" to "off," as suggested.

P3L16-18: The sentence starting with "Many airborne ... " contains information that is not relevant here and should be removed.

Deleted

P3L23: Section 2 ("Experimental") already includes a large number of results. Either the section should be renamed or the results moved into the results section.

We added some descriptive text to the Experimental section about the particle sizing methods and moved the sizing results, including Figure 1, into the first part of the results section on sizing. We also split this part of the results section to make the manuscript easier to follow, including changing the sub-headings as follows:

From "3.1.1 LSSP mode particle size measurements and indication of particle bouncing"

To "3.1.1 Timing of the LSSP signals, particle $d_{va-MS}$ and $d_{va-LS}$ sizing, and categories based on time"

and "3.1.2 Intensities of the LSSP signals, particle $d_o$ sizing, and categories based on intensity"

Subsequent headings were renumbered.

P4L11 (and many other places): the unit "bits-ns" should read "bits*ns" since it is the product of the hight (bits) and the width (ns) of the peaks in the ToF mass spectra.

We researched this and found standard notations of units do not use either "-" or "*" when they indicate the units are multiplied together. In addition, units do not have "s" when plural is indicated. So we changed "bits-ns" to "bit ns". We also found that we needed to change any units with "/" to superscripted minus signs.

P4L25: Between "as" and "instruments" a word is missing, either "of" or "for".

Added "for"

P4L28: The LS-AMS is not capable of measuring "single particle mass" but either "single particle composition" or "non-refractory single particle mass".

Added "non-refractory"

P5L2-3: What does "LS-triggered particle scattered light signal" mean? Rephrase.

Changed "… whereas LSSP data were saved for each chopper cycle when LS-triggered particle scattered light signal above threshold."

To "… whereas LSSP data were saved for each chopper cycle when the intensity of the scattered light signal was above the data acquisition threshold."

P5L12: "the corresponding scattered light signal" should probably be "the corresponding light scattering signal".

Changed

P5L13-14: Is there any background subtraction for the mass spectra applied for single particle information?

Yes. The backgrounds from the longest and/or shortest chopper cycle (PToF) times at each m/z are subtracted from the particle region of the LSSP data. We revised the text to include this information.

"The backgrounds from the longest and/or shortest times at each *m/z* in the LSSP spectra from a single chopper cycle are subtracted from the particle region in the same way the background is subtracted for standard PToF data."

P5L16-17: "The cases ... are called coincident particles." This does not make sense. Rephrase.

Changed "The cases when more than one particle passes through the chopper slit per chopper cycle with scattered light signals above the threshold are called coincident particles."

To "Sometimes two or more particles passed through the chopper slit during one chopper cycle and triggered saving LSSP data. These cases are found during post-processing as having more than one peak in the scattered light signal and are called coincident particles."

P5L27: Replace "the third measurement of ... " with "the third type of measurement of ... "

Changed

P6L5: " ... an additional brief amount of time ... ". Can you provide typical numbers for this?

We revised this section about the offset time and the revised paragraph is:

"The offset time (0.3-0.35 ms) is too long to indicate an error in the positions of the laser or the vaporizer and is independent of particle size, as shown later in Figure 2c. By examining the Gaussian fit reported by Lee et al. (2015), we derived an offset time for that study of about 0.42 ms (see Table 1). The offset time of 0.2 ms used by Cross et al. (2009) was not defined in the same way and cannot be compared directly to offsets from the Gaussian fits. The offset time in our data set appears to be systematic and may be related to a number of delays for the MS signals that do not occur for the LS signals. There is some uncertainty in $t_{LS}$ and $t_{ms}$ due to not knowing exactly when the particles transited the chopper slit (Cross et al., 2007), which is about 0.17 ms for our system. This is apparent in the width of the Gaussian distribution for the ammonium nitrate particles (top of Figure 1). For prompt particles, not including the uncertainty in the time for particles to pass through the chopper, the largest uncertainty in PToF sizing from $t_{ms}$ is believed to be due to vaporization (Huffman et al., 2005; Day et al., 2011). Vaporization event lengths (defined as full width half max of the single particle MS signals) depend on the species and can range from 25 μs for ammonium nitrate particles to less than 60 μs for ammonium sulfate particles (Drewnick et al., 2015). The additional time needed for the neutral molecules to move from the vaporizer to the electron beam and for ions to move from the ion source to the orthogonal extraction region are likely much shorter than the offset time. Altogether, these times are too short to account for the systematic offset time, so it most likely incorporates some additional time needed to process the single particle mass spectra during data acquisition in LSSP mode, which may be slightly longer for the HR-ToF AMS used by Lee et al. (2015) than for the C-ToF AMS that we used here."

P6L7: What do you mean with "different signal paths"?

See above revised paragraph on the offset time.

P6L8: "the slowest of these processes and largest uncertainty ... ". Can you provide an estimate on how long this is and how large uncertainty is?

See above revised paragraph on the offset time.

P6L11-30: In these paragraphs there are several statements that should be phrased more correctly, e.g. "The second method ... is the vacuum aerodynamic diameter ... ", " ... calibrated with known particles ... ", " ... where the calculated particle velocity uses ... ", or " ... that the laser not only counts ... but also counts ... " (the laser does not count, it can be used to count).

These paragraphs were rearranged and rephrased.

P6L22: How can the maximum of the Gaussian distribution be before Toffset, which is defined as the mean of the distribution?

The reviewer is correct that the offset time is defined as the mean of the distribution and must be at the maximum. Here, we depicted the offset time from the SENEX distribution for all of the histograms, including the ammonium nitrate distribution which is slightly narrower than the SENEX distribution. We revised the wording to clarify this.

Changed "The mean of this distribution is the offset time ($t_{offset}$), and this value for SENEX was 0.35 ms (see Table 1)."

To "Several features of the histogram distribution for ammonium nitrate particles are apparent: there are no particles at large time differences, the histogram has a Gaussian shape, the width at the base of this fit (~ 0.18 ms) is approximately the time available for particles to pass through the chopper slit (~ 0.17 ms), and the mean of the Gaussian fit to the histogram is a non-zero offset time ($t_{offset}$). ...

For the SENEX particles, the mean of its Gaussian distribution fit, $t_{offset}$, is 0.35 ms (see Table 1) and is depicted as the vertical blue line in all four histograms of Figure 1. For ammonium nitrate particles with a 100% collection efficiency, $t_{offset}$ is 0.31 ms, slightly smaller than the SENEX offset time. The similar offset times for the four histograms indicate that the times of the maximum MS signal are later than the estimated time by a consistent amount."

P6L26-28: Why is dva-LS calculated from the laboratory calibration of dva and not calibrated individually?

Unfortunately, we did not collect the appropriate LSSP data from PSLs to do this calibration for SENEX. We added that information to the experimental section along with a verification of the method that we used for SENEX:

In the Experimental section, we added "This velocity can be calibrated with PSLs in a similar manner to $d_{va\text{-}MS}$ (Cross et al., 2007), but unfortunately we did not collect the appropriate LSSP data from PSLs to calibrate $d_{va\text{-}LS}$ during the SENEX project. Hence, we used an alternate method to determine $d_{va\text{-}LS}$ which involved plotting histograms of $t_{LS}$ for laboratory particles that rapidly evaporated in the mass spectrometer (discussed in Sect. 3.1.1)"

In the results section 3.1.1., we added "While this derivation is not as straightforward as a direct calibration of the particle velocity from light scattering for PSL particles, we verified that the method worked well when calculating $d_{va\text{-}LS}$ for PSLs using LSSP data from another velocity-$d_{va\text{-}MS}$ PSL calibration dataset for a subsequent field project. Furthermore, a direct calibration of $d_{va\text{-}LS}$

would be limited to particle sizes above the light-scattering detection limit and cannot include air which is used in the $d_{va-MS}$ calibration."

P7L1: What do you mean with "compared to MS mode"? Reword.

This sentence was reworded with the end of the paragraph.

Changed "The ratio of these LS counts per second in LSSP mode to that in the adjacent MS mode can be used to calculate the light scattering duty cycle due to dead time while saving individual LSSP events. The LSSP light scattering duty cycle was number-concentration dependent with an average value of 35% compared to MS mode. Therefore, each LSSP mode measured single particle mass or number was normalized by the average light scattering duty cycle factor from the preceding and following MS cycles to account for the dead saving time. These internal LS counts when sampling in MS mode are also compared below to number concentrations from independent particle number distribution measurements."

To "Counts per second in LSSP mode are lower than those in the adjacent MS modes due to dead time when particles cannot be counted while individual LSSP events are being saved. This duty cycle factor averaged 35% and decreased with increasing number concentration. Therefore, the average duty cycle factor based on the preceding and following MS cycles was used to calculate the number-based or mass-based mass distributions from LSSP mode (see Sect. 3.2.2)."

P7L11: The IE for nitrate is claimed to be 7E-7 ions/g. This unit is wrong – or the IE of this instrument is orders of magnitude lower compared to other AMSs.

We thank the reviewer for catching this error. The units should be ions per molecule and this was corrected in the revised manuscript.

P8L4 (and many other locations in the text): A clear differentiation between "signal" and "signal intensity" (i.e. the magnitude of a signal) should be made throughout the text. Similarly, wording like "maximum light scattering intensity" (P8L10) should be replaced by something like "maximum light scattering signal intensity" since not the intensity of the scattered light but of the signal from the measurement of the scattered light is meant.

We replaced these terms as suggested throughout the manuscript.

P8L12: "criteria" should be "criterium"

Changed in both places.

P8L14-15: The light scattering intensities do not vary over a significant range because the laser beam is broad, but because the beam intensity has a shape (likely Gaussian shape) across its cross section.

We looked into this and revised the text as follows:

From "However, because the laser beam is designed to be broad to capture the particles, the light scattering intensities varied over a significant range for the same size particles."

To "The maximum scattered light intensities varied over a significant range for the same size particles because the system was not optimized for sizing particles optically. Gaussian fits of individual scattered light pulses from the slowest (75-85 m s$^{-1}$) particles were about 9-12 µs wide and the data from the PMT were recorded every 10 µs, which missed the true maximum of the scattered light pulse for many particles. The scattered light pulse widths and particle velocities indicate that our laser beam was about 0.7-0.9 mm wide, which is much smaller than for the

first LS-AMS study (~2 mm wide, Cross et al., 2007). This is still larger than the calculated particle beam width at the laser position (ranging from 0.13 to 0.59 mm, Huffman et al., 2005), indicating that all of the particles should be passing through the laser beam."

P8L24: "axis" should be "axes". What is a "mass spectrum arrival time"?

Changed "axis" to "axes" and revised phrases with "arrival time" in them so that term is only used for when particles are expected to arrive at the vaporizer.

P8L31-P9L6: The "delayed particles" were explained by particles that bounce off the vaporizer and vaporize at other, colder surfaces. To explain the time delays observed in the measurements the particles have to travel with very low velocities after bouncing off the vaporizer. This explanation is very speculative. Do the authors have any evidence for this? I could imagine if the particles bounce of the vaporizer in such a way that they have to loose most of their energy, they must be strongly deformed during the bounce process which I could imagine results in good contact with the vaporizer. Therefore I would expect that they stick to the vaporizer like ammonium nitrate and other liquid particles do. What do the authors suggest as the process that causes a loss of the vast majority of kinetic energy of the particles and at the same time avoid sticking of the particles to the vaporizer?

We estimated the minimum distances the particles could travel after striking the vaporizer from the range of particle velocities measured with light scattering and the observed delay times. These distances are much larger than the confines of the ionization source region, therefore the delayed particles must have either lost some kinetic energy at some point or, alternatively, bounced several times before vaporization. We do not know which occurred. We added the second possibility (multiple bounces) to the manuscript. If kinetic energy was lost, it may have been converted into plastic deformation and fracturing (Miyakawa et al., 2013), which is now mentioned in the manuscript.

"Given the range of particle velocities measured in the PToF region of 70-150 m s$^{-1}$ for particles between 1000 to 100 nm, respectively, prior to particle impaction on the vaporizer, this represents a loss of as much as 90% of the initial kinetic energy upon a bounce or multiple bounces within the vaporizer. The lost kinetic energy for the particles that bounced may have been converted into plastic deformation and fracturing (Miyakawa et al., 2013). Alternatively, the delays in appearance of the mass spectral signals could have been due to multiple bounces prior to vaporization (Robinson et al., 2017; Hu et al., 2016)."

P9L8-15: This paragraph is very speculative. Do the authors have any evidence for this behavior of the particles?

This paragraph was deleted. It was based on detailed SIMION modeling of particles in the ion source.

P10L1-2: "The single particle mass is near the chemical detection limit … ". For which size of particles is the mass near the LOD?

The MS detection limit was defined for "null" particles as 600 bit ns per particle or about 38 ions per particle (Table 1). Using the IE value for nitrate (7e-7 ions per molecule) plus the molecular weight and density of ammonium nitrate (80 g per mole and 1.72 g per cubic centimeter, respectively), this corresponds to a 200 nm diameter ammonium nitrate particle. However, this is not an absolute size detection limit and Figure 5 indicates the approximate mass per particle detection limit of about 4e-15 g.

We added the following statements to the new Section 3.1.2 after mentioning the MS detection limit of 36 ions for the July 6 flight: "The average LSSP MS detection limit for all flights analyzed is 38 ions (see Table 1). For dry particles composed of pure ammonium nitrate, this MS detection limit corresponds to an individual particle with a minimum size $d_p$ of about 200 nm or $d_{va\text{-}MS}$ of about 275 nm."

P10L1-12: According to this discussion appearance of a particle as a "null" particle is not a proof or hint of particle bounce but for a large fraction of the particles (most of the mass measured with an AMS is typically in the 100-400 nm range) could as well simply be due to low particle signal. This should be discussed when discussing bounce.

We added text to the beginning of this section.

"For the particles that are detected by light scattering, the overall fraction of particles that are detected by the mass spectrometer are affected by particle bounce, refractory composition, and/or ion detection limit. The particles that are detected optically but not detected chemically with the mass spectrometer for any of these reasons are defined as "null" particles (see Figure 2a). Prompt and delayed particles are defined above in Section 3.1.1 by the timing of their mass spectral signals (see Figures 1 and 2c). Our previous studies suggest that particle phase is an important factor determining whether or not particles bounce. Larger particles may also have a tendency to bounce more than smaller particles. Some particles contain refractory components (e.g., soot, sea salt, or dust) that do not vaporize at typical AMS vaporization temperatures. If particles are predominantly composed of these species, they will not produce viable mass spectra. The ion detection limit is relevant for particles that are too small to generate sufficient ions from an individual particle. Furthermore, this detection limit varies with the sensitivity of the specific mass spectrometer and likely between different mass spectrometers. Note that the optical detection limit can vary between instruments too.

As discussed later in Sections 3.2.1 and 3.3.2, the in situ CE from the LS-AMS is defined as the fraction of particles that are detected by the mass spectrometer (all particles detected by light scattering minus the particles without viable mass spectra) divided by the total number of particles detected by light scattering. This definition of CE is equivalent to the fraction of prompt plus delayed particles. The mass-based in situ CE detected by light scattering is defined as the mass of particles from their MS signals divided by the mass of particles from their volume determined by their light scattering size, $d_{va\text{-}LS}$, and an estimated density. Thus, both the fractions of these particles and CE from LSSP data are affected by phase, size, composition, and mass spectrometer sensitivity."

P10L26: Add "for" between "as" and "the SENEX".

Added

P10L30-P11L8: Without more information on how detector sensitivity affects data acquisition and without any interpretation or discussion of the differences in the mass spectra observed at different detector sensitivities, this paragraph does not provide very helpful information.

Since the varying detector sensitivity was not relevant for interpreting the differences between prompt and delayed spectra, we removed the results and discussion of the varying detector sensitivity for the laboratory particles from the manuscript.

P11L15: "The location where the delayed particles impact and vaporize is likely further away from the vaporizer center ... ". Do the authors mean that delayed particles impact near the edge of the vaporizer? In this case a shift towards more delayed particles should be observed when

scanning the particle beam across the vaporizer. Furthermore, not every location further away of the vaporizer is colder than the vaporizer. The hottest location in the ion source is the filament. Be more clear here.

This is related to the general comment from the first referee about vaporization temperature. We revised our discussion on the mass spectral differences to incorporate the possibility of particles vaporizing on several surfaces with uncertain (hotter, cooler, or the same) temperatures or the vapors having more/less wall collisions prior to ionization.

"The consistent differences in the mass spectra between prompt and delayed particles from both ambient and nominally-identical chemical composition laboratory particles suggests that there are different processes at the vaporizer for prompt and delayed particles. The two likely explanations are that the evolved gas from the delayed particles experienced fewer wall collisions before ionization and/or that the delayed particles vaporized from surfaces at different temperatures. The bottom of the conical vaporizer, where prompt particles should vaporize, is a location where the evolved gas molecules likely experience some wall collisions and could decompose before ionization. Gas molecules from particles vaporizing from other surfaces, such as the top of the vaporizer or the ionization chamber, would experience fewer wall collisions where they could decompose. For both ammonium nitrate and ammonium sulfate particles, there is far more fragmentation when the vapors are contained in a capture vaporizer prior to ionization compared to the conical vaporizer at the same temperature (Hu et al., 2016).

Vaporizer temperatures also affect fragmentation. Lower vaporizer temperatures increase the ions at *m/z* 46 relative to *m/z* 30 for ammonium nitrate and at *m/z* 80 and 81 relative to the other ions for ammonium sulfate (Hu et al., 2016). Spectra from most but not all organic compounds have more fragmentation when vaporized at 600 C than at 200 C (Canagaratna et al., 2015). One piece of evidence about vaporization temperature is the width of the vaporization event, which is wider at lower temperatures and is fairly constant above species-dependent temperatures (Drewnick et al., 2015). For the mixed particles that were generated specifically to compare here with the SENEX data, the peaks in the mass spectra as a function of vaporization time were analyzed to check for slower vaporization. A proxy for peak width in the mass spectra (peak area divided by height) was not statistically significant between the prompt and delayed particles, indicating that the vaporization times were roughly comparable. Since a doubling of the peak widths was not observed for the delayed particles here, it is unlikely that they vaporized at temperatures lower than 300 °C (Drewnick et al., 2015). Because the spectra were saved every 32 µs and vaporization event lengths in the AMS are on the order of 30-60 µs and constant for pure ammonium sulfate particles at temperatures between 400 and 800 °C, the data collected in this brief lab study could not be used to validate the possibility of a less drastic change in vaporization temperature. Equivalent vaporization timescales between prompt and delayed particles along with the same issue of insufficient precision were also reported for the laboratory study of SOA particles (Robinson et al., 2017). The increased fragmentation with a hotter vaporizer could be due to changes in the vaporization process itself or to evolved gas molecules hitting a hotter surface, or both.

While the timing indicates that the delayed particles may have lost a large amount of kinetic energy or bounced multiple times before vaporization, it is unclear where the delayed particles are vaporizing in the vaporization/ionization source region. The microporous vaporizer itself could provide the first surface for particles to strike after an initial bounce. Another possible surface is the baffle which reduces stray light from the hot filament from reaching the LS

detector. This baffle has a small hole to transmit particles into the vaporization/ionization source region and roughly forms the side opposite of the vaporizer. Bounced particles could strike the room-temperature baffle if they exit the vaporizer on a trajectory nearly opposite of the initial particle beam. The interior of the ionization chamber is another possible surface. It has ceramic washers and its mounting points are thermally grounded relative to the baffle, filament, and vaporizer. Heat is conducted radiantly between the various hot surfaces (filament and vaporizer) and the ionization chamber, and by conduction through the thin metal of the ionization chamber. The temperature may be different on the sides near and away from the filament. Since we do not have thermocouples on the ionization chamber or the baffle, there are no direct measurements of their temperatures. This makes it difficult to evaluate the relative importance of vaporization temperature compared to fewer wall collisions of the evolved gases from the delayed particles."

P11L10-23: Here the different chemical signal (e.g. less fragmentation) of the delayed particles is explained by vaporization of these particles off cooler surfaces. Where do the authors assume that these surfaces are? In the AMS ionizer the hottest point is the filament, not the vaporizer. Therefore some of the other surfaces might be even hotter than the vaporizer. If particles vaporize from other colder surfaces one would expect longer vaporization times for the individual particles. The authors report that this is not observed and explain this by the large range of vaporization temperatures where the vaporization time is not extended (400-800 ∘C). Which vaporization temperature would be expected to obtain the observed changes in the fragmentation pattern? Is this consistent with this range of vaporization temperatures for quick vaporization?

There are two possible ways to obtain spectra with less fragmentation – either the vaporization temperature is lower or there are less particle (or gas) collisions with the surface. We did not perform experiments looking at the vaporizer temperature and surrounding ionization source region temperature along with precise measurements of the vaporization event time scales to sufficiently answer these questions.

See revised paragraphs above.

P11L25: "mass spectral peak widths" sounds like the width of the peaks in the mass spectra while the duration of particle vaporization is meant. Be clearer.

Changed "… the mass spectral peak widths were analyzed to check for slower vaporization …"

To "…the peaks in the mass spectra as a function of vaporization time were analyzed to check for slower vaporization …"

P12L8-12: The AMS ionization efficiency strongly depends on the exact location of the vaporizer within the ion source. If the vaporizer (i.e. the location from where the evolved vapor originates) is moved, IE drops strongly. Therefore I would expect strongly reduced ion signals (i.e. measured particle mass) from particles vaporizing at a completely different location within the ion source. This would not be in agreement with the observations made in this work and the suggested processes occurring in the ion source. Furthermore, in section 3.1.4 I miss a discussion on the reasons for the missing mass in the delayed particles. Is there a different composition for the delayed and prompt particles? In addition: how certain is it that the null particles are also of the same composition as the measured ones? Is the loss in particle mass different for different instruments or what influence does the large difference in the criterion for MS detection (see Table 1) have?

There are many points raised in this comment.

The first point is about the location of vaporizer affecting the ionization efficiency, the second point is on reasons for missing mass in the delayed particles, and the third point is on differing composition between the prompt and delayed particles.

We analyzed our data in a manner similar to what was presented by Robinson et al. (2017). The analysis is presented in a new figure (Figure 6) and added the following text:

"The delayed particle mass could be lower on average than the prompt particle mass because the evolved gas from the delayed particles was produced in a region where the vapors are not as efficiently ionized by the electron beam. This explanation was recently proposed based on a laboratory study of monodisperse, alpha-pinene secondary organic aerosols (SOA) that had a low number-based CE (0.30), a significant fraction of delayed particles (53% of all LS particles with MS signals), and a clear trend of more ions per particle for prompt particles than for delayed particles (Robinson et al., 2017). Since the SENEX data are from a range of particle sizes, we normalized the total LSSP ion signals to the cube of their $d_{va-LS}$ and plotted the averages as a function of delay time up to 3.5 ms. Figure 6 shows the results from two flights on July 6 and July 3 along with results from ammonium sulfate particles and confirms that delayed particles have a slightly lower total ion signal per unit volume than prompt particles. When considering only delayed particles, all three cases show a negative slope for the total ion signal per unit volume as a function of delay time, but individually the slopes are not significant or marginally statistically significant at the 2 sigma level. The largest change appears to be an average of ~50% lower signal for particles with the longest delay times for the flight on July 3. Thus, the efficiency of producing ions from the delayed particles is not always consistently lower than from prompt particles and more experiments are needed to investigate this possible explanation for a reduced mass. It is not clear from the field data why there was a minimal reduction in ion signal; inefficient ionization of the vapors that evolved from delayed particles cannot be precluded."

We also note to the reviewer that moving the vaporizer within the ion source affects the electric fields within the ion source. So particles vaporizing from a different spot on a fixed vaporizer may not be equivalent to moving the vaporizer.

We also added "The roughly 20% reduction in mass of the delayed particles compared to prompt particles could be partly due to some refractory material in the delayed particles that is not measured by the AMS. The bulk mass fraction of refractory black carbon (rBC) was on average ~1% of the measured aerosol mass for these flights (Warneke et al., 2016), so rBC does not account for all of the reduced mass."

The fourth point is about the composition of null particles compared to the measured ones. Since this section is on the prompt and delayed particles with mass spectra, it does not discuss null particles. We added or revised text in other places.

In the section on the fraction of prompt, delayed, and null particles: "Some particles contain refractory components (e.g., soot, sea salt, or dust) that do not vaporize at typical AMS vaporization temperatures. If particles are predominantly composed of these species, they will not produce viable mass spectra."

In the new section 3.3.3, we discussed additional considerations of the LSSP-based CEs: "CE is traditionally defined based on mass comparisons. There is an assumption in applying either a calculated or an in situ CE to the measured mass loadings from MS mode that the chemical composition of the undetected mass is the same as the detected mass, which is likely true

during sampling of air masses with mostly mixed secondary aerosol particles. In general for SENEX, the number- and mass-based CE from the LSSP data shown in Figure 9 and Figure S3 are comparable within experimental uncertainties for the particles sampled. In Figure 8a, the integrated mass from the PToF mass distribution using the average CE from the parameterization of 0.6 for the flight on July 6 (dashed curve) is also within the combined experimental uncertainties of the integrated mass from the UHSAS mass distribution (solid black curve). The flight-averaged number-based CE was 0.58. Hence, the SENEX field data did not show any large discrepancies between the number- and mass-based CEs when averaged for the entire flight.

It is also assumed that particles detected optically are representative of all particles sampled by the AMS and have the same chemical composition as the particles that are too small to be detected by LSSP mode. In air masses where newly formed and growing (Aitken mode) particles are present, this assumption is not necessarily valid. The number- and mass-based CEs may be different if there are significant differences in the number-based CE as a function of size, as briefly described by Huffman et al. (2005). In power plant plumes that were sampled by the PToF mode during SENEX, we sometimes observed smaller acidic sulfate particles with larger mixed-composition particles. Because these power plant plumes were transected quickly by the aircraft, the small particles were not sampled consistently with LSSP mode, and so the effect of a varying composition with size could not be evaluated here."

The last two points are about how different instruments and criteria for MS detection may affect the results and are discussed in the paragraph at the end of section 3.1.5 (old Section 3.1.4).

"Although the two studies used different definitions for prompt and delayed particles, changing this definition does not alter the measured average chemical ion signals."

We also stated and added details: "The reasons for much lower delayed particle mass compared to prompt ones remain unclear, but differences in the ambient aerosols measured may contribute to the different delayed particle mass. The Mexico City study was conducted on the ground near the metropolitan area where ~10% of PM2.5 mass was black carbon (Retama et al., 2015), whereas the black carbon was on average ~1% of the mass for SENEX (Warneke et al., 2016). As the null fraction in Cross et al. (2009) was not higher than in this study and the number-based CE was 0.49, the much lower mass from the delayed particle chemical ion signals observed by Cross et al. (2009) could be due to particles containing more refractory material during the Mexico City study than during SENEX."

P12L14-27: With about 10% black carbon content in the fine aerosol observed in Mexico City it is hard to believe that the much lower measured delayed particle mass (less than half of the prompt ones) observed in this study is caused by larger black carbon content during that measurement. What was the BC content during this study?

Changed "The Mexico City study was conducted on the ground near the metropolitan area where ~10% of PM2.5 mass was black carbon (Retama et al., 2015)."

To "The Mexico City study was conducted on the ground near the metropolitan area where ~10% of PM2.5 mass was black carbon (Retama et al., 2015), whereas the black carbon was on average ~1% of the mass for SENEX (Warneke et al., 2016)."

We also added to Section 3.1.5 on the derived LSSP mass: "The roughly 20% reduction in mass of the delayed particles compared to prompt particles could be partly due to some refractory

material in the delayed particles that is not measured by the AMS. The bulk mass fraction of refractory black carbon (rBC) was on average ~1% of the measured aerosol mass for these flights (Warneke et al., 2016), so rBC does not account for all of the reduced mass."

We also added this paragraph about a flight with biomass burning influence to the section on the LSSP-based CEs (3.3.2):

"The default CE parameterization value of 0.5 may be too high during some parts of this flight and large parts of other flights (Figure S3). This was also observed with other field data using a mass-based comparison to evaluated CE (Middlebrook et al., 2012) and may indicate that the default CE of 0.5 is slightly high in the parameterization. One flight in particular (July 3) had the highest null fractions (Figure 3) and corresponding lower CEs for most of the flight compared to other flights. This flight had an overall higher mass loading of refractory black carbon (rBC) from biomass burning in the sampled air (0.36 µg sm$^{-3}$, whereas the average for all of the flights analyzed here was 0.14 µg sm$^{-3}$). If an individual particle is mostly rBC, there may not be enough ion signal from the non-refractory components to detect it with the mass spectrometer. In Toronto, a higher null fraction was measured when urban, rBC-containing particles were evaporated with a vaporizer instead of an infrared laser (Lee et al., 2015). It is also possible that the collection efficiency is lower for the organic fraction of biomass burning aerosols. The effect of biomass burning particles on the measured CE with the LS-AMS needs further investigation."

P12L28: The title of section 3.2 is not very well chosen. Most of the findings presented here are not observations using the AMS LSSP mode but are still characterization of the LSSP and PTOF measurements.

Changed "Observations using the AMS LSSP mode"

To "Characterizing LSSP measurements by comparisons with MS, PToF, and UHSAS data"

P13L16: Since here general ways to generate mass distributions from AMS instruments with LS module are described it is not appropriate to use an effective particle density of 1.55 g/cm3. This is valid for this data set only.

We added to the Experimental section: "To calculate single particle mass from the LSSP size, $d_{va-LS}$, the effective particle density is estimated to be 1.55 g cm$^{-3}$ from a calculated weighted average density of the SENEX AMS dataset that was composed of 50-70% organic material (estimated density of 1.25 g cm$^{-3}$ from (Cross et al., 2007;Kiendler-Scharr et al., 2009;Zelenyuk et al., 2008)) and 30-50% inorganic material (primarily dry ammonium sulfate with a density of 1.75 g cm$^{-3}$ from (Perry and Green, 1997))."

P13L25: Define "UHSAS".

Defined

P14L8: The ratio between the AMS PTOF and the other distributions for large particle sizes is not constantly four as claimed here, but varies strongly.

Deleted "a factor of four"

P14L12: The delayed particles do not only create a bias towards the larger size end of the mass distribution, but as shown in Figure 1 can also leak into the next chopper cycle where they contribute to the lower size end of the distribution and increase the measured background, i.e. lower the effectively measured distribution.

This comment indicated that there might be a problem with Figure 1 and we found that the ammonium sulfate histogram included null particles that did not have viable mass spectra which increased the noise level of the histogram. Hence, we revised it to only include prompt and delayed particles.

We added the following paragraph to the mass distribution comparison section.

"On the small size of the mass distributions sizes ($d_{va-LS}$ ~ 100 to 300 nm), there is additional mass measured in the PToF mode that does not appear in the LSSP data (Figure 8a). This is not a bias in the PToF data at small sizes because it is also observed in the UHSAS data. These particles are too small for the scattered light signal to consistently trigger saving data in LSSP mode (see Figure 2) and it is uncommon for particles to appear in the early part of the 8.3 ms long chopper cycle, even for particles such as pure ammonium sulfate that have a high tendency to bounce (Figure 1). Furthermore, the PToF data are acquired by aggregating the bulk mass spectral signals over the sampling period rather than by aggregating single particle mass spectral signals or individual particle counts from the laser. Thus, the PToF mode measures the mass from small particles which are not large enough to efficiently scatter light in the LSSP mode or generate enough ions for a clear signal from a single particle."

P14L24ff: In section 3.3.2 I miss a discussion on the potential influence of the mixture state of the particles and size-dependent composition on determined CE factors. While the traditional CE factor is defined on a mass basis, using the LS a number-based CE factor (which is partially transformed into a mass-based CE) is used. This relies on the assumption that all particles (the bounced and the detected ones) have the same composition – even for particles smaller than the lower particle size cut-off of the LSP. Externally mixed aerosol should result in different fractions of bouncing particles (e.g. pure ammonium nitrate and other liquid particles would not bounce while the majority of pure ammonium sulfate particles would bounce). This should at least be discussed.

To the new section on LSSP-based CEs, we added these paragraphs (some with text from the old section 3.3.2):

"3.3.3 Additional considerations for the LSSP-based collection efficiencies

CE is traditionally defined based on mass comparisons. There is an assumption in applying either a calculated or an in situ CE to the measured mass loadings from MS mode that the chemical composition of the undetected mass is the same as the detected mass, which is likely true during sampling of air masses with mostly mixed secondary aerosol particles. In general for SENEX, the number- and mass-based CE from the LSSP data shown in Figure 9 and Figure S3 are comparable within experimental uncertainties for the particles sampled. In Figure 8a, the integrated mass from the PToF mass distribution using the average CE from the parameterization of 0.6 for the flight on July 6 (dashed curve) is also within the combined experimental uncertainties of the integrated mass from the UHSAS mass distribution (solid black curve). The flight-averaged number-based CE was 0.58. Hence, the SENEX field data did not show any large discrepancies between the number- and mass-based CEs when averaged for the entire flight.

It is also assumed that particles detected optically are representative of all particles sampled by the AMS and have the same chemical composition as the particles that are too small to be detected by LSSP mode. In air masses where newly formed and growing (Aitken mode) particles are present, this assumption is not necessarily valid. The number- and mass-based CEs may be different if there are significant differences in the number-based CE as a function of size, as

briefly described by Huffman et al. (2005). In power plant plumes that were sampled by the PToF mode during SENEX, we sometimes observed smaller acidic sulfate particles with larger mixed-composition particles. Because these power plant plumes were transected quickly by the aircraft, the small particles were not sampled consistently with LSSP mode, and so the effect of a varying composition with size could not be evaluated here.

For comparison, mass-based and number-based CEs have been reported from other studies. The Bakersfield study described a discrepancy between the average number- and mass-based CEs, where the number based value was ~0.5 and the mass-based value from ensemble measurements was 0.8 (Liu et al., 2013). The authors proposed that a mismatch of vaporization and data acquisition time scales reduced the detected chemical ion signals from single particles compared to the ensemble measurements; yet this discrepancy was not resolved. The in situ CE from LSSP mode measurements were also determined and compared with AMS ensemble and independent measurements for the Mexico City study (Cross et al., 2009). The number- and mass-based CE was on average ~0.5 for the 75-h sampling period of LSSP data and showed some size dependence with the smallest particles having a high CE (low null fraction and higher prompt fraction) than the larger particles.

While the aircraft data reported here show a wide range of CE due to air mass variations, such variability in the LSSP mode CE has not been reported previously. Variations in ensemble CE were not reported in the previous ambient studies but could have been possible due to the diurnal variability in the ambient measurements from Mexico City, where in the morning there were small particles composed of predominantly hydrocarbon-like organic aerosol (HOA) which appeared to have a higher CE (Cross et al., 2009). Hence, LSSP data could also show that mixing state plays a role in the measured CE."

P15L10: In this context not the PTOF but the MS mode is relevant.

It applies to both, so we changed "… PToF mode …"

To "… MS or PToF mode …"

P15L6-24: In Figure 8 the mass and number-based LSSP CE factors sometimes differ by much more than their given uncertainty. What could be the reason for this? Is the assumed uncertainty too small?

We thank the referee for noticing this. The mass-based CE from the LSSP data has additional uncertainties from the uncertainty in particle size, so the statistical uncertainty in that value is too small. We added the following text to the section on comparing CEs:

"In addition to this statistical variability, the mass-based CE from the LSSP data has as much as 27% uncertainty from the measured particle volume from $d_{\text{va-LS}}$ due to ±0.17 ms uncertainty in $t_{\text{LS}}$. The error bars for the mass-based LSSP CE (yellow points in Figures 9 and S3) were not included in the figures for clarity."

We also added the 9% uncertainty in $d_{\text{va-LS}}$ due to 0.17 ms uncertainty in $t_{\text{LS}}$.to the first section on sizing results (3.1.1).

"The true particle velocity in the AMS, however, does not include $t_{\text{offset}}$ and is obtained simply as $L_{\text{LS}}/t_{\text{LS}}$, with about ±0.17 ms uncertainty for our chopper duty cycle (2%) and rotational speed (~120 Hz) in the time that the particles pass through the chopper slit (Cross et al., 2007). This timing uncertainty results in a size uncertainty of 9% for both $d_{\text{va-MS}}$ and $d_{\text{va-LS}}$."

P16L7: The authors claim that "particle bouncing physics" was investigated. What kind of particle bouncing physics was measured in this work?

We deleted that phrase.

P16L14-16: Please provide clear information on the limitations of the method.

At the end of this paragraph we added:

"As shown in Figure 8a, LSSP mode clearly cannot measure a large fraction of the ambient aerosol mass distribution. Furthermore, the duty cycle is quite high for LSSP mode, reducing the number of particles recorded and needing information from the adjacent MS modes to normalize LSSP data. The MS detection limit is also important for the new event-trigger AMS mode, which saves individual particle spectra from a single chopper cycle when the mass spectral signals of selected ions are above a threshold."

Figure 1/2/3/5/7/8: Present the units as recommended by IUPAP etc.: e.g. "Maximum Scattered Light Signal (V)" should be "Maximum Scattered Light Signal / V".

We did a check of 8 recent AMT articles and none of them used this notation on the figures for the units. We will revise the figure labels if the AMT editor requests it.

Figure 1: For the ammonium sulfate particles almost no time dependence of the frequency of delayed particles occurs. How can this be explained? Is this potentially an artefact? What is the "mixed composition"?

We found an error in the ammonium sulfate histogram (null particles were previously included) and replaced it (so it was an artifact). We added the mixed composition species to the legend.

Figure 2/4: The labels "a)" and "b)" are very hard to see.

We made the labels larger.

Figure 2, caption: make clear that not the scattered light signal but the intensity of the scattered light signal is shown. It should read "above 0.04 V (red points)", i.e. add "V".

Added "V" to the caption and revised wording from "maximum scattered light signal" to "maximum scattered light intensity."

Figure 8 (and Figure S2): Since the altitude at which the measurements were taken is never mentioned or discussed in the manuscript it could be removed from the graph to make it a bit less crowded.

Changed. Also, we added a new Figure 6, so these are now Figure 9 and Figure S3.

Supplement P1L19: I suggest using the term "dead time" instead of "extra time".

Changed.

P1L27: "multiply by chopper" should read "multiplied by the chopper".

Changed.

P2, Table S1: Some of the coefficients change by almost a factor of 2. How large is the uncertainty of determination of these coefficients. Is it expected that they change over the course of a flight? Would not an average number be the better choice?

Because the aircraft instrument has no power between flights, is sometimes off for a few days between flights, and is powered three hours before takeoff, these coefficients can vary

significantly. Spectra from the background and filtered air can vary during a flight as the instrument is pumped down. Since the coefficients listed in Table S1 were only used to process the bulk MS data and not for the LSSP data (the fragmentation pattern for air is not included in the matrix for LSSP processing), we removed it from the manuscript and revised our discussion of the data processing.

[a]Now at: Universities Space Research Association, Columbia, MD 21046, USA and NASA Goddard Space Flight Center, Atmospheric Chemistry and Dynamic Laboratory, Greenbelt, MD 20771, USA
[b]Now at: Leibniz Institute for Tropospheric Research, Department of Physics, Leipzig, 04318, Germany

*Correspondence to*: Ann M. Middlebrook. (ann.m.middlebrook@noaa.gov)

**Abstract.** A light scattering module was coupled to an airborne, compact time-of-flight aerosol mass spectrometer (LS- AMS) to investigate collection efficiency (CE) while obtaining non-refractory aerosol chemical composition measurements during the Southeast Nexus (SENEX) campaign. In this instrument, particles ~~typically larger than ~ 250 nm in vacuum aerodynamic diameter~~ scatter light from an internal laser beam and trigger saving individual particle mass spectra. Nearly all of the single particle data with mass spectra that were triggered by scattered light signals were from particles larger than ~ 280 nm in vacuum aerodynamic diameter. Over 33,000 particles are characterized as either prompt (27%), delayed (15%), or null (58%), according to the  time and intensity of their total mass spectral signals. The particle mass from single particle spectra is proportional to that derived from the light scattering diameter ($d_{va-LS}$) but not to that from the particle time-of-flight (PToF) diameter ($d_{va-MS}$) from the time of the maximum mass spectral signal.  The total mass spectral signal from delayed particles was about 80% of that from prompt ones for the same $d_{va-LS}$ . Both field and laboratory data indicate that the relative intensities of various ions in the prompt spectra show more fragmentation compared to the delayed spectra. The particles with a delayed mass spectral signal likely bounced off the vaporizer and vaporized later on another surface within the confines of the ionization source. Because delayed particles are detected  by the mass spectrometer later than expected from their $d_{va-LS}$ size, they can affect the interpretation of particle size (PToF) mass distributions, especially at  larger sizes. The collection efficiency, measured by the average number or mass fractions of particles optically detected that had measureable mass spectra, varied significantly (0.2-0.9) in different air masses.  The measured CE generally agreed with

the CE parameterization based on ambient chemical composition, including for acidic particles that had a higher CE as expected from previous studies.The measured CE agreed well with a previous parameterization when CE > 0.5 for acidic particles, but was sometimes lower than the minimum parameterized CE of 0.5.

**1 Introduction**

5  Aerosol size, chemical composition and mass loading are important parameters used to estimate the impact of aerosols on direct (aerosol-radiation) ly scattering sunlight or being cloud condensation nuclei toand indirect (aerosol-cloud)ly affect radiation balance and climate effects (e.g. Ramanathan et al., 2001). The spatial and temporal distribution of ambient aerosols is highly inhomogeneous, owing to different sources, meteorological conditions, atmospheric processes, and their relatively short atmospheric lifetimes compared to greenhouse gases. The Aerodyne aerosol mass spectrometer (AMS) is a fast time

10  response instrument capable of quantifying size-resolved non-refractory aerosol chemical composition (e.g. Jayne et al., 2000; Jimenez et al., 2003; Drewnick et al., 2005; Canagaratna et al., 2007) and has been widely used to measure the real time aerosol ensemble organic, sulfate, nitrate, ammonium, and chloride (non-sea salt) mass loadings globally (e.g. Zhang et al., 2007a; Jimenez et al., 2009). Evaluation of aerosol mass loading accuracy measured by the AMS is important to estimate the impact of aerosols on climate, biogeochemical health, and aerosol formation processes such as aerosol hygroscopicity (Levin et al.,

15  2014; Brock et al., 2016) and aerosol acidity (Hennigan et al., 2015; Zhang et al., 2007b).

The basic principle of the AMS method is to focus ambient aerosols with an aerodynamic lens onto a hot vaporizer and analyze the evolved gases with an electron-impact ionization mass spectrometer. Not all particles introduced to the inlet are vaporized andthermally desorbed and the resulting gas-phase molecules ionized. The net overall transmission and detection efficiency is

20  called collection efficiency (CE) and expressed as the product of aerodynamic lens transmission efficiency for spherical particles, the loss of transmission due to particle beam broadening, and the efficiency of detecting a particle that vaporizes on impaction (Huffman et al., 2005). The particle transmission losses in the AMS lens can be determined by the lens particle transmission curve (e.g., Bahreini et al., 2008). Beam width probe experiments (Huffman et al., 2005) found that CE less than 100% is not due to particle beam broadening but is likely due to particles bouncing off the vaporizer. Therefore, with well-

25  characterized lens transmission efficiency, the dominant uncertainty in CE is likely to be the particles bouncing on the vaporizer. A varying collection efficiency (CE) of particles by the AMS potentially introduces large uncertainty in AMS measurements. CE is quantified by the ratio of the mass (or number) of particles detected by the AMS to that of particles introduced into the inlet (Matthew et al., 2008). It rangeds from 0.3-1 to one with the highest frequency around 0.45 in our ambient measurements (Middlebrook et al., 2012). This indicates that the varying CE in some cases and therefore may induce

30  an uncertainty as large as a factor of 3 in the aerosol mass measured by AMS. CE less than 100% in AMS measurements was previously demonstrated by comparing aerosol mass loadings measured by the AMS with that measured by other instruments such as the pParticle-iInto-lLiquid sSampler combined with an iIon cChromatography analyzer (PiILS-IC) or an optical

particle counter (ultra-high sensitivity aerosol spectrometer or UHSAS) (e.g. Takegawa et al., 2005; Middlebrook et al., 2012).  Laboratory and field studies showed that CE values depend on aerosol chemical composition and relative humidity (Matthew et al., 2008; Middlebrook et al., 2012). Based on this, a parameterization

5 for the composition-dependent CE was developed (Middlebrook et al., 2012) and is now  commonly applied to ambient AMS measurements. CE obtained by comparison of AMS measurements with other instruments like the PiLS-IC and UHSAS may be affected by the measurement uncertainties of other instruments and lens transmission efficiency. In situ measurements of CE can directly quantify the detection efficiency of particles that impact the vaporizer. In situ CE measurements and evaluation of the CE parameterization are therefore important to reduce the uncertainty in the AMS measurements.

A light scattering (LS) module has been developed to integrate into AMS instruments (LS-AMS) to detect single particles before they impact on the vaporizer (Cross et al., 2007). Th LS-AMS determines  the in situ CE  by comparing the number (or mass) of particles  detected by the mass spectrometer signals to the total number (or mass) of particles  detected by the scattered light signals.

15 Using the LS-AMS instrument, the mean values of CE for ambient particles at three ground sites near Mexico City (Cross et al., 2009), Bakersfield, CA (Liu et al., 2013),  downtown Toronto (Lee et al., 2015), and in the eastern Canadian forest (Slowik et al., 2010) w  0.49, 0.52  0.37, and 0.6, respectively.  CE for the Mexico City ground site varied only about ±10% over the full sampling period (Cross et al., 2009), which may be due to relatively constant ambient aerosol chemical composition . Airborne studies of air masses with widely different chemical composition

20 provide an opportunity for investigating the capability of LS-AMS  to measure CE variations.

Beside the ability to determine in situ CE, the LS-AMS has also been used to derive particle density by comparing the optical size with the vacuum aerodynamic size (Cross et al., 2007), distinguish single particle chemical composition types (Cross et al., 2007; Liu et al., 2013; Freutel et al., 2013), particle internal and external mixing properties (Robinson et al., 2013), and

25 validate the interpretation of AMS factors from a positive matrix factorization using cluster analysis of the LS module data (Lee et al., 2015). As this work aims to use LS-AMS to investigate AMS measurement uncertainties, analysis regarding the above perspectives is not included.

This study provides the first airborne single particle measurements from an LS-AMS instrument. These measurements

30 were performed onboard the NOAA WP-3D aircraft sampling various air masses over the continental United States during the Southeast Nexus of Air Quality and Climate (SENEX) campaign in May and June 2013 (Warneke et al., 2016).  This study focuses on using single particle data to investigate airborne AMS measurement uncertainties.  CE was measured by LS-AMS during

this field study and compared to the CE parameterization based on the aerosol chemical composition and relative humidity (Middlebrook et al., 2012). The single particle data are also used to examine particles that  on the vaporizer and the impact of inefficiently collected particles on the chemical ion signals and the  AMS particle time-of-flight data.

5 **2 Experimental**

[revised manuscript text omitted]

Using LSSP mode, t are three  ways to measure  particle size : 1) vacuum aerodynamic diameter based on the particle velocit  from the  mass spectral  information  (a.k.a. the traditional AMS vacuum aerodynamic diameter or $d_{va\text{-}MS}$), 2) vacuum aerodynamic diameter based on the particle velocit  from the light scattering information  (light scattering vacuum aerodynamic diameter or $d_{va\text{-}LS}$), and 3) optical diameter (or $d_o$,) from the scattered--light intensity of individual particles. ~~The third measurement of particle size, the optical diameter or $d_o$, can be obtained from a calibration of scattered light intensity and compared to $d_{va}$ (Cross et al., 2007). As we show in Section 3.1, $d_o$ is not the optimal measurement for size in this system. Also, the two measurements of vacuum aerodynamic diameter ($d_{va}$ and $d_{va\text{-}LS}$) provide additional information on how particles are detected by the AMS.~~

The vacuum aerodynamic diameter  is  defined for the AMS by Eq. (44) from DeCarlo et al. (2004):

$$d_{va} = d_m \times \frac{\rho_{eff}}{\rho_0} \tag{1}$$

where $d_m$ is the electrical mobility diameter, $\rho_{eff}$ is the effective particle density (in g cm$^{-3}$), and $\rho_0$ is the standard density (= 1 g cm$^{-3}$). The vacuum aerodynamic diameter is related to the particle velocity measured by the AMS (Jayne et al., 2000; DeCarlo et al., 2004), and the additional subscripts of "MS" and "LS" are used here to indicate the method of calculating particle velocity. For $d_{va\text{-}MS}$, the particle velocity is determined from  the time of the maximum intensity in the mass spectral signal (also known as PToF time or  divided by  the distance between the chopper wheel and the vaporizer ($L_{vp}$) and this velocity was calibrated using polystyrene latex spheres (PSLs) of known sizes. In this manner, mass distributions as a function of $d_{va\text{-}MS}$ or $d_m$ can be determined from AMS instruments in PToF mode without the LS module (Jayne et al., 2000).

~~the measured PToF time ($t_{ms}$) includes an additional brief amount of time needed for particles to vaporize, the neutral molecules to move from the vaporizer to the electron beam, ions to move from the ion source to the orthogonal extraction region, and different signal paths in the data acquisition system. For prompt particles and not including the uncertainty in the time for particles to pass through the chopper, the slowest of these processes and largest uncertainty in PToF sizing is due to vaporization (D. Day, personal communication, November 16, 2015).~~

For $d_{va\text{-}LS}$ the  particle velocity  is calculated from the time of the maximum intensity of the scattered light signal ($t_{LS}$)

10 divided by the distance between the chopper wheel and the laser beam ($L_{LS}$). This velocity can be calibrated with PSLs in a similar manner to $d_{va\text{-}MS}$ (Cross et al., 2007), but unfortunately we did not collect the appropriate LSSP data from PSLs to calibrate $d_{va\text{-}LS}$ during the SENEX project. Hence, we used an alternate method to determine $d_{va\text{-}LS}$ which involved plotting histograms of $t_{LS}$ for laboratory particles that rapidly evaporated in the mass spectrometer (discussed in Sect. 3.1.1). The third type of measurement of particle size, the optical diameter or $d_o$, can be obtained from a calibration of the scattered -light

15 intensity and compared to $d_{va\text{-}MS}$ to obtain information about the particle density using Eq. (1) (Cross et al., 2007). ~~In this case, $t_{LS}$ does not include additional time for detecting mass spectral signals, so the calibration coefficients will vary slightly from the traditional $d_{va}$ calibration. Alternatively, $d_{va\text{-}LS}$ can be determined using the same calibration values as $d_{va}$ after accounting for the additional time. Assuming that the velocity of a particle is constant in the vacuum chamber, the estimated arrival time at the vaporizer, $t_{est}$, is calculated from the time the particle passes through the laser beam as:~~

20

~~where $t_{LS}$ and $L_{LS}$ are defined above and $L_{vp}$ is the distance between the chopper wheel and the vaporizer. For prompt particles (e.g., ammonium nitrate), a histogram of the time differences between the maximum mass spectrum signal time ($t_{ms}$) and the estimated arrival time ($t_{est}$) has a Gaussian distribution (Figure 1). The mean of this distribution is the offset time ($t_{offset}$), and this value for SENEX was 0.35 ms (see Table 1). Twice the width of the Gaussian distribution for prompt particles is~~

25 ~~approximately the time available for particles to pass through the chopper slit; here it is ~ 0.12 ms. Particles with $t_{ms} > (t_{est} + t_{offset}) + 3 \times$ the Gaussian width are defined as "delayed" (see Table 1 and Sections 3.1.1 and 3.1.2 below) and are represented by particles on the right hand side of the cyan line at 0.53 ms in Figure 1. $d_{va\text{-}LS}$ is then obtained by the $d_{va}$ laboratory calibration values and the derived particle velocity accounting for the offset time as:~~

The laser is used to count particles in all three sampling modes every time the intensity of the scattered light signal is above the specified threshold, except that the small fraction of coincident particles in LSSP mode are counted as only one particle.

scattering threshold set in the data acquisition software in LSSP mode, but also counts the number of all sampled particle above this threshold in MS (both sampling and background) and PToF modes. The ratio of these LS Ccounts per second in LSSP mode are lower than those to that in the adjacent MS modes can be used to calculate the light scattering duty cycle due to dead time when particles cannot be counted while saving individual LSSP events are being saved. This duty cycle factor averagedThe LSSP light scattering duty cycle was number concentration dependent with an average value of 35% compared to MS mode and decreased with increasing number concentration. Therefore, the average duty cycle factor based on the preceding and following MS cycles was used to calculate the number-based or mass-based mass distributions from LSSP mode (see Sect. 3.2.2)each LSSP mode measured single particle mass or number was normalized by the average light scattering duty cycle factor from the preceding and following MS cycles to account for the dead saving time. These internal LS counts when sampling in MS mode are also compared below to number concentrations from independent particle number distribution measurements.

The ionization efficiency of the instrument was calibrated with pure, dry ammonium nitrate particles several times before, during, and after the field project. The Igor ToF AMS calibration analysis software version 3.1.5 was used with the PToF-calibrated size ($d_{va\text{-}MS}$) to calculate the nitrate ionization efficiency (IE). When all of the calibration data were combined, the ionization efficiency was linearly proportional over a wide range of detector sensitivities to the airbeam (AB) signal at $m/z$ 28 (signal from air) with an intercept slight offset: IE = $1.29\text{x}10^{-7} + 1.24\text{x}10^{-12}\times$AB. For a typical AB value of $4.5\text{x}10^{5}$ Hz, the IE for nitrate was about $7\text{x}10^{-7}$ ions molec$^{-1}$/g. The unit mass resolution MS and PToF data were analyzed using the Igor ToF AMS analysis toolkit (a.k.a. Squirrel) version 1.52L. For the MS and PToF data, Five coefficients from the standard default AMS fragmentation table described by Allan et al. (2004) wasere adjusted slightly for each flight for the measured fragmentation pattern from water and measured contributions of various species to the filtered air signals (see Table S1 for values). The current default values for the relative ionization efficiency (RIE) default values forof sulfate, organic and non-refractory chloride (of 1.2, 1.4 and 1.3, respectively) were used except thatbut the values of nitrate and ammonium were changed to 1.05 and 3.9.

The LSSP data were processed using the Igor LS analysis toolkit (a.k.a. Sparrow) version 1.04F, with a modification to account for the longer particle flight chamber. The default list of $m/z$ values in Sparrow was used to generate the mass spectra from individual particles. LSSP mass spectra are then processed using the default fragmentation table matrix and the above relative ionization efficiencies, except the contribution from air is removed from the matrix since only the mass spectral signals from the particle are selected. The backgrounds from the longest and/or shortest times at each $m/z$ in the LSSP spectra from a single chopper cycle are subtracted from the particle region in the same way the background is subtracted for standard PToF data. To calculate single particle mass from the LSSP size, $d_{va\text{-}LS}$, the effective particle density is estimated to be 1.55 g cm$^{-3}$ from a calculated weighted average density of the SENEX AMS dataset that was composed of 50-70% organic material (estimated

density of 1.25 g cm$^{-3}$ from Cross et al., 2007; Kiendler-Scharr et al., 2009; Zelenyuk et al., 2008) and 30-50% inorganic material (primarily dry ammonium sulfate with a density of 1.75 g cm$^{-3}$ from Perry and Green, 1997).

The LS-ToF-AMS was onboard on NOAA WP-3 aircraft and flew over the continental US to sample a variety of air masses during the Southeast Nexus (SENEX) campaign from June to July 2013, as part of a large collaboration study Southeast Atmospheric Study (SAS). Detailed information of the field campaign is provided in Warneke et al. (2016) and http://www.esrl.noaa.gov/csd/projects/senex/. Over the course of the project, there were 17 research flights. Besides the AMS measurements presented here, dry particle number distributions from ~ 0.07 to 1.0 µm were measured withby an ultra high sensitivity aerosol size spectrometer (UHSAS) (Brock et al., 2016) onboard the same aircraft and are used to derive the particle mass distributions for comparison in this study. For comparisons between the AMS and UHSAS instruments, we account for the AMS lens transmission efficiency (Liu et al., 2007), which was measured for our lens to be similar to the predicted transmission (Bahreini et al., 2008). Since the AMS sensitivity was poor at the beginning of the field project and UHSAS data were not available for the last flight, the data reported here are from 7 flights from June 26 until July 8 (Flights 10-16).

**3 Results and discussion**

**3.1 LSSP signal processing and categorization of particlesParticle bounce at the vaporizer**

**3.1.1 LSSP mode particle size measurements and indication of particle bouncing**

**3.1.1 Timing of the LSSP signals, particle $d_{\text{va-MS}}$ and $d_{\text{va-LS}}$ sizing, and categories based on time**

Two of the ways to determine particle size in the LS-AMS involve measuring the particle velocity using the particle time-of-flight distances between the chopper and the vaporizer ($L_{\text{vp}}$) or between the chopper and the laser ($L_{\text{LS}}$) and the corresponding time of the maximum MS signal ($t_{\text{ms}}$) or the maximum LS signal ($t_{\text{LS}}$) (Cross et al., 2007). Particles are accelerated into the vacuum chamber at the lens exit based on their size (smaller particles are accelerated to faster speeds than larger particles) and particle velocities are assumed to be constant by the time they reach the chopper wheel. Their velocities based on the MS signals ($= L_{\text{vp}}/t_{\text{ms}}$) were calibrated for $d_{\text{va-MS}}$ size using PSL particles. To calculate $d_{\text{va-MS}}$ from individual particles, they must have viable chemical ion signals (mass spectra or MS). Yet, they do not need to have viable MS signals in order to measure $d_{\text{va-LS}}$ based on the timing of their LS signals (as shown below).

The time the particle passes through the laser beam ($t_{\text{LS}}$) can be used to estimate the arrival time at the vaporizer ($t_{\text{est}}$) as follows:

$$t_{est} = t_{LS} \times \frac{L_{vp}}{L_{LS}} \tag{2}$$

where $t_{LS}$, $L_{vp}$, and $L_{LS}$ are defined as above. For our chamber dimensions, $t_{est} = 1.49 \times t_{LS}$. Times of the maximum LS and MS signals for an example of LSSP data are indicated in Figure S2 along with $t_{est}$, which is earlier than the maximum MS signal time ($t_{ms}$).

5 Figure 1 displays histograms of the time differences between the time of the maximum MS signal ($t_{ms}$) and the estimated arrival time ($t_{est}$) based on Eq. (2) for polydispersed, laboratory-generated and SENEX particles. A similar histogram was previously generated, described, and shown in the SI by Lee et al. (2015). Ammonium nitrate particles are known to have a collection efficiency around 100% whereas ammonium sulfate particles have a collection efficiency around 25% (Matthew et al., 2008) and their histograms of the time differences are not the same. Several features of the histogram distribution for ammonium

10 nitrate particles are apparent: there are no particles at large time differences, the histogram has a Gaussian shape, the width at the base of this fit (~ 0.18 ms) is approximately the time available for particles to pass through the chopper slit (~ 0.17 ms), and the mean of the Gaussian fit to the histogram is a non-zero offset time ($t_{offset}$). The ammonium sulfate histogram has a maximum near the mean of the ammonium nitrate distribution and has a significant tail of particles that were detected at later times. The two other histograms (mixed composition and SENEX particles) are not as narrow as the ammonium nitrate

15 histogram and have fewer particles than the ammonium sulfate histogram at slower times.

For the SENEX particles, the mean of its Gaussian distribution fit, $t_{offset}$, is 0.35 ms (see Table 1) and is depicted as the vertical blue line in all four histograms of Figure 1. For ammonium nitrate particles with a 100% collection efficiency, $t_{offset}$ is 0.31 ms, slightly smaller than the SENEX offset time. The similar offset times for the four histograms indicate that the times of the

20 maximum MS signal are later than the estimated time by a consistent amount. Once the offset time is taken into account, the times of the maximum MS signals are similar to $t_{est}$ from Eq. (2) for the ammonium nitrate particles. This implies that the two particle velocities are constant when an offset time is included.

The offset time (0.3-0.35 ms) is too long to indicate an error in the positions of the laser or the vaporizer and is independent of

25 particle size, as shown later in Figure 2c. By examining the Gaussian fit reported by Lee et al. (2015), we derived an offset time for that study of about 0.42 ms (see Table 1). The offset time of 0.2 ms used by Cross et al. (2009) was not defined in the same way and cannot be compared directly to offsets from the Gaussian fits. The offset time in our data set appears to be systematic and may be related to a number of delays for the MS signals that do not occur for the LS signals. There is some uncertainty in $t_{LS}$ and $t_{ms}$ due to not knowing exactly when the particles transited the chopper slit (Cross et al., 2007), which is

30 about 0.17 ms for our system. This is apparent in the width of the Gaussian distribution for the ammonium nitrate particles (top of Figure 1). For prompt particles, not including the uncertainty in the time for particles to pass through the chopper, the largest uncertainty in PToF sizing from $t_{ms}$ is believed to be due to vaporization (Huffman et al., 2005; Day et al., 2011). Vaporization event lengths (defined as full width half max of the single particle MS signals) depend on the species and can range from 25 µs for ammonium nitrate particles to less than 60 µs for ammonium sulfate particles (Drewnick et al., 2015).

The additional time needed for the neutral molecules to move from the vaporizer to the electron beam and for ions to move from the ion source to the orthogonal extraction region are likely much shorter than the offset time. Altogether, these times are too short to account for the systematic offset time, so it most likely incorporates some additional time needed to process the single particle mass spectra during data acquisition in LSSP mode, which may be slightly longer for the HR-ToF AMS used

5   by Lee et al. (2015) than for the C-ToF AMS that we used here.

We used the particle velocity calibration for $d_{va-MS}$ to determine $d_{va-LS}$ after taking the offset time into account in the following way. The particle velocity calibration for PtoF sizing of traditional AMS instruments includes this offset time in the measured $t_{MS}$. In contrast, $t_{LS}$ does not include additional time for detecting mass spectral signals, so the calibration coefficients will vary

10  slightly from the traditional $d_{va-MS}$ calibration. To determine the $d_{va-LS}$ particle size based on the $d_{va-MS}$ velocity calibration, an adjustment of the offset time accounting for the chamber dimensions needs to be added to the time of the maximum LS signal:

$$adjusted\ timing\ for\ velocity_{LS} = t_{LS} + t_{offset} \times \frac{L_{LS}}{L_{vp}} \tag{3}$$

$d_{va-LS}$ is then obtained by the $d_{va-MS}$ laboratory calibration values and the derived particle velocity accounting for the offset time as:

15  $$velocity_{LS} = \frac{L_{LS}}{t_{LS} + t_{offset} \times \frac{L_{LS}}{L_{vp}}} \tag{4}$$

We thus have two measurements of vacuum aerodynamic diameter ($d_{va-MS}$ and $d_{va-LS}$) that can provide further information about the LSSP particles detected by the AMS. While this derivation is not as straightforward as a direct calibration of the particle velocity from light scattering for PSL particles, we verified that the method worked well when calculating $d_{va-LS}$ for PSLs using LSSP data from another velocity-$d_{va-MS}$ PSL calibration dataset for a subsequent field project. Furthermore, a direct

20  calibration of $d_{va-LS}$ would be limited to particle sizes above the light-scattering detection limit and cannot include air which is used in the $d_{va-MS}$ calibration. The true particle velocity in the AMS, however, does not include $t_{offset}$ and is obtained simply as $L_{LS}/t_{LS}$, with about ±0.17 ms uncertainty for our chopper duty cycle (2%) and rotational speed (~120 Hz) in the time that the particles pass through the chopper slit (Cross et al., 2007). This timing uncertainty results in a size uncertainty of 9% for both $d_{va-MS}$ and $d_{va-LS}$.

In addition to sizing the particles, the timing differences of the maximum MS signals minus the estimate time for particles to arrive are used to define if particles appear "promptly", are "delayed", or "early." The distinction between these categories of particles is based on if this time difference is within the boundaries of the Gaussian fit (prompt) or outside of the boundary (early or delayed). The boundaries are defined explicitly at $t_{offset}$±3×the Gaussian width, which is the same definition as used

30  by Lee et al. (2015) (see Table 1). Early particles are outside the lower boundary (here it is 0.16 ms) and delayed particles are outside the upper boundary (0.53 ms here). These two boundaries are the vertical cyan lines in Figure 1, and the categories based on these boundaries are described in more detail in later sections. Since very few particles in the SENEX data were early, they are not included in this analysis.

In this case, $t_{LS}$ does not include additional time for detecting mass spectral signals, so the calibration coefficients will vary slightly from the traditional $d_{va}$ calibration. Alternatively, $d_{va-LS}$ can be determined using the same calibration values as $d_{va}$ after accounting for the additional time. Assuming that the velocity of a particle is constant in the vacuum chamber, the estimated arrival time at the vaporizer, $t_{est}$, is calculated from the time the particle passes through the laser beam as:

5 $$t_{est} = t_{LS} \times \frac{L_{vp}}{L_{LS}} \tag{2}$$

where $t_{LS}$ and $L_{LS}$ are defined above and $L_{vp}$ is the distance between the chopper wheel and the vaporizer. For prompt particles (e.g., ammonium nitrate), a histogram of the time differences between the maximum mass spectrum signal time ($t_{ms}$) and the estimated arrival time ($t_{est}$) has a Gaussian distribution (Figure 1). The mean of this distribution is the offset time ($t_{offset}$), and this value for SENEX was 0.35 ms (see Table 1). Twice the width of the Gaussian distribution for prompt particles is

10 approximately the time available for particles to pass through the chopper slit; here it is ~ 0.12 ms. Particles with $t_{ms} > (t_{est} + t_{offset}) + 3\times$the Gaussian width are defined as "delayed" (see Table 1 and Sections 3.1.1 and 3.1.2 below) and are represented by particles on the right hand side of the cyan line at 0.53 ms in Figure 1. $d_{va-LS}$ is then obtained by the $d_{va}$ laboratory calibration values and the derived particle velocity accounting for the offset time as:

$$velocity = \frac{L_{LS}}{t_{LS} + t_{offset} \times \frac{L_{LS}}{L_{vp}}} \tag{3}$$

15 As we show in Section 3.1, $d_0$ is not the optimal measurement for size in this system. Also, the two measurements of vacuum aerodynamic diameter ($d_{va}$ and $d_{va-LS}$) provide additional information on how particles are detected by the AMS.

**3.1.2 Intensities of the LSSP signals, particle $d_0$ sizing, and categories based on intensity**

The intensities of LSSP signals are important for sizing and categorizing particles from the LSSP data sets. The total
20 (integrated) mass spectral (MS) signals from individual particles are shown versus the maximum intensities of their scattered light (LS) signals in Figure 2a for the flight on July 6, 2013. The particles are distinguished by the optical (LS) and chemical ion (MS) detection limits. Particle light scattering signal can be used as an indicator of particle size. In LSSP mode, the light scattering signals above a certain threshold in the data acquisition software were saved and also triggered saving of their mass spectra. This threshold for data saving was set low to save particles with light scatteringscattered light signals near the detection
25 limit and therefore include some of the saved LSSP data that were triggered by noise (<10%). For post-processing, the optical detection limit for SENEX is defined as a maximum scattered light intensity > 0.04 V and a signal-to-noise ratio (S/N) > 3. Less than 10% of the LSSP data were triggered by noise (black points in Figure 2) and are excluded from further analysis.

The maximum intensity of the scattered light signal can be used as an indicator of particle size ($d_0$). The correlation between
30 single particlethe maximum intensity of the single particle scattered light light scattering signals and the derived particle diameter $d_{va-LS}$ from velocity (see method section 2Section 3.1.1) for all LS-triggered events (red, above the optical detection limit, and black, below the optical detection limit) for the flight on July 6, 2013 is plotted in Figure 2ba. The optical detection

 There were a significant number of  light scattering triggers for particles larger than ~ 280 nm ~~light scattering signal above this limit was ~ 170 nm inwhich is close to what was reported by~~ Liu et al.

(2013), who reported that, with  a slightly different criterion (signal-to-noise S/N ≥ 5), the smallest particle ever detected was 180 nm and the 50% detection probability was at 430 nm $d_{va}$.  Figure 2b shows that $d_{va-LS}$ has a generally positive correlation with the maximum intensity of the scattered light  signal  as scattered light signals are approximately proportional to cube of the particle physical diameter.  The maximum scattered light intensities varied over a significant range for the same size particles because the system was not optimized for sizing particles optically. Gaussian fits of individual scattered light pulses from the slowest (75-85 m s$^{-1}$) particles were about 9-12 µs wide and the data from the PMT were recorded every 10 µs, which missed the true maximum of the scattered light pulse for many particles.  The scattered light pulse widths and particle velocities indicate that our laser beam was about 0.7-0.9 mm wide, which is much smaller than for the first LS-AMS study (~2 mm wide, Cross et al., 2007). This is still larger than the calculated particle beam width at the laser position (ranging from 0.13 to 0.59 mm, Huffman et al., 2005), indicating that all of the particles should be passing through the laser beam. Therefore, the maximum scattered light  intensities were only used as a diagnostic and not used to derive particle size in this study.

Particles above the LS detection limit are further categorized by the intensity of their total mass spectral (MS) signals as either being "null" or "chemically detected". Null particles are defined as those above the LS detection limit that do not have total MS signals above the MS detection limit (gray points in Figure 2a). The categorization of the chemically detected particles into prompt (red points in Figure 2a and c) or delayed particles (blue points in Figure 2a and c) is related to the timing of their MS signals (see Section 3.1.1).  The MS detection limit for single particles varied slightly from flight to flight depending on detector sensitivity, and for the flight data shown in Figure 1 it was 600 bit ns particle with a single ion area = 16.9 bit ns ion or about 36 ions in the individual particle mass spectra. The average LSSP MS detection limit for all flights analyzed is 38 ions (see Table 1). For dry particles composed of pure ammonium nitrate, this MS detection limit corresponds to an individual particle with a minimum size $d_p$ of about 200 nm or $d_{va-MS}$ of about 275 nm.

Particles with detectable  LS and MS signals  are subsequently categorized by  the time of the maximum MS signal as prompt or delayed particles (see Section 3.1.1). The  time of single particles as measured by the maximum mass spectral

signals is plotted against the time of maximum intensity of scattered light signals for the flight on July 6, 2013 in Figure 2c. The corresponding $d_{va\text{-}MS}$ and $d_{va\text{-}LS}$ values are plotted on the right and top axis. The solid line, defined as $t_{ms} = t_{est} + t_{offset}$, is the expected time for the particles to arrive at the vaporizer based on Eq. (2) plus the offset time,  and particles with times that fall on or near this line are defined as "prompt" particles. The slope of the solid line is 1.5, as expected by  the ratio of the distances in the AMS between the chopper wheel, laser beam, and vaporizer according to Eq. (2). The intercept is the same offset time as indicated in Figure 1. The dashed line, defined as ($t_{ms} = t_{est} + t_{offset}$ + 3×Gaussian width), is used to distinguish the particles with times for the maximum mass spectral signal  that are significantly later than expected (delayed particles). This delayed time for the maximum mass spectra signal  varied over a wide range, from 0.02—3.1 ms with an average of 1.0 ms. The time delays are also apparent in the Figure 1 histograms of ammonium sulfate, mixed composition, and SENEX data.

Our data are consistent with the hypothesis that delayed particles represent particles that bounce off the vaporizer and subsequently vaporizing on another surface in the source region (Cross et al., 2009). The delays are too long to be explained solely by the time it takes particles to vaporize, for example ammonium sulfate vaporizes in less than 60 µs (Drewnick et al., 2015). Moreover, our laboratory data show that ammonium nitrate, with no measurable bounce, does not produce delayed particles  events (Figure 1). The ion source is less than two centimeters in size, so particles that are delayed 500 to 1000 µs before hitting another surface would have velocities of a few to 20 m s⁻¹. Given the range of particle velocities measured in the PToF region of 70-150 m s⁻¹ for particles between 1000 to 100 nm, respectively, prior to particle impaction on the vaporizer, this represents a loss of as much as 90% of the initial kinetic energy upon a bounce or multiple bounces within the vaporizer. The lost kinetic energy for the particles that bounced may have been converted into plastic deformation and fracturing (Miyakawa et al., 2013). Alternatively, the delays in appearance of the mass spectral signals could have been due to multiple bounces prior to vaporization (Robinson et al., 2017; Hu et al., 2016). Particles without detectible chemical ion signals could be those that bounced far away from the ionization source cage region and could not be vaporized and ionized efficiently.

~~The ion source is designed so that many of the molecules leaving the vaporizer pass through the electron beam. Some bouncing particles will pass through the electron beam and such particles would acquire significant charge (Ziemann et al., 1995). At velocities below very roughly 10 m s⁻¹, the charged particle trajectories can be considerably modified by the electric fields inside the source region, possibly affecting what happens to particles that are not detected promptly by the mass spectrometer. For example, if a 150 nm $d_{va}$ particle with a velocity of 5 m s⁻¹ after bouncing goes through the electron beam, it will probably be pulled through the ion extraction hole. Particles first entering the ionization source are traveling too fast to be deflected from their initial paths if they are charged in the electron beam. Detailed modeling of this effect is beyond the scope of this paper.~~

**3.1.32 Fractions of prompt, delayed, and null particles**

For the particles that are detected by light scattering, the overall fraction of particles that are detected by the mass spectrometer are affected by particle bounce, refractory composition, and/or ion detection limit. The particles that are detected optically but not detected chemically with the mass spectrometer for any of these reasons are defined as "null" particles (see Figure 2a).

5    Prompt and delayed particles are defined above in Section 3.1.1 by the timing of their mass spectral signals (see Figures 1 and 2c). Our previous studies suggest that particle phase is an important factor determining whether or not particles bounce. Larger particles may also have a tendency to bounce more than smaller particles. Some particles contain refractory components (e.g., soot, sea salt, or dust) that do not vaporize at typical AMS vaporization temperatures. If particles are predominantly composed of these species, they will not produce viable mass spectra. The ion detection limit is relevant for particles that are too small

10   to generate sufficient ions from an individual particle. Furthermore, this detection limit varies with the sensitivity of the specific mass spectrometer and likely between different mass spectrometers. Note that the optical detection limit can vary between instruments too.

As discussed later in Sections 3.2.1 and 3.3.2, the in situ CE from the LS-AMS is defined as the fraction of particles that are

15   detected by the mass spectrometer (all particles detected by light scattering minus the particles without viable mass spectra) divided by the total number of particles detected by light scattering. This definition of CE is equivalent to the fraction of prompt plus delayed particles. The mass-based in situ CE detected by light scattering is defined as the mass of particles from their MS signals divided by the mass of particles from their volume determined by their light scattering size, $d_{va\text{-}LS}$, and an estimated density. Thus, both the fractions of these particles and CE from LSSP data are affected by phase, size, composition,

20   and mass spectrometer sensitivity.

According to the chemical ion signal intensities and the relationship between expected and real chemical ion signals arrival time, single particles detected in LSSP mode are classified as prompt, delayed, and null. The criteria for each of these classifications and comparison among different studies are presented in Table 1. An optically detected particle that has

25   chemical ion signals below the threshold is classified as null. Optically and chemically detectible single particles are classified as "prompt" or "delayed" according to their arrival time of maximum mass spectra signal below or above the dashed line in Figure 2b, which are also defined by the time differences in the Figure 1 histograms. Figure 3 shows the particle types (prompt, delayed, and null) as a function of $d_{va\text{-}LS}$ for the research flights from June 26 to July 8. On average the prompt, delayed and null fractions for this study are 27%, 15%, and 58%, respectively, in this study. These fractions are compared with other LS-

30   AMS studies and the details of all definitions for delayed and prompt particles are listed in Table 1. The respective fractions are 23%, 26%, and 51% in Cross et al. (2009) and 46%, 4%, and 48% in Liu et al. (2013). Due to the improvement of LS module and its data analysis software, the definition to separate prompt and delayed particles in this study is the same as the more recent study by Lee et al. (2015) but different from that in Cross et al. (2009) and Liu et al. (2013) (see table 1 for

comparison). The particles were defin by Cross et al. (2009) as those with $t_{ms} > (t_{est}+200$ µs), whereas here they are defined as those with $t_{ms} > (t_{est}+t_{offset}+3\times$Gaussian width $\approx t_{est}+530$ µs) (see Table 1).  Consequently, a higher delayed particle fraction was reported in  Cross et al.'s (2009) study. The definition of delayed particles in Liu et al. (2013) is difficult to directly compare to  our study without information about their offset time. The prompt + delayed fractions are about 50% for both Cross et al. (2009) and Liu et al. (2013). In this study, the combined fraction is  slightly lower (42%), which may be due to a lower sensitivity and a higher MS (chemical) detection threshold that probably lead to higher null rates .

We can get a better understanding of the measured fractions by examining them in more detail. Figure 3 shows the particle types (prompt, delayed, and null) as a function of $d_{va-LS}$ for the research flights from June 26 to July 8. In general, the Prompt and delayed fractions decline at small ($d_{va-LS} < 350$ nm) or large ($d_{va-LS} > 550$ nm) size particles. The single particle mass is near the chemical signal detection limit in this study due to low sensitivity, so the smallest particle size ($d_{va-LS} = 250$ nm) shown here is larger than $d_{va-LS} = 200$ nm observed by Cross et al. (2009) and particles with $d_{va-LS} < 250$ nm in this study are mostly null particles (not shown). This is further supported by lower prompt + delayed fractions at the smallest particle size, depicted in Figure 3, compared to that observed by Cross et al. (2009) and the prompt + delayed fractions at the smallest particle size were even lower for the flights (not shown here) with a lower sensitivity. While the null fraction was relatively high for the 250-300 nm size bin, the number of particles in this bin were quite low, so that the overall fraction of prompt + delayed particles is not significantly affected by single particles near the detection limit of the mass spectrometer for SENEX. The LSSP measurements reported for the Bakersfield study had a larger fraction of null particles at the smallest sizes (Liu et al., 2013) and it is unclear if that may also be related to sensitivity. The reduced prompt + delayed fractions at the largest sizes are similar to what was observed by Cross et al. (2009) and Liu et al. (2013) and are likely due to the larger particles having more kinetic  energy when arriving at the vaporizer or containing more refractory material (e.g. dust) as suggested by Cross et al. (2009). The maximum fraction of delayed particles appeared at a larger size (525 nm) than that of prompt particles (375 nm), which may be a result of more bouncing at larger size.

It is important to note that the single particle mass spectral detection limit affects whether or not individual particles are detected in LSSP mode. However, this is not necessarily a factor for the bulk collection efficiency during normal AMS operation because particles are aggregated in both MS and PToF modes. In other words, while single particles may not be detected individually by the mass spectrometer at the smallest sizes, the mass of small particles can be detected when their signals are added together. This is discussed later with the comparisons of mass distributions in Section 3.2.2.

**3.1. Mass spectral differences between prompt and delayed particles**

Because this study had a significant fraction of delayed particles (about a third of the particles with chemical signals), potential differences in their chemical composition were explored. The average mass spectra of prompt and delayed particles for all SENEX flights analyzed are plotted in Figure 4a. Delayed particles have relatively higher organic signals at $m/z$ 43, 45, and >

5   60, sulfate signals at $m/z$ 98 and 81, and nitrate signals at $m/z$ 46. Prompt particles have relatively higher organic signals at $m/z$ 44, sulfate signals at $m/z$ 48 and 64, and nitrate signals at $m/z$ 30. The chemical ion signals of prompt and delayed particles are different on average, which may indicate the mechanism for producing these delayed particles. Two possible explanations are that the delayed particles vaporized at different conditions because they did not vaporize upon initial impaction or  that chemical and/or physical

10   differences may have caused these particles to have different bouncing characteristics, resulting in different spectra.

To better interpret the difference in spectra of prompt versus delayed particles from SENEX, we conducted limited laboratory experiments using nominally identical particles that produced both prompt and delayed spectra. For these experiments,  dry, poly-dispersed  particles were generated

15   from a simple aqueous mixture of  organic dicarboxylic and carbonyl acids, ammonium organic acid salts, ammonium sulfate, and ammonium nitrate. A total of 1058 light-scattering events were recorded and analyzed using the same criteria as for the SENEX particles except for a lower limit on the number of ions detected in the mass spectra. Of these, about 12% were below the noise level for actual particle light-scattering events. Of the particles above the light-scattering noise, 47% were prompt, 13% were delayed, and 37% were null. The histogram of the time for the maximum MS signal minus

20   the estimated arrival time for these mixed composition particles is included in Figure 1, and is more similar in shape to the SENEX data than either pure, dry ammonium nitrate (narrow, Gaussian distribution) or ammonium sulfate (distribution with a significant trailing edge tail of delayed particles).

The laboratory data here confirm that nominally identical particles can produce differences in the prompt versus delayed

25   spectra. The  mass spectra for the mixed composition particles  are shown in Figure 4b and had similar patterns to the SENEX data, with more prominent peaks at $m/z$ 44, 48, and 64 in the prompt particle spectra and more prominent peaks at $m/z$ 43, 45, 46, 81, 98 and organic peaks with $m/z \geq 60$ in the delayed particle spectra. For these mixed composition particles, the $m/z$ 30 peak was slightly more prominent in the prompt particles and $m/z$ 46 was more prominent in the delayed ones.

30    For the ammonium sulfate   particles shown in Figure 1, the sulfate pattern of high $m/z$ 81, and 98 in the delayed particles and high $m/z$ 48 and 64 in the prompt particles was consistently observed (not shown). The peaks associated with ammonium

(and water) did not appear to show any systematic differences between prompt and delayed particles. For pure ammonium nitrate  particles, none of the LSSP data were classified as delayed (see Figure 1).

 Differences in the spectra of prompt and delayed particles were also recently reported for ammonium sulfate and laboratory-generated secondary organic aerosols (SOA), both with a substantial fraction of delayed particles (Robinson et al., 2017). Indeed, the pattern of sulfate ions between prompt and delayed particles in their study is similar to what is shown in Figure 4 for the SENEX and mixed composition particles and what we observed with our ammonium sulfate particles (not shown), with the peaks at $m/z$ 81 and 98 more prominent in the delayed particles and the peaks at $m/z$ 48 and 64 more prominent in the prompt particles. Hence, it appears that the delayed particles have less fragmentation of sulfate and nitrate ions than the prompt ones. Thus,  difference_s_ between the prompt and delayed spectra in the SENEX data set could be solely due to identical particles vaporizing under distinct conditions. All of the differences in the mass spectra are consistent with more thermal decomposition and fragmentation with prompt particles.

The consistent differences in the mass spectra between prompt and delayed particles from both ambient and nominally-identical chemical composition laboratory particles suggests that there are different processes at the vaporizer for prompt and delayed particles. The two likely explanations are that the evolved gas from the delayed particles experienced fewer wall collisions before ionization and/or that the delayed particles vaporized from surfaces at different temperatures. The bottom of the conical vaporizer, where prompt particles should vaporize, is a location where the evolved gas molecules likely experience some wall collisions and could decompose before ionization. Gas molecules from particles vaporizing from other surfaces, such as the top of the vaporizer or the ionization chamber, would experience fewer wall collisions where they could decompose. For both ammonium nitrate and ammonium sulfate particles, there is far more fragmentation when the vapors are contained in a capture vaporizer prior to ionization compared to the conical vaporizer at the same temperature (Hu et al., 2016).

Vaporizer temperatures also affect fragmentation. Lower vaporizer temperatures increase the ions at $m/z$ 46 relative to $m/z$ 30 for ammonium nitrate and at $m/z$ 80 and 81 relative to the other ions for ammonium sulfate (Hu et al., 2016).  Spectra from most but not all organic compounds have more fragmentation when vaporized at 600 C than at 200 C (Canagaratna et al., 2015). ~~The signal at $m/z$ 44 is more prominent in prompt particles compared to delayed particles. This is consistent with more thermal decomposition and fragmentation to $CO_2^+$ with prompt particles. Prompt particles probably vaporize at higher temperature surfaces resulting in more thermal decomposition. The gas molecules vaporized from prompt particles have a higher temperature and internal energy, and thus may fragment more in the electron beam. Moreover, collisions of vaporized~~

gas with the vaporizer during prompt events can also contribute to additional thermal decomposition. The location where the delayed particles impact and vaporize is likely further away from the vaporizer center which could result in fewer wall collisions of the vaporized species and consequently less potential for additional thermal decomposition. The data showed higher detected signals at $m/z > 60$ for larger molecular weight organic compounds in delayed particles compared to prompt particles, probably resulting from less decomposition and fragmentation of the species from delayed particles. The signal at $m/z$ 44 is more prominent in prompt particles compared to delayed particles. This is consistent with more thermal decomposition and fragmentation to $CO_2^+$ with prompt particles. Hence, the emerging picture is that the observed signal difference in the prompt and delayed particles for SENEX was likely due to the delayed particles vaporizing from a surface at slightly lower temperature than the vaporizer or due to fewer collisions between vaporized gas and hot surfaces.

One piece of evidence about vaporization temperature is the width of the vaporization event, which is wider at lower temperatures and is fairly constant above species-dependent temperatures (Drewnick et al., 2015). For the mixed particles that were generated specifically to compare here with the SENEX data, the peaks in the mass spectral peak widths as a function of vaporization time were analyzed to check for slower vaporization. (Drewnick et al., 2015). A proxy for peak width in the mass spectra (peak area divided by height) was not statistically significant between the prompt and delayed particles, indicating . that the vaporization times were roughly comparable. Since a doubling of the peak widths was not observed for the delayed particles here, it is unlikely that they vaporized at temperatures lower than 300 °C (Drewnick et al., 2015). Because the spectra were saved every 32 μs and vaporization event lengths in the AMS are on the order of 30-60 μs and constant for pure ammonium sulfate particles at temperatures between 400 and 800 °C, the data collected in this brief lab study could not be used to validate the possibility of a less drastic change in differing vaporization temperatures. Equivalent vaporization timescales between prompt and delayed particles along with the same issue of insufficient precision were also reported for the laboratory study of SOA particles (Robinson et al., 2017). The increased fragmentation with a hotter vaporizer could be due to changes in the vaporization process itself or to evolved gas molecules hitting a hotter surface, or both.

While the timing indicates that the delayed particles may have lost a large amount of kinetic energy or bounced multiple times before vaporization, it is unclear where the delayed particles are vaporizing in the vaporization/ionization source region. The microporous vaporizer itself could provide the first surface for particles to strike after an initial bounce. Another possible surface is the baffle which reduces stray light from the hot filament from reaching the LS detector. This baffle has a small hole to transmit particles into the vaporization/ionization source region and roughly forms the side opposite of the vaporizer. Bounced particles could strike the room-temperature baffle if they exit the vaporizer on a trajectory nearly opposite of the initial particle beam. The interior of the ionization chamber is another possible surface. It has ceramic washers and its mounting points are thermally grounded relative to the baffle, filament, and vaporizer. Heat is conducted radiantly between the various hot surfaces (filament and vaporizer) and the ionization chamber, and by conduction through the thin metal of the ionization chamber. The temperature may be different on the sides near and away from the filament. Since we do not have thermocouples

on the ionization chamber or the baffle, there are no direct measurements of their temperatures. This makes it difficult to evaluate the relative importance of vaporization temperature compared to fewer wall collisions of the evolved gases from the delayed particles.

**3.1.54 Aerosol Derived mass difference between from prompt and delayed particles**

The previous section demonstrated that the spectra of prompt and delayed particles identified by the LS-AMS for the SENEX study were different. Here, we examine the measured mass from individual particles distinguished by these two categories. In general, For a given size determined by $d_{va\text{-LS}}$, the chemical ion signals from prompt particles were slightly larger than those from delayed particles for a given size determined by $d_{va\text{-LS}}$. An example comparing the sSingle particle mass obtained from the mass spectrometer chemical ion signals was compared to that derived from aerosol size $d_{va\text{-LS}}$ of prompt (red) and delayed (blue) particles of for the flight on July 6, 2013 is shown in Figure 5. The effective particle density is estimated to be 1.55 g cm$^{-3}$ according to the average ammonium sulfate to organic mass ratio of 0.96, ammonium sulfate density of 1.77 g cm$^{-3}$ and typical organic density of 1.4 g cm$^{-3}$. Single particle masses derived from the mass spectrometer chemical ion signals were well correlated with that derived from measured $d_{va\text{-LS}}$ for prompt particles with a correlation coefficient of 0.87 and a slope of 1.04 with an intercept defined tofixed at be zero0 for this flight. This good correlation between the mass of the particle from the mass spectrum and the size of the particle measured by the timeing of the maximum scattered light signal indicates that LSSP mode is reasonably quantitative on average on a single particle basis. The slope depends on the accuracy of the effective particle density, $d_{va\text{-LS}}$ measurements, and relative ionization efficienciesy (RIEs). For the same size $d_{va\text{-LS}}$, the mean ratio of individual particle mass from chemical ion signals from the delayed particles (blue points) was 0.78 of the mass from prompt particles (red points). This ratio does not depend on the accuracy of the effective particle density, $d_{va\text{-LS}}$, or RIE and was similar in all flights sampling different air masses during this field campaign. While the timing and spectra suggest that these delayed particles vaporized with different conditions, most of the mass was detected.

The delayed particle mass could be lower on average than the prompt particle mass because the evolved gas from the delayed particles was produced in a region where the vapors are not as efficiently ionized by the electron beam. This explanation was recently proposed based on a laboratory study of monodisperse, alpha-pinene secondary organic aerosols (SOA) that had a low number-based CE (0.30), a significant fraction of delayed particles (53% of all LS particles with MS signals), and a clear trend of more ions per particle for prompt particles than for delayed particles (Robinson et al., 2017). Since the SENEX data are from a range of particle sizes, we normalized the total LSSP ion signals to the cube of their $d_{va\text{-LS}}$ and plotted the averages as a function of delay time up to 3.5 ms. Figure 6 shows the results from two flights on July 6 and July 3 along with results from ammonium sulfate particles and confirms that delayed particles have a slightly lower total ion signal per unit volume than prompt particles. When considering only delayed particles, all three cases show a negative slope for the total ion signal per unit volume as a function of delay time, but individually the slopes are not significant or marginally statistically significant at the 2 sigma level. The largest change appears to be an average of ~50% lower signal for particles with the longest delay

times for the flight on July 3. Thus, the efficiency of producing ions from the delayed particles is not always consistently lower than from prompt particles and more experiments are needed to investigate this possible explanation for a reduced mass. It is not clear from the field data why there was a minimal reduction in ion signal; inefficient ionization of the vapors that evolved from delayed particles cannot be precluded.

While the timing and spectra suggest that these delayed particles vaporized outside of the vaporizer at a lower temperature, most of the mass was detected.

The roughly 20% reduction in mass of the delayed particles compared to prompt particles could be partly due to some refractory material in the delayed particles that is not measured by the AMS. The bulk mass fraction of refractory black carbon (rBC) was on average ~1% of the measured aerosol mass for these flights (Warneke et al., 2016), so rBC does not account for all of the reduced mass. In contrast to the SENEX study, the individual particle mass signal for delayed particles during the Mexico City study was less than half of that for the prompt particles (Cross et al, 2009). Although the two studies used different definitions for prompt and delayed particles, changing this definition does not alter the measured average chemical ion signals. Also, both studies used the particle size from light scattering information to calculate the volume. Here the particle size was the vacuum aerodynamic diameter from the maximum time of the maximum scattered light signal ($d_{va\text{-}LS}$) whereas for the Cross et al. (2009) work the volume was determined from $d_{va\text{-}LS}$ and the optical diameter ($d_o$) from the maximum intensity of the scattered light signal. However, these slightly different particle size-based volume calculations between this study and Cross et al. (2009) will not affect the relative signal intensity difference in single particle mass between the prompt and delayed particles. The vaporizers in these two campaigns are designed to be identical. The reasons for much lower delayed particle mass compared to prompt ones remain unclear, but differences in the ambient aerosols measured may contribute to the different delayed particle mass. The Mexico City study was conducted on the ground near the metropolitan area where ~10% of PM2.5 mass was black carbon (Retama et al., 2015) whereas the black carbon was on average ~1% of the mass for SENEX (Warneke et al., 2016). As the null fraction in Cross et al. (2009) was not higher than in this study and the number-based CE was 0.49, the much lower mass from the delayed particles chemical ion signals observed by Cross et al. (2009) could be due to particles containing more refractory material during the Mexico City study than during SENEX.

**3.2 Observations using the AMS LSSP modeCharacterizing LSSP measurements by comparisons with MS, PToF, and UHSAS data**

**3.2.1 Comparing mass fractions between MS and LSSP modes**

One method of evaluating data from the LSSP mode is to compare the average mass fractions of the main species from the LSSP spectra to thoseat measured in the ensemble MS mode. Mass fractions of non-refractory aerosol organic, sulfate, ammonium, nitrate and chloride measured by MS mode versus LSSP mode for all the SENEX flights analyzed (June 26 to July 8) are shown in Figure 76. Data in the MS mode adjacent to the LSSP mode were interpolated and compared to the LSSP

mode data. Considering that the standard deviation of sulfate mass fraction of standard $(NH_4)_2SO_4$ aerosols measured by LSSP mode is about 10% and the MS and LSSP mode were not sampling at the same time during the aircraft measurements, the chemical mass fractions of the various species are well-correlated between the two modes. When averaged, the 30-second single particle data sampled in lower troposphere over the continental US could be representative of the ensemble chemical

5    composition mass fractions. The reasons why the ammonium and nitrate mass fractions were slightly higher in the single particle data are not clear. Since nitrate mass fractions were low for the ensemble data, the uncertainties are relatively larger. Potential explanations for the higher ammonium in the LSSP data may be related to the difference in detected particle size range between LSSP and MS mode or the low sensitivity which could artificially increase the signals for ammonium (Hings et al., 2007) and may be especially important for the single particle data. Overall, the AMS single particle mass fractions are

10   generally comparable to the ensemble measurements, which was also reported previously for Mexico City (Cross et al., 2009). The good correlation in Figure 7 indicates that the LSSP data were on average representative of the bulk relative composition in spite of the sampling and processing biases of the single particle technique.

**3.2.2 Size-resolved mass distribution comparison**

There are three  independent ways to generate mass distributions from an AMS instrument with a light-scattering

15   module: (1) traditional PToF mode distributions, with non-refractory ensemble composition as a function of $d_{va\text{-}MS}$, (2) particle counts from the LSSP mode laser as a function of $d_{va\text{-}LS}$, converted into a mass distribution by assuming an effective particle density of 1.55 g cm$^{-3}$, and (3) LSSP mode mass from the single particle chemical ion signals as a function of $d_{va\text{-}LS}$. For th ese comparisons, the PToF distributions were normalized to the mass loadings from MS mode that were derived from the complete fragmentation patterns in the mass spectra (Allan et al., 2004) and the composition-derived collection efficiency

20   (Middlebrook et al., 2012). The number-based and mass-based LSSP mode mass distributions were scaled to the laser counts from the adjacent MS mode, which did not record $d_{va\text{-}LS}$ and mass spectrum information, to account for the significant time spent in saving the single particle information in LSSP mode. This potentially counted particles that were above the light-scattering threshold set in the data acquisition software yet below the optical detection limit set for analyzing the LSSP data. These particles accounted for 4% of the total LS-triggered events on average and the mass percentage from these particles

25   would be much smaller considering that they are the small particles below optical detection limit. Details on the LSSP mass distribution calculations are in the SI. Figure 8a shows these three mass distributions from ambient air  below 3000 m in altitude during the flight on July 6, 2013. A calculated mass distribution from the ultra-high sensitivity aerosol size spectrometer (UHSAS) instrument (solid black curve) is also depicted in Figure 8a, where the UHSAS number distribution is multiplied by the AMS lens transmission efficiency (Liu et al., 2007; Bahreini et al., 2008) and converted to mass as a

30   function of $d_{va}$ using Eq. (1) and by assuming an effective particle density of 1.55 g cm$^{-3}$.

The four curves in Figure 8a demonstrate various properties of the LS- AMS system. The LSSP mode number-based mass distribution (red curve) compared to that from the UHSAS instrument (black curve) indicates that the AMS laser

system here accounted for the mass from most aerosol particles with $d_{va-LS}$ > 440 nm and essentially none of the mass from particles smaller than $d_{va-LS}$ < 280 nm.  Although ~~particles as small as $d_{va}$ ~170 nm couldthe~~ LSSP data acquisition were recorded for $d_{va-LS}$ as small as ~ 170 nm , a very small number of these particles were detected and there generally were no corresponding MS signals until the particles  were larger than $d_{va-LS}$ ~280 nm. The laser and optics used in the LS-AMS are clearly not optimized to detect small ambient particles. Because the two (black and red) distributions are nearly identical for $d_{va-LS}$ > 600 nm, the standard AMS lens transmission function (Liu et al., 2007) appears to be valid for the upper size range. The particle mass from chemical ion signals (grey) is lower than the laser counted particle mass (red) because not all of the particles optically detected produced detectible chemical signals. The ratio of the gray shaded area to the area under the red curve in Figure 8a  is a mass-based measure of the collection efficiency and is discussed further in Section 3.3.2. The uncertainty introduced by including the particles with scattered light signals below the detection limit for normalizing to the LS counts from the adjacent MS cycles is eliminated when calculating this mass ratio. The AMS PToF mass distribution (dashed curve) has  more mass from particles at the large size end (> 700 nm) than all of the other distributions  (discussed further below in Sect. 3.3.1).

**3.3 Relevance of LSSP results to standard AMS measurements**

**3.3.1 Measurements of the traditional AMS vacuum aerodynamic diameter $d_{va-MS}$**

All particles that scatter light are included in the size distributions in Figures 8b and 8c to highlight the differences between using the time of the maximum scattered light signal to size the particles ($d_{va-LS}$ in Figure 8b) or the time of the maximum mass spectral signal ($d_{va-MS}$ in Figure 8c).  The delayed particles create a bias towards the larger size end of the traditional PToF mass distribution. The mass from the traditional PToF distribution (Figure 8a, dashed curve) is higher than all the other curves for particles larger than 700 nm even considering the overall uncertainty.  Compared to the distributions plotted as a function of $d_{va-LS}$, mass distributions as a function of $d_{va-MS}$ have less mass for the intermediate sizes ($d_{va-MS}$ ~ 350 to 500 nm) and more mass at the larger sizes ($d_{va-MS}$ > 700 nm).

On the small size of the mass distributions sizes ($d_{va-LS}$ ~ 100 to 300 nm), there is additional mass measured in the PToF mode that does not appear in the LSSP data (Figure 8a). This is not a bias in the PToF data at small sizes because it is also observed in the UHSAS data. These particles are too small for the scattered light signal to consistently trigger saving data in LSSP mode (see Figure 2) and it is uncommon for particles to appear in the early part of the 8.3 ms long chopper cycle, even for particles such as pure ammonium sulfate that have a high tendency to bounce (Figure 1). Furthermore, the PToF data are acquired by

aggregating the bulk mass spectral signals over the sampling period rather than by aggregating single particle mass spectral signals or individual particle counts from the laser. Thus, the PToF mode measures the mass from small particles which are not large enough to efficiently scatter light in the LSSP mode or generate enough ions for a clear signal from a single particle.

5    The broadening of the traditional AMS mass distribution to larger sizes due to delayed particles was also observed during the Mexico City and Bakersfield field studies (Cross et al., 2009; Liu et al., 2013) and the alpha-pinene SOA laboratory study (Robinson et al., 2017). Thus, $d_{va\text{-}LS}$ instead of $d_{va\text{-}MS}$ is a more reliable parameter to represent particle size in the AMS. On average, the relative composition as a function of size for the different distributions does not appear substantially different. The bias toward increased mass at the larger sizes from delayed particles needs to be considered when interpreting standard
10    AMS PToF mass distributions data, especially if corrections to account for inlet transmission efficiency have been applied at large sizes.

**3.3.2 Comparing the LSSP collection efficiency toAccuracy of the AMS parameterized collection efficiency**

The light scattering module can be used to measure in situ AMS collection efficiency (CE), defined here as the ratio of number (or mass) of particles with both detectable chemical (prompt and delayed) and optical signals to the total number (or mass) of
15    all particles with detectable optical signals. The number-based CE from the LSSP data does not include counts from saved data where the light scatteringscattered light signal was close to the noise level (optical detection limit) and is defined here as the (prompt + delayed) particle counts divided by (prompt + delayed + null) counts. The mass-based CE from LSSP data is defined in Section 3.2.2 as the ratio of the particle mass from the chemical ion signals (e.g., mass distribution shown as the gray area in Figure 8a) to the particle mass from the laser counts (e.g., the area under the red curve in Figure 8a).
20    Both number- and mass-based LSSP mode CE values are calculated for the 30-second intervals average of LSSP data every five minutes. As mentioned before, because significant time was needed to save single particle optical and chemical information, LSSP mode did not record all particles sampled. There is an assumption for the measured CE that the undetected mass is the same as the detected mass, which is likely true due to random detection. It is also assumed that particles detected optically are representative of all particles sampled by the AMS and have the same chemical composition as the particles that
25    are too small to be detected by LSSP mode. In air masses where newly formed and growing (Aitken mode) particles are present, this assumption is not necessarily valid.

The CE measurements based on LSSP particle number or mass varied from about 0.2 – 0.9 for this study. The flight that had the largest range of CE from about 0.4 to 0.9 based on the measurements was on July 6, 2013 and is shown in Figure 8.
30    Besides the CE measurements, the CE parameterization based on ensemble measured chemical composition from MS mode (Middlebrook et al., 2012), which is commonly used in AMS data analysis software, is also plotted in Figure 8. In contrast to CE measurements from LSSP mode data, the CE parameterization covers the entire particle size range detected in PToF mode (Figure 7a). The three values for CE shown in Figure 8 were correlated and generally agreed considering experimental

uncertainties for this flight.  Both number- and mass-based LSSP mode CE values for all the flights studied here are calculated for the 30-second intervals average of LSSP data every five minutes and are shown in Figures 9a and S3. As mentioned before, because significant time was needed to save single particle optical and chemical information, LSSP mode did not record all particles sampled. All of the CE measurements based on LSSP particle number or mass varied from about 0.2 – 0.9 for this study. The flight that had the largest range of CE from about 0.4 to 0.9 based on the LSSP measurements was on July 6, 2013 and is emphasized in Figure 9.

The error bars on the number-based LSSP CE values in these figures (blue points) are the statistical variations of CE ($\sigma_{CE}$) for the LSSP data  and  is estimated to be less than ±0.08 for the 5 min average data based on variations of a binomial distribution as:

$$\sigma_{CE} = \frac{\sqrt{np(1-p)}}{n} \tag{4}$$

where n is number of particles with optical signals above detection limit, and p is the probability of optically detected particles that can be chemically detected, varying from 0.3 to 0.9. In addition to this statistical variability, the mass-based CE from the LSSP data has as much as 27% uncertainty from the measured particle volume from 9% uncertainty in $d_{va\text{-}LS}$. The error bars for the mass-based LSSP CE (yellow points in Figures 9 and S3) were not included in the figures for clarity.

 The large range of CE  values for the flight on July 6 were clearly not due to statistical variation. For this flight, the ratio of MS-mode ammonium to predicted ammonium from full neutralization of sulfate plus nitrate varied more than for all of the other flights, indicating that the aerosol on this flight was at times significantly more acidic on average than for other flights. The relative humidity for this particular flight was also a bit higher on average than the other flights. Thus, the acidity and relative humidity likely had an influence on the CE for this flight more than on the other flights.

 The parameterization of CE based on the bulk MS mode composition data incorporates variations in CE due to acidity and sampling line relative humidity (Middlebrook et al., 2012) and is commonly used in AMS data analysis software.  In general, the good correlation indicates that aerosol chemical composition and relative humidity dependent CE parameterization (Middlebrook et al., 2012) can accurately capture the general variability of CE at least for the cases when significant variation in CE is due to change of aerosol acidity. CE values above 0.5 were primarily due to the presence of acidic sulfate particles during this flight and were

 In contrast to CE measurements from LSSP mode data, the CE parameterization covers the entire particle size range detected in MS or PToF mode (Figure 8a). Since the bulk MS mode data are saved every 10 s, the parameterized CE is calculated for that time interval and is shown in Figures 9 and S3 as red lines connecting points.

There are many factors influencing the point-by-point CE comparison shown in Figure 9b and all three types of CE determinations have limitations. The CE parameterization has about 20% uncertainty (based on Middlebrook et al., 2012) and it could contribute to the noise in the red traces of Figures 9a and S3. In addition, there are statistical variations on the LSSP-based CEs as described above. Also we did not parse the data sets for low statistics from either low LS particle counts or low total bulk mass, which is approximately only an issue for the high-altitude data points. LSSP and MS data were not obtained at the same time and the air masses sampled could be changing rapidly since the aircraft is moving about 100 m s$^{-1}$, such that each CE data point from the parameterization is about 1 km apart, each LSSP CE data point represents a 3 km average, and each LSSP CE data point is 30 km apart. Two minute averages of the CEs from the parameterization were used to generate the comparison plot in Figure 9b. For this flight on July 6, the (observed or calculated) range in CE is much larger on average than for the other flights, it varied on larger temporal (spatial) scales, and the CE variability was outside of the combined error bars.

Given all of these uncertainties, the three values for CE shown in Figure 9 were correlated and generally agreed well. CE values above 0.5 were primarily due to the presence of acidic-sulfate particles during this flight and were determined by both composition-dependent CE parameterization from the ensemble MS mode data and in situ LSSP mode measurements. The good correlation indicates that the aerosol chemical composition and relative humidity dependent CE parameterization (Middlebrook et al., 2012) can accurately capture the general variability of LSSP-based CEs at least for the cases when significant variation in CE is due to change of aerosol acidity.

The default CE parameterization value of 0.5 may be too high during some parts of this flight and large parts of other flights (Figure S3). This was also observed with other field data using a mass-based comparison to evaluated CE (Middlebrook et al., 2012) and may indicate that the default CE of 0.5 is slightly high in the parameterization.  One flight in particular (July 3) had the highest null fractions (Figure 3) and corresponding lower CEs for most of the flight compared to other flights. This flight had an overall higher mass loading of refractory black carbon (rBC) from biomass burning in the sampled air (0.36 µg sm$^{-3}$, whereas the average for all of the flights analyzed here was 0.14 µg sm$^{-3}$). If an individual particle is mostly rBC, there may not be enough ion signal from the non-refractory components to detect it with the mass spectrometer. In Toronto, a higher null fraction was measured when urban, rBC-containing particles were evaporated with a vaporizer instead

of an infrared laser (Lee et al., 2015). It is also possible that the collection efficiency is lower for the organic fraction of biomass burning aerosols. The effect of biomass burning particles on the measured CE with the LS-AMS needs further investigation.

**3.3.3 Additional considerations for the LSSP-based collection efficiencies**

5    CE is traditionally defined based on mass comparisons. There is an assumption in applying either a calculated or an in situ CE to the measured mass loadings from MS mode that the chemical composition of the undetected mass is the same as the detected mass, which is likely true during sampling of air masses with mostly mixed secondary aerosol particles. In general for SENEX, the number- and mass-based CE from the LSSP data shown in Figure 9 and Figure S3 are comparable within experimental uncertainties for the particles sampled. In Figure 8a, the integrated mass from the PToF mass distribution using the average

10   CE from the parameterization of 0.6 for the flight on July 6 (dashed curve) is also within the combined experimental uncertainties of the integrated mass from the UHSAS mass distribution (solid black curve). The flight-averaged number-based CE was 0.58. Hence, the SENEX field data did not show any large discrepancies between the number- and mass-based CEs when averaged for the entire flight.

15   It is also assumed that particles detected optically are representative of all particles sampled by the AMS and have the same chemical composition as the particles that are too small to be detected by LSSP mode. In air masses where newly formed and growing (Aitken mode) particles are present, this assumption is not necessarily valid. The number- and mass-based CEs may be different if there are significant differences in the number-based CE as a function of size, as briefly described by Huffman et al. (2005). In power plant plumes that were sampled by the PToF mode during SENEX, we sometimes observed smaller

20   acidic sulfate particles with larger mixed-composition particles. Because these power plant plumes were transected quickly by the aircraft, the small particles were not sampled consistently with LSSP mode, and so the effect of a varying composition with size could not be evaluated here.

     While the aircraft data reported here show a wide range of CE due to air mass variations, such variability in the LSSP mode

25   CE has not been reported previously. For comparison, mass-based and number-based CEs have been reported from other studies. The Bakersfield study described a discrepancy between the average number- and mass-based CEs, where the number based value was ~0.5 and the mass-based value from ensemble measurements was 0.8 (Liu et al., 2013). The authors proposed that a mismatch of vaporization and data acquisition time scales reduced the detected chemical ion signals from single particles compared to the ensemble measurements; yet this discrepancy was not resolved. The in situ CE from LSSP mode

[remaining 105,822 characters of this post omitted]